# Chromium-ruthenium oxide solid solution electrocatalyst for highly efficient oxygen evolution reaction in acidic media

Yichao Lin[1,2], Ziqi Tian[1,2], Linjuan Zhang[3], Jingyuan Ma[4], Zheng Jiang [4], Benjamin J. Deibert[5], Ruixiang Ge[1,2] & Liang Chen[1,2]

The development of active, acid-stable and low-cost electrocatalysts for oxygen evolution reaction is urgent and challenging. Herein we report an Iridium-free and low ruthenium-content oxide material ($Cr_{0.6}Ru_{0.4}O_2$) derived from metal-organic framework with remarkable oxygen evolution reaction performance in acidic condition. It shows a record low overpotential of 178 mV at 10 mA cm$^{-2}$ and maintains the excellent performance throughout the 10 h chronopotentiometry test at a constant current of 10 mA cm$^{-2}$ in 0.5 M $H_2SO_4$ solution. Density functional theory calculations further revealed the intrinsic mechanism for the exceptional oxygen evolution reaction performance, highlighting the influence of chromium promoter on the enhancement in both activity and stability.

[1] Ningbo Institute of Materials Technology and Engineering, Chinese Academy of Sciences, 315201 Ningbo, Zhejiang, China. [2] University of Chinese Academy of Sciences, 100049 Beijing, China. [3] Shanghai Institute of Applied Physics, Chinese Academy of Sciences, 201800 Shanghai, China. [4] Shanghai Synchrotron Radiation Facility, Shanghai Institute of Applied Physics, Chinese Academy of Sciences, 201204 Shanghai, China. [5] Department of Chemistry and Chemical Biology, Rutgers, The State University of New Jersey, Piscataway, NJ 08854, USA. These authors contributed equally: Yichao Lin, Ziqi Tian. Correspondence and requests for materials should be addressed to L.C. (email: chenliang@nimte.ac.cn)

Oxygen evolution reaction (OER) or the water oxidation plays a key role in clean energy technologies, including hydrogen production through water electrolysis, electrochemical or photoelectrochemical $CO_2$ reduction and reversible fuel cells for production of clean electricity[1–3]. Essentially, the process of OER is a four electron and four proton coupled electrochemical reaction, demanding a higher energy (i.e., higher overpotential) to overcome the kinetic barrier than the hydrogen evolution reaction (HER), which is a two electron-transfer reaction. In the past decades, substantial research effort has been devoted to the design and development of OER electrocatalysts with enhanced electrode kinetics and stability. To date, various OER catalysts, such as transition metal oxides[4–6], perovskite[7,8], and layered structure materials[9,10], have been reported. However, these OER electrocatalysts still suffer from sluggish kinetics and/or low stability in acidic media. Compared with alkaline conditions, OER catalysis under acidic conditions is much more preferable because acidic electrolyte has higher ionic conductivity and fewer unfavorable reactions[11,12]. In addition, commercially available water electrolysis assemblies use cation exchange membrane, e.g., Nafion, as the ionic conductor, which requires OER to be operated in acidic environment. Unfortunately, most of the known active metal oxides cannot survive under harsh acidic operating conditions. Currently, rutile-structured ruthenium (Ru) and iridium (Ir) oxides are the two best catalysts for OER in acidic media[13–15]. It is widely accepted that $RuO_2$ has higher activity but lower stability than $IrO_2$[16–18]. Thus, to develop OER catalysts with both high activity and stability, the use of mixed phase or solid solution of $RuO_2$ and $IrO_2$ has been investigated[19–22]. Very recently, three new types of Ir-based double perovskites[23], multiphase $IrNiO_x$ or $IrO_x/SrIrO_3$[24,25], and pyrochlores-structured Ir-based oxides[26], were reported to be active and relatively stable toward OER in acidic media. However, we note that little attention has been paid to design cheaper Ru-based electrocatalysts, particularly with low Ru-content, for OER in acidic condition[27]. Indeed, it is desirable to modulate the electronic structure by replacing part of Ru with suitable transition metals in order to improve the OER activity. Furthermore, the replacement by cheaper transition metal is also advantageous in terms of cost.

Metal-organic frameworks (MOFs), a unique type of porous materials with ultrahigh porosity, tunable pore sizes and morphology, and well-characterized crystalline architectures, have emerged as excellent templates or platforms for preparing electrocatalysts with high performance, such as N-doped porous carbon, metal oxide nanocomposites[3,28]. In light of these successful studies, we propose to design Ru-based electrocatalysts based on MOF templates, which can make use of the porosity to load Ru sources and the original metal node as promoter. Herein, we present a low-cost Ir-free rutile-structured chromium-ruthenium oxide electrocatalyst (i.e., $Cr_{0.6}Ru_{0.4}O_2$) derived from MIL-101 (Cr) which exhibits record low overpotential and high stability toward OER in acidic media. We chose MIL-101 (Cr) because of its ultra-high surface area (above 3000 $m^2\,g^{-1}$) and large pore sizes (2.9–3.2 nm) that can facilitate the loading of Ru precursors[29]. Moreover, density functional theory (DFT) calculations suggested that Cr plays a critical role in improving the stability and OER activity by tuning the electronic structure of $RuO_2$ phase. The resulting $Cr_{0.6}Ru_{0.4}O_2$ electrocatalyst exhibits an overpotential of 178 mV at 10 mA $cm^{-2}$, a small Tafel slope (58 mV $dec^{-1}$), and stable chronopotentiometric performance under 10 mA $cm^{-2}$ in 0.5 M $H_2SO_4$ solution for 10 h, which outperforms the most active OER electrocatalysts reported to date, such as $BaYIrO_6$[23], $IrO_x/SrIrO_3$[25], and $Y_2Ru_2O_{7-\delta}$[27].

## Results

**Preparation and characterization of RuCl₃-MIL-101(Cr) and Cr₀.₆Ru₀.₄O₂.** The route to the preparation of $RuCl_3$-MIL-101 (Cr) precursor and $Cr_{0.6}Ru_{0.4}O_2$ powders is illustrated in Fig. 1. $RuCl_3$ was firstly loaded into the pores of MIL-101 (Cr) by means of impregnation. After loading $RuCl_3$, the color of MIL-101 (Cr) changed from light green to brown (the color of $RuCl_3$) (Supplementary Figure 1), visually indicating the successful loading of $RuCl_3$. The resulting $RuCl_3$-MIL-101 (Cr) composite was further annealed under air at temperatures between 450 and 600 °C for 4 h to fabricate $Cr_{0.6}Ru_{0.4}O_2$ powders. $RuCl_3$-MIL-101 (Cr) was characterized using a combination of power X-ray diffractions (PXRD), scanning electron microscopy (SEM), inductively coupled plasma-mass spectroscopy (ICP-MS) and $N_2$ adsorption/desorption measurements at 77 K. As shown in Fig. 2a. PXRD pattern of the resulting $RuCl_3$-MIL-101 (Cr) was essentially identical to that of original MIL-101 (Cr), suggesting that the crystalline structure of MIL-101 (Cr) was preserved after loading $RuCl_3$. The reduced intensity of the peaks below 7° after loading $RuCl_3$ can be attributed to the pore filling of MIL-101 (Cr), which has also been observed in PEI incorporated MIL-101 (Cr)[30]. There was no peak for $RuCl_3$, indicating that $RuCl_3$ did not crystalize in the pores of MIL-101 (Cr) but was adsorbed on the pore surface. SEM characterization was conducted to analyze the morphology of MIL-101 (Cr) before and after loading $RuCl_3$. As displayed in Fig. 2b, MIL-101 (Cr) has an octahedral morphology with small particle size (~100 nm), which can effectively facilitate the diffusion of $RuCl_3$ into MIL-101 (Cr) pores. After loading $RuCl_3$, morphology change of MIL-101 (Cr) was not observed. ICP-MS was employed to evaluate the loading amount of $RuCl_3$ in MIL-101 (Cr). The measured atomic ratio of Cr/Ru was 6:4, corresponding to 37.8 wt% $RuCl_3$ content in $RuCl_3$-MIL-101 (Cr). $N_2$ adsorption/desorption measurements of MIL-101 (Cr) and $RuCl_3$-MIL-101 (Cr) were further conducted to evaluate their surface area and pore volume (Supplementary Figure 2). MIL-101 (Cr) exhibited a saturated $N_2$ uptake of 1050 $cm^3\,g^{-1}$, which was consistent with values reported in literatures[31,32]. The corresponding pore volume and BET surface area were calculated to be 1.63 $cm^3\,g^{-1}$ and 3373 $m^2\,g^{-1}$, respectively. Upon the loading of $RuCl_3$, the pore volume and Brunauer−Emmett−Teller (BET) surface area were decreased to 0.97 $cm^3\,g^{-1}$ and 1783 $m^2\,g^{-1}$, respectively.

Fine powders with the composition of $Cr_{0.6}Ru_{0.4}O_2$ were obtained by annealing $RuCl_3$-MIL-101 (Cr) under air for 4 h at a series of temperatures between 400 and 650 °C, denoted as $Cr_{0.6}Ru_{0.4}O_2$ (T, T is the annealing temperature). As shown in the PXRD patterns (Fig. 2c), the increased intensity of peaks with annealing temperature indicates that the higher annealing temperature can lead to better cystallinity of $Cr_{0.6}Ru_{0.4}O_2$. When the annealing temperature was lower than 450 °C, very poor crystalline samples were formed. The PXRD patterns of $Cr_{0.6}Ru_{0.4}O_2$ powders annealed above 500 °C are essentially identical, and can be indexed as a solid solution of rutile $CrO_2$ and $RuO_2$ with tetragonal system and P42/mnm space group (the refined lattice parameters are listed in Supplementary Table 1 and the standard PXRD patterns of $CrO_2$ and $RuO_2$ were shown in Supplementary Figure 3a for comparison). The structure of $Cr_{0.6}Ru_{0.4}O_2$ is refined by Rietveld refinement (Supplementary Figure 3b). As shown in Fig. 2d, Cr and Ru atoms are randomly distributed in the metal sites of the $Cr_{0.6}Ru_{0.4}O_2$ lattice. These metal atoms are edge−sharing and octahedrally coordinated to form chains along the [0 0 1] direction. Each chain is connected to four neighboring chains by shared corners. The $MO_6$ octahedra are tetragonally distorted, thus these M−O bond distances are not equal. SEM images show that the morphologies of $Cr_{0.6}Ru_{0.4}O_2$ powders became smaller, and their surface became much rougher

after annealing (Fig. 3a and Supplementary Figure 4). Transmission electron microscopy (TEM) images indicate that the individual particles are composed of much smaller nanocrystals (~15 nm) (Fig. 3b–d, Supplementary Figures 5-8). High resolution TEM (HR-TEM) image (Fig. 3e) and the corresponding fast Fourier transform (FFT, Fig. 3f) indicate that these nanocrystals are single-crystalline. Between these nanocrystals in a single $Cr_{0.6}Ru_{0.4}O_2$ particle, many mesopores exist, facilitating the mass transfer in the OER process. The $N_2$ adsorption/desorption isotherms further confirm that $Cr_{0.6}Ru_{0.4}O_2$ powders are porous with BET surface areas between 50 and 90 $m^2 g^{-1}$ (Fig. 2e and Supplementary Table 2). Barrett–Joyner–Halenda (BJH) pore size analysis reveals that the pore sizes of $Cr_{0.6}Ru_{0.4}O_2$ particle are larger than 10 nm (generated from the aggregation of nanocrystals in an individual particle as shown in TEM images) and increase with the annealing temperature (Supplementary Figure 9). This trend can be ascribed to the larger volume contraction of $Cr_{0.6}Ru_{0.4}O_2$ nanocrystals within a single particle at higher annealing temperature. High-angle annular dark-field scanning transmission electron microscopy (HAADF-STEM) was employed to analyze the element distribution in a single nanocrystal. The resulting EDS mapping images (Fig. 3g) show that Cr, Ru, and O are uniformly distributed over the entire $Cr_{0.6}Ru_{0.4}O_2$ nanocrystal, demonstrating the formation of a single phase of Cr and Ru oxide solid solution (the mapping images for a wider region are shown in Supplementary Figure 10). In addition, the EDS analysis indicates that Cr/Ru ratio is 0.56:0.44, generally consistent with the ICP-MS result (Supplementary Figure 11). Furthermore, we performed atomic-resolution HAADF-STEM and electron energy loss spectroscopy (EELS) mapping characterization. As shown in Fig. 3h–j, the atomic-solution HAADF-STEM images clearly demonstrate the well crystalized single nanocrystals. EELS analysis of a randomly selected region in a single nanocrystal confirmed the coexistence of Ru and Cr atoms. The corresponding EELS elemental mapping with subnanometer resolution (Fig. 3k) also shows a uniform uncorrelated spatial distribution of Cr, Ru, and O.

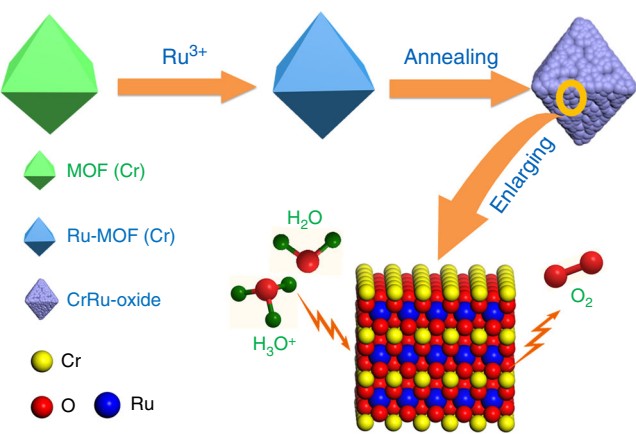

**Fig. 1** Schematic illustration of the preparation of $Cr_{0.6}Ru_{0.4}O_2$ electrocatalysts for OER application in acid media

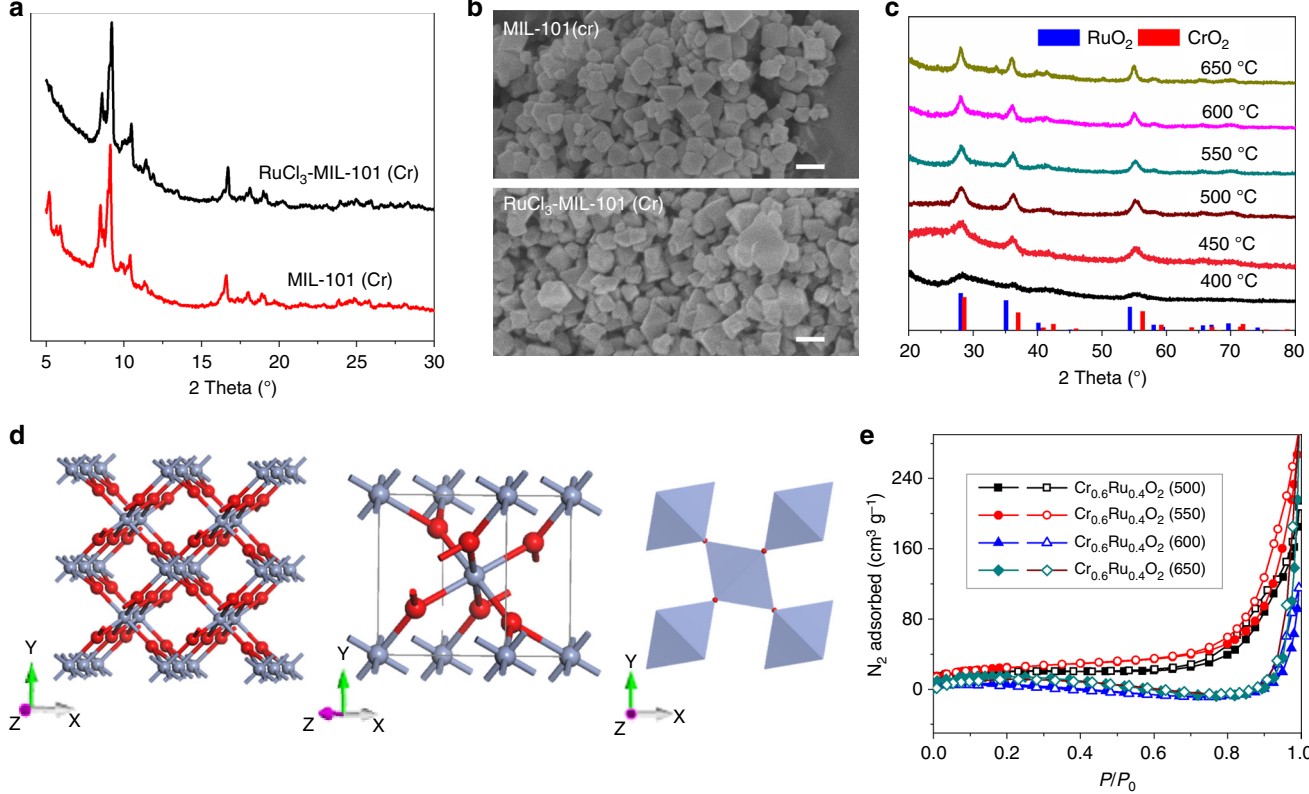

**Fig. 2** Structural characterizations of RuCl₃-MIL-101(Cr) and $Cr_{0.6}Ru_{0.4}O_2$ (550). **a**, **b** PXRD patterns and SEM images of MIL-101 (Cr) before and after loading RuCl₃ (scale bars, 200 nm); **c** PXRD patterns of $Cr_{0.6}Ru_{0.4}O_2$ powders annealed at different temperatures. The reference patterns of $CrO_2$ and $RuO_2$ were obtained from Jade 2004 (JCPDS No.09-0332 and 43-1027); **d** Crystal structure of $Cr_{0.6}Ru_{0.4}O_2$ (550): (left) packing image, (middle) unit cell, (right) corner shared octahedral $MO_6$ structure. Color code: blue (60% Ru, 40% Cr), red (O). **e** 77 K $N_2$ adsorption/desorption isotherms of $Cr_{0.6}Ru_{0.4}O_2$ powders annealed at different temperatures

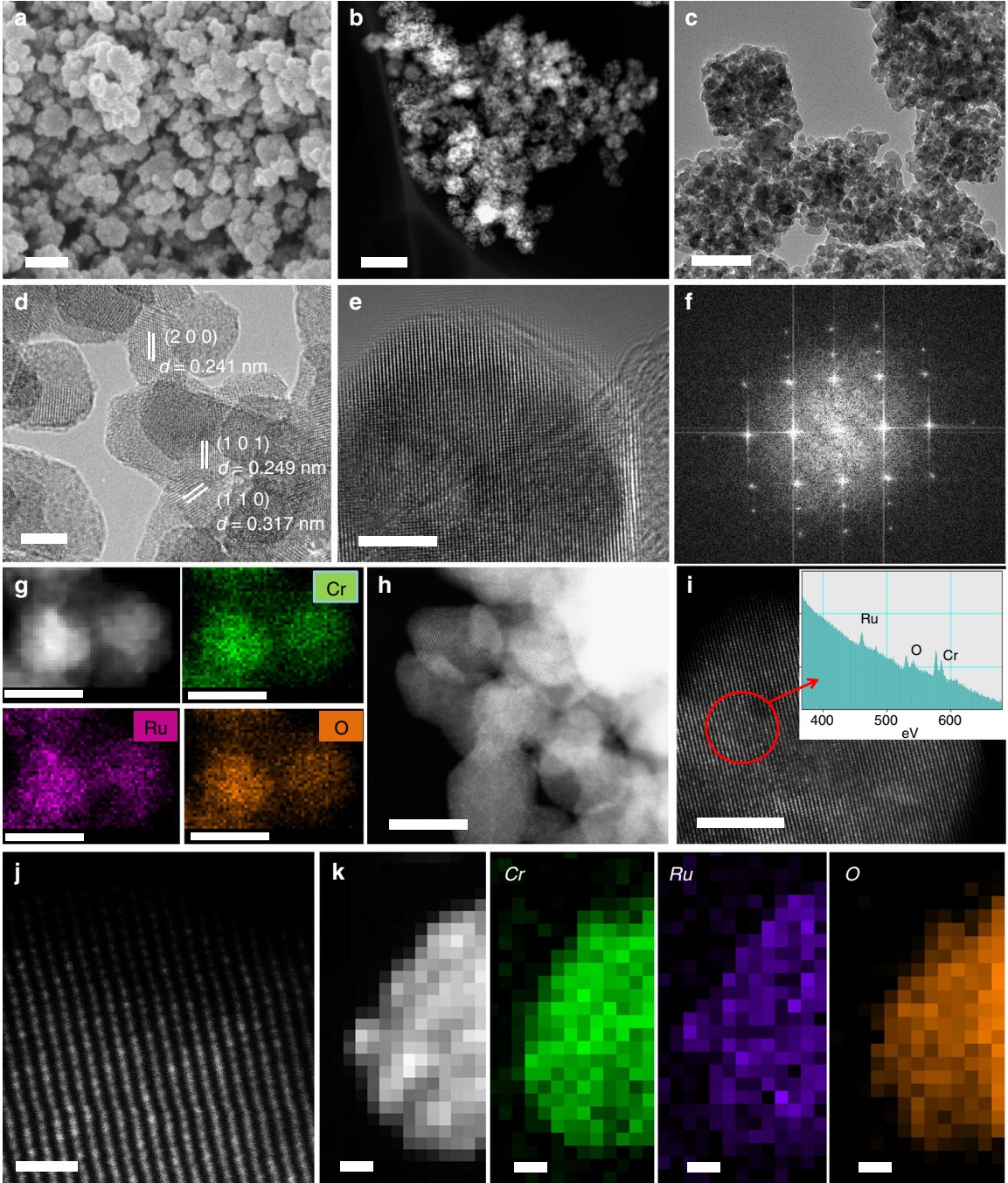

**Fig. 3** Morphology and elemental mapping images of $Cr_{0.6}Ru_{0.4}O_2$ (550). **a** SEM image (scale bar, 200 nm); **b** Dark field TEM image (scale bar, 200 nm); **c** TEM image (scale bar, 50 nm); **d** HR-TEM image (scale bar, 5 nm); **e** HR-TEM image of a single nanocrystal (scale bar, 5 nm); **f** The corresponding FFT image; **g** HAADF-STEM image, corresponding EDS element mapping showing the distribution of Cr, Ru, and O (scale bars, 10 nm); **h–j** atomic-resolution HAADF-STEM images and EELS analysis (inset of i), scale bars: 10, 5, and 1 nm, respectively; **k** EELS maps (scale bars, 1 nm)

**OER catalytic performance in strong acidic media.** The OER activity of $Cr_{0.6}Ru_{0.4}O_2$ powders annealed at different temperatures was studied in a strong acidic media (0.5 M $H_2SO_4$). The $Cr_{0.6}Ru_{0.4}O_2$ based electrodes were prepared by drop-casting a water/ethanol and Nafion-based ink of $Cr_{0.6}Ru_{0.4}O_2$ on the glassy carbon disk (see more details in methods section). Figure 4a shows the linear sweep voltammetry (LSV) results, where the rising current indicates the region where oxygen evolution occurred. $Cr_{0.6}Ru_{0.4}O_2$ (450), $Cr_{0.6}Ru_{0.4}O_2$ (500) and $Cr_{0.6}Ru_{0.4}O_2$ (550) exhibit excellent initial OER activities, with onset potential of ~1.33 V vs. RHE, which represents an overpotential of ~100

mV. In addition, according to the suggested benchmark criteria[33], $Cr_{0.6}Ru_{0.4}O_2$ (450), $Cr_{0.6}Ru_{0.4}O_2$ (500), and $Cr_{0.6}Ru_{0.4}O_2$ (550) exhibited overpotentials of 175, 178 and 178 mV at the current density of 10 mA cm$^{-2}$, respectively. $Cr_{0.6}Ru_{0.4}O_2$ (600) and $Cr_{0.6}Ru_{0.4}O_2$ (650) show slightly higher OER overpotentials (186 and 200 mV at 10 mA cm$^{-2}$, respectively), but still lower than those reported in literatures[23,25]. Note that there is little difference in the PXRD patterns for $Cr_{0.6}Ru_{0.4}O_2$ electrocatalysts annealed above 500 °C, the slightly lower OER activity for $Cr_{0.6}Ru_{0.4}O_2$ (600) and $Cr_{0.6}Ru_{0.4}O_2$ (650) might be ascribed to the lattice strain, which was also observed on $IrO_2$[34].

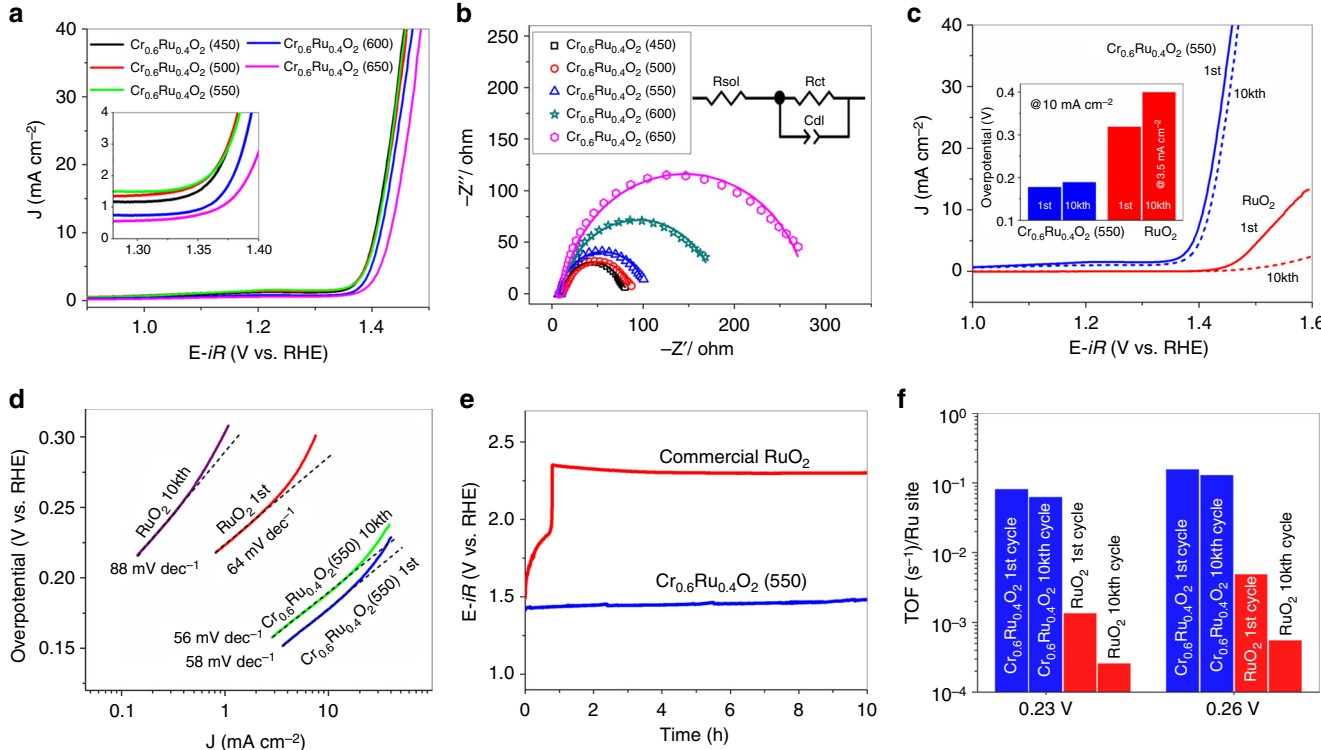

**Fig. 4** OER activity of $Cr_{0.6}Ru_{0.4}O_2$ annealed at different temperature. **a** Electrocatalytic OER activities of $Cr_{0.6}Ru_{0.4}O_2$ (450–650) nanoparticiles; **b** Nyquist plots at 1.395 V. Solid curves are the fitting results by using the equivalent circuit shown in the inset; **c** LSVs of $Cr_{0.6}Ru_{0.4}O_2$ (550) and commerical $RuO_2$ for the first and 10,000th cycle. Inset shows the comprarsion of overpotentials for $Cr_{0.6}Ru_{0.4}O_2$ (550) and $RuO_2$ at the current density of 10 mA cm$^{-2}$ at the first and 10,000th cycle. For $RuO_2$ after 10,000 cycles, the overpotential is corresponded to 3.5 mA cm$^{-2}$ which is the maxium current density of its LSV curve; **d** Tafel plots of $Cr_{0.6}Ru_{0.4}O_2$ (550) and $RuO_2$ at first and 10,000th cycle; **e** Chronopotentiometry performance under constant current density of 10 mA cm$^{-2}$ up to 10 h. **f** TOF results of $Cr_{0.6}Ru_{0.4}O_2$ (550) and $RuO_2$ at the first and 10,000th cycle

Electrochemical impedance spectroscopy (EIS) measurement was further employed to reveal the catalytic property during OER. As shown in Fig. 4b, all the EIS spectra (Nyquist plots) at 1.395 V display a depressed semicircle, suggesting a charge-transfer process during the OER. These Nyquist plots can be well fitted by a simple equivalent electrical circuit which is composed of three components: solution resistance ($R_{sol}$), charge transfer resistance ($R_{ct}$), and double layer capacitance ($C_{dl}$)[35–37]. The charge transfer resistance of $Cr_{0.6}Ru_{0.4}O_2$ electrocatalyst generally increases with the applied annealing temperature. The larger charge transfer resistance for $Cr_{0.6}Ru_{0.4}O_2$ electrocatalyst at higher annealing temperature can also be attributed to the lattice strain effects[34]. For $Cr_{0.6}Ru_{0.4}O_2$ (550), the charge transfer resistance is 97.2 Ω, which is much smaller than that of commercial $RuO_2$ (>4000 Ω) tested under the same conditions (Supplementary Figure 12, Supplementary Table 3), demonstrating a much faster kinetics for OER. Here the high Rct of commercial $RuO_2$ is due to the fact that the OER reactions on $RuO_2$ catalyst do not occur at 1.395 V. We thus measured the EIS spectra of $RuO_2$ at higher voltages (Supplementary Figure 13). It shows that the Rct of $RuO_2$ dramatically decreased with the increasing voltage applied. At 1.55 V, the Rct is 45.6 Ω, and the corresponding area-specific Rct is 3.2 Ω cm$^2$, comparable to those of literatures reported[38]. In addition, we also measured the EIS spectra of $Cr_{0.6}Ru_{0.4}O_2$ (550) at higher voltages (Supplementary Figure 14). The results show that $Cr_{0.6}Ru_{0.4}O_2$ (550) has rather small Rct, with a value of 10.6 Ω at 1.47 V, corresponding to the area-specific Rct value of 0.7 Ω cm$^2$. The durability of $Cr_{0.6}Ru_{0.4}O_2$ electrocatalysts were assessed by cycling the catalysts between 1.2 and 1.6 V at a sweep

rate of 100 mV s$^{-1}$ in 0.5 M $H_2SO_4$ for 10,000 cycles. For $Cr_{0.6}Ru_{0.4}O_2$ electrocatalyst annealed at 450 °C, the overpotential (at 10 mA cm$^{-2}$) dramatically decreased from 177 mV at the first cycle to 242 mV at the 10,000th cycle (Supplementary Figure 15) due to the relatively unstable structures under the acidic solutions. In contrast, $Cr_{0.6}Ru_{0.4}O_2$ electrocatalyst annealed above 500 °C exhibited stable OER performance, with slight overpotential decrease (<20 mV at 10 mA cm$^{-2}$) after 10,000 cycles (Supplementary Figure 15). Notably, $Cr_{0.6}Ru_{0.4}O_2$ (550) showed only 11 mV overpotential decrease (at 10 mA cm$^{-2}$) after 10,000 cycles (Fig. 4c). The high stability of $Cr_{0.6}Ru_{0.4}O_2$ (550) was also confirmed by TEM images of $Cr_{0.6}Ru_{0.4}O_2$ (550) after 10,000 cycles, where no crystallinity or morphology change was observed (Supplementary Figure 16). In addition, the ICP-MS experiments (Supplementary Table 4) show that less than 2.5% Ru and 8% Cr of $Cr_{0.6}Ru_{0.4}O_2$ (550) were dissolved in the acidic electrolyte solution after 10,000 cycles, which results in the slight degradation of OER performance. Note that such leaching content is smaller than those of recently reported excellent OER catalysts for acidic condition[23,27].

In terms of both activity and stability, $Cr_{0.6}Ru_{0.4}O_2$ (550) represents the best-performance OER electrocatalyst among the $Cr_{0.6}Ru_{0.4}O_2$ electrocatalysts annealed between 450 and 650 °C, with overpotential (at 10 mA cm$^{-2}$) of 178 mV at the first cycle and 189 mV at the 10,000th cycle. For further comparison, the OER performance of commercial $RuO_2$ powder with particle size of ~30 nm (Supplementary Figure 17) was also tested under the same conditions. As shown in Fig. 4c, $RuO_2$ exhibited much lower activity and stability compared to $Cr_{0.6}Ru_{0.4}O_2$ (550). The

overpotential at 1 and 10 mA cm$^{-2}$ of $RuO_2$ were measured to be 240 mV and 297 mV, respectively, which are consistent with those reported in literatures (Supplementary Table 5)[38]. After 10,000 cycles, the OER activity was dramatically decreased and became even negligible compared to the initial cycle. Figure 4d shows the Tafel plots of $Cr_{0.6}Ru_{0.4}O_2$ (550) and $RuO_2$ at the first and 10,000th cycle. The Tafel slope for $RuO_2$ dramatically rose from 64 mV dec$^{-1}$ to 88 mV dec$^{-1}$ after 10,000 cycles. In contrast, the Tafel slope for $Cr_{0.6}Ru_{0.4}O_2$ (550) slightly decreased from the initial value of 58 mV dec$^{-1}$ to 56 mV dec$^{-1}$ after 10,000 cycles. To further confirm the difference on stability of $Cr_{0.6}Ru_{0.4}O_2$ (550) and $RuO_2$ in catalytic performance, chronopotentiometry was examined under a constant current density. According to the suggested benchmark criteria in previous reports[23,33], a current density of 10 mA cm$^{-2}$ was used in the present study. Figure 4e shows the corresponding potential change for both $Cr_{0.6}Ru_{0.4}O_2$ (550) and $RuO_2$. The potential for $RuO_2$ electrocatalyst changed from 1.5 to 1.9 V in 40 min and rose sharply to 2.19 V, essentially losing all its activity. On the contrary, the $Cr_{0.6}Ru_{0.4}O_2$ (550) electrocatalyst remained essentially stable throughout the 10 h chronopotentiometry test. Furthermore, the turnover frequency (TOF) of $Cr_{0.6}Ru_{0.4}O_2$ (550) and $RuO_2$ was also calculated by dividing the number of oxygen molecules generated by the number of Ru sites under an assumed 100% Faradaic efficiency (Fig. 4f)[39]. $Cr_{0.6}Ru_{0.4}O_2$ (550) showed a TOF value of 0.15 s$^{-1}$ at the overpotential of 260 mV for the first cycle and slightly decreased to 0.13 s$^{-1}$ for the 10,000th cycle. However, under the same condition, the TOF of $RuO_2$ was decreased by an order of magnitude, changing from $4.9 \times 10^{-3}$ s$^{-1}$ at the first cycle to $5.5 \times 10^{-4}$ s$^{-1}$ for the 10,000th cycle. The same TOF change trend was also observed at an overpotential of 230 mV. It should be noted that all the Ru atoms including inaccessible ones in bulk were treated as surface sites in this TOF calculation, which thus underestimated the true TOF values[40]. In addition, we further calculated the electrochemically active surface area (ECSA), roughness factor of $Cr_{0.6}Ru_{0.4}O_2$ (550) and $RuO_2$ electrode, and plotted the LSVs with respect to the ECSA (Supplementary Figure 18-20, Supplementary Table 6). The results show that the enhanced activity of OER performance of $CrO_2$-$RuO_2$ solid solution is not just enhanced by the ECSA, and the intrinsic activity arising from the Cr ions plays an more important role.

In short, $Cr_{0.6}Ru_{0.4}O_2$ (550) exhibits superior OER performance compared to $RuO_2$ catalysts or other $RuO_2$-based catalysts reported to date. Notably, it even outperforms the $IrO_2$-based catalysts, which represent the state-of-the-art electrocatalyst for OER in acidic media (Table 1). An exhaustive comparison with other reported OER catalysts in acidic media is shown in Supplementary Table 7. It shows that the mass activity of $Cr_{0.6}Ru_{0.4}O_2$ (550) at 270 mV (229 A g$^{-1}$) is also much higher than those reported in literatures. In addition, the OER performance of $CrO_2$ powder was also measured as a reference. As expected, no OER activity was observed on $CrO_2$ powder (Supplementary Figure 21), suggesting that the synergic effects of Ru(IV) and Cr (IV) components in $Cr_{0.6}Ru_{0.4}O_2$ structure are responsible for the excellent OER performance.

We further synthesized a series of Cr–Ru oxides with different Cr ratios to investigate the Cr role on the catalytic property. By varying the mass of $RuCl_3$ in THF solution, we prepared MIL-101-$RuCl_3$ precursors with different $RuCl_3$ loading, and then obtained Cr–Ru oxides with Cr/Ru ratios varying from 9:1 to 6:4 after annealing (Cr/Ru ratios were determined by ICP-MS measurements). Figure 5 shows the corresponding morphology and structure evolution of Cr–Ru oxides. Directly annealing MIL-101(Cr) without loading $RuCl_3$ at 450 °C, we only obtained $Cr_2O_3$ nanoparticles with high crystallinity (Supplementary Figure 22).

After loading $RuCl_3$ into MIL-101 (Cr), $CrO_2$-$RuO_2$ solid solution phase started to emerge after annealing. This is because $RuO_2$ and $CrO_2$ share the same rutile structure and have similar lattice constants, and the presence of Ru would induce the formation of $RuO_2$–$CrO_2$ solid solution. For $Cr_{0.91}Ru_{0.09}O_{2-\delta}$ and $Cr_{0.83}Ru_{0.17}O_{2-\delta}$ ($\delta$ was used to balance the valance of $Cr^{3+}$ for the powders with mixed phase of $Cr_2O_3$ and $CrO_2$–$RuO_2$ solid solution) with low Ru content, the major phase is still $Cr_2O_3$, which can be clearly observed from the PXRD patterns in Fig. 5e. In contrast, for $Cr_{0.72}Ru_{0.28}O_{2-\delta}$ with higher Ru content, the $CrO_2$–$RuO_2$ solid solution turn to be the major phase, and for $Cr_{0.67}Ru_{0.33}O_2$ and $Cr_{0.6}Ru_{0.4}O_2$, pure phase of $CrO_2$–$RuO_2$ solid solution was formed. Note that, all the peaks shift slightly to the left side as the Ru content increases, which is a characteristic of $RuO_2$–$CrO_2$ solid solution. We further calculated the lattice parameters of $Cr_{1-x}Ru_xO_2$ with solid solution as the major or pure phase (i.e., $Cr_{0.72}Ru_{0.28}O_{2-\delta}$, $Cr_{0.67}Ru_{0.33}O_2$, and $Cr_{0.6}Ru_{0.4}O_2$). As shown in Supplementary Figure 23, the c parameter varies nearly linearly with the composition. This quasi-linear relationship is in good agreement with the Vegard's law. Although the shift of the $a$ parameter shows the same trend as the c parameter when the Ru content increases, there is a deviation for the $a$ parameter according to the Vegard's law. This deviation was possibly due to the little difference of $a$ parameter between $RuO_2$ ($a = 4.499$ Å) and $CrO_2$ ($a = 4.421$ Å), and/or the existence of some defects in the lattice along the $a$ axis[41,42]. It is noteworthy that there is a pre-oxidation peak of the solid solution samples, which can be ascribed to the pre-oxidation of Cr. However, no such pre-oxidation peak was observed on $Cr_2O_3$ sample annealed at 450 °C, which can be attributed to its large crystal size (Supplementary Figure 24a) and relatively low active surface area that could cause low conductivity and activity. We thus prepared $Cr_2O_3$ with much smaller particle sizes (Supplementary Figures 22 and 24b) by annealing MIL-101(Cr) at lower temperature (300 °C). Indeed, herein we also observed this pre-oxidation peak on the $Cr_2O_3$ with less crystallinity (Supplementary Figure 25), albeit the peak was weak. The position of pre-oxidation peak of $Cr_2O_3$ was slightly higher than that of $Cr_{1-x}Ru_xO_2$, which was possibly due to the synergistic effect of Cr and Ru in $Cr_{1-x}Ru_xO_2$. Due to the saturation adsorption limit, we are unable to prepare Cr–Ru oxides with Cr/Ru ratio lower than 0.6/0.4. LSV results show that the OER performance of Cr–Ru oxides is highly correlated to the Ru/Cr ratio. $Cr_{0.91}Ru_{0.09}O_{2-\delta}$ and $Cr_{0.83}Ru_{0.17}O_{2-\delta}$ show moderate performance due to the high content of inactive $Cr_2O_3$ phase. With increasing Ru composition, the OER activity can be dramatically improved because the $CrO_2$–$RuO_2$ solid solution evolved as the major phase or even pure phase (Fig. 5f). However, it is noteworthy that the OER performance of $Cr_{0.91}Ru_{0.09}O_2$ with small amount of $CrO_2$-$RuO_2$ solid solution phase is still higher than that of $RuO_2$, highlighting the crucial role of Cr ions on the improved activity towards OER. We also measured the OER performance of mixed $RuO_2$ and $CrO_2$ sample. The result shows that mixed $RuO_2$ and $CrO_2$ has very poor OER performance, even much lower than that of pristine $RuO_2$. Note that the conductivity plays an important role in the OER process, and it might not be a good comparison to the chromium-ruthenium oxides if some residual carbon species inherent from MOF precursor exist in our samples. Therefore, we further preformed Raman and thermogravimetric (TG) measurement of the samples to detect the residual carbon. As shown in Supplementary Figures 26 and 27, no signal of the residual carbon can be observed. Nevertheless, we added carbonaceous additive (commercial acetylene black that has high conductivity) to the mixed $CrO_2$–$RuO_2$, denoted as mixed $CrO_2$–$RuO_2$/C. As shown in Supplementary Figure 28, the OER activity of mixed $CrO_2$–$RuO_2$/C was enhanced after the addition of carbon black,

**Table 1 Selected catalysts with high OER performance**

| Catalysts | Substrate | Electrolyte | Overpotential at specific current density | Chronopotentiometry at specific current density | Ref. |
|---|---|---|---|---|---|
| $RuO_2$ | Ti | 0.5 M $H_2SO_4$ | 240 mV@1 mA cm$^{-2}$ | – | 38 |
| $IrO_2$ | GCE | 0.1 M $H_4ClO_4$ | ~430 mV @10 mA cm$^{-2}$ | – | 14 |
| $BaYIrO_6$ | Au | 0.1 M $H_4ClO_4$ | ~315 mV @10 mA cm$^{-2}$ | 1 h@10 mA cm$^{-2}$ | 23 |
| $IrO_x/SrIrO_3$ | Cu wire | 0.5 M $H_2SO_4$ | 270-290 mV @10 mA cm$^{-2}$ | 30 h@10 mA cm$^{-2}$ | 25 |
| $Y_2Ru_2O_{7-\delta\bar{y}}$ | GCE | 0.1 M $H_4ClO_4$ | 270 mV @1 mA cm$^{-2}$ | 8 h@ 1 mA cm$^{-2}$ | 27 |
| IrCoNi PHNCs | CFP | 0.1 M $HClO_4$ | 303@10 mA cm$^{-2}$ | 3.3 h@5 mA cm$^{-2}$ | 67 |
| $W_{0.57}Ir_{0.43}O_{3-\delta}$ | FTO | 1 M $H_2SO_4$ | 370@10 mA cm$^{-2}$ | 0.6 h@10 mA cm$^{-2}$ | 68 |
| IrNi NCs | CFP | 0.1 M $HClO_4$ | 280@10 mA cm$^{-2}$ | 2 h@5 mA cm$^{-2}$ | 69 |
| Ir | GF | 0.5 M $H_2SO_4$ | 290@10 mA cm$^{-2}$ | 10 h@10 mA cm$^{-2}$ | 70 |
| $Cr_{0.6}Ru_{0.4}O_2$ (550) | GCE | 0.5 M $H_2SO_4$ | 178 mV @10 mA cm$^{-2}$ | 10 h@ 10 mA cm$^{-2}$ | This work |

but still lower than that of pure $RuO_2$, indicating the important synergistic effect of $Cr^{4+}$ role as a participating lattice ion.

**Intrinsic mechanism for the excellent OER performance**. We first performed X-ray photoelectron spectroscopy (XPS) to access the surface chemical state of $Cr_{0.6}Ru_{0.4}O_2$ (550). As shown in Fig. 6a, there are two sets of doublet peaks for Ru 3d in the region between 280 and 290 eV, corresponding to the doublet peaks for Ru (IV) $3d_{5/2}$, $3d_{3/2}$ and their satellite peaks[43]. The primary Ru $3d_{5/2}$ and $3d_{3/2}$ peaks of $RuO_2$ centered at 280.6 and 284.8 eV, respectively, which are consistent with literatures[44,45]. A shift to higher binding energy can be clearly observed on $Cr_{0.6}Ru_{0.4}O_2$ (550) compared to $RuO_2$, suggesting a lower electron density at the Ru sites. This can be attributed to the electron withdrawing effect of Cr (IV) in the lattice. Note that the observed C1s peaks are resulting from the carbon adhesive tape used in XPS measurement and environmental corrosion carbon. Indeed, as mentioned above, Raman characterization and thermogravimetric analysis confirmed that there is negligible carbon component in the $Cr_{0.6}Ru_{0.4}O_2$ catalysts (Supplementary Figures 26 and 27). For Cr 2p, three sets of doublet peaks can be observed on $Cr_{0.6}Ru_{0.4}O_2$ (550) and $CrO_2$ in the region between 570 and 595 eV (Fig. 6b). The primary peaks at ~576.0 eV correspond to Cr (IV) $2p_{3/2}$[46,47]. For Cr (IV) $2p_{3/2}$ of $Cr_{0.6}Ru_{0.4}O_2$ (550), a shift to lower binding energy is observed compared to $CrO_2$, implying a higher electron density at Cr sites, which confirms the withdrawing effect of Cr (IV) in $Cr_{0.6}Ru_{0.4}O_2$ (550). For the other two peaks in the Cr $2p_{3/2}$ region, the smaller ones at ~575.0 eV can be assigned to Cr (III) $2p_{3/2}$, which implies the appearance of a small amount of Cr (III) sites on the outer surface of $Cr_{0.6}Ru_{0.4}O_2$ (550) and $CrO_2$ crystals[47,48], and the larger ones at ~577.9 eV can be assigned to $CrO_2H$, which likely resulted from the reaction between Cr (IV) and the proton from environment[49]. Additional XPS spectra for wide scan and Ru 3p regions are shown in Supplementary Figures 29 and 30.

To elucidate the atomic structure of $Cr_{0.6}Ru_{0.4}O_2$(550), X-ray absorption spectroscopy (XAS) characterization was further employed. Figure 7a shows the X-ray absorption near edge structure (XANES) of Ru K-edge region of the rutile-type $Cr_{0.6}Ru_{0.4}O_2$(550). Pure Ru metal foil and $RuO_2$ powder were also measured as reference. The shoulder near the adsorption threshold of Ru foil is corresponding to the 1s to 4d transition. For $RuO_2$ and $Cr_{0.6}Ru_{0.4}O_2$(550), this shoulder is weaker, because the increased lattice symmetry prevents the mixing of 4d and 5p orbitals. The observed transition energy of XANES (corresponding to the 1s to 5p transition) for $RuO_2$ and $Cr_{0.6}Ru_{0.4}O_2$ is higher than that for Ru. This can be attributed to the formation of Ru–O bonds, which pushes up the empty

state of 5p oribials of Ru atoms[50]. We further analyzed the absorption energy ($E_0$, determined from the first maximum in the first-order derivative), which is proportional to the oxidation state of transition metals[51,52]. We found that the absorption energy for $Cr_{0.6}Ru_{0.4}O_2$(550) ($E_0 = 22129.9$ eV) was similar with the value of $RuO_2$ ($E_0 = 22129.5$ eV), implying that the oxidation state of Ru in $Cr_{0.6}Ru_{0.4}O_2$(550) is close to +4. The slightly higher absorption energy can be attributed to the electron withdrawing effect of the neighboring lattice $Cr^{4+}$ ion, consistent with the XPS analysis results. Ru K-edge extended X-ray absorption fine structure (EXAFS) analysis was used to reveal the initial information on the Ru−O and Ru−Ru bonds. The corresponding Fourier transformed (FT) radial structure based on the $k^2$-weighted EXAFS is displayed in Fig. 7c. The peak at 1.59 Å for $RuO_2$ is associated with the back scattering of Ru−O in the first shell[27]. In contrast, the Ru−O bond length in $Cr_{0.6}Ru_{0.4}O_2$(550) is slightly shortened to 1.55 Å, in line with the slightly higher absorption energy for $Cr_{0.6}Ru_{0.4}O_2$(550). The peaks at 2.73 and 3.21 Å for $RuO_2$ arise from the back scatterings of Ru−Ru in the second and third shell[53]. These peaks for $Cr_{0.6}Ru_{0.4}O_2$(550) are assigned to the back scatterings Ru–Ru and Ru–Cr. The decreased intensity (i.e., vibrational amplitude) should be ascribed to the extremely small particle size (less than 15 nm)[54,55]. Furthermore, these peaks are also shifted to the left. Clearly, the presence of Cr can profoundly alter the local electronic structures of Ru and the associated Ru-O bonding, which directly determine the OER activity. Accordingly, Cr K-edge XANES and EXAFS were also used to examine the Cr oxide state, and Cr–O bond in $Cr_{0.6}Ru_{0.4}O_2$(550) (Fig. 7b, d). The absorption energy of $Cr_{0.6}Ru_{0.4}O_2$(550) ($E_0 = 6006.7$ eV) is higher than that of Cr metal ($E_0 = 5989.0$ eV), but close to the value of $CrO_2$ ($E_0 = 6006.8$ eV). In addition, the peak in the region of pre-edge absorption is also a characteristic of the formation of $Cr^{4+}$, corresponding to the 1s to 3d transition[56]. The slightly lower absorption energy can be attributed to the electron withdrawing effect of $Cr^{4+}$ ion, in agreement with the XPS results and Ru oxidation analysis. In addition, as shown in Fig. 7d, the Cr–O length for $CrO_2$ is 1.47 Å. In $Cr_{0.6}Ru_{0.4}O_2$(550), the Cr–O is slightly elongated to 1.50 Å, in accordance with the EXAFS result of Ru K-edge. For comparison, we also measured the Cr K-edge XANES of $Cr_{0.6}Ru_{0.4}O_2$ (450). As shown in Supplementary Figure 31, the intensity of the pre-edge peak of $Cr_{0.6}Ru_{0.4}O_2$(450) is above that of $Cr_{0.6}Ru_{0.4}O_2$(550), indicating a lower symmetry environment of the Cr atoms in $Cr_{0.6}Ru_{0.4}O_2$(450)[56]. It confirmed that the fine structure of $Cr_{0.6}Ru_{0.4}O_2$(450) is different from that of $Cr_{0.6}Ru_{0.4}O_2$(550), i.e., the pure phase rutile Cr–Ru oxide has not been well formed.

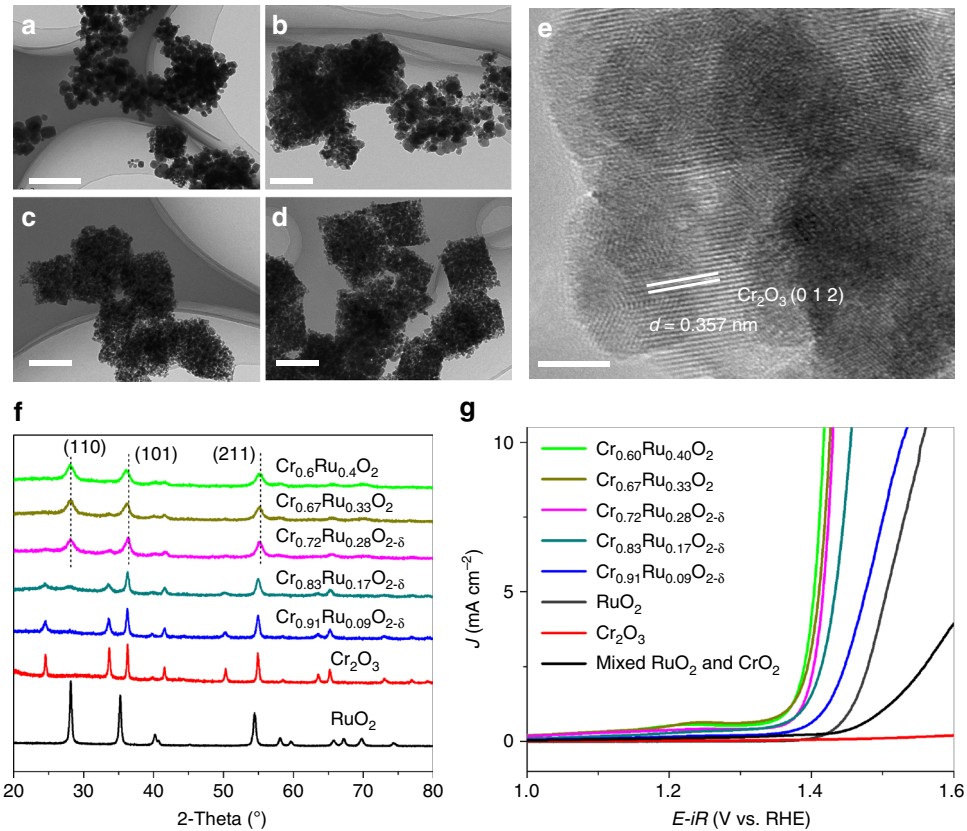

**Fig. 5** Evolution of chromium–ruthenium oxides with different Cr/Ru ratios: **a–d** TEM images of $Cr_{0.91}Ru_{0.09}O_{2-\delta}$, $Cr_{0.83}Ru_{0.17}O_{2-\delta}$, $Cr_{0.72}Ru_{0.28}O_{2-\delta}$ and $Cr_{0.67}Ru_{0.33}O_2$, respectively (scale bars, 200 nm); **e** HR-TEM image of $Cr_{0.83}Ru_{0.17}O_{2-\delta}$ (scale bar, 5 nm); **f** PXRD patterns, $Cr_2O_3$ powder was obtained from directly annealing pure MIL-101 (Cr); **g** LSV results

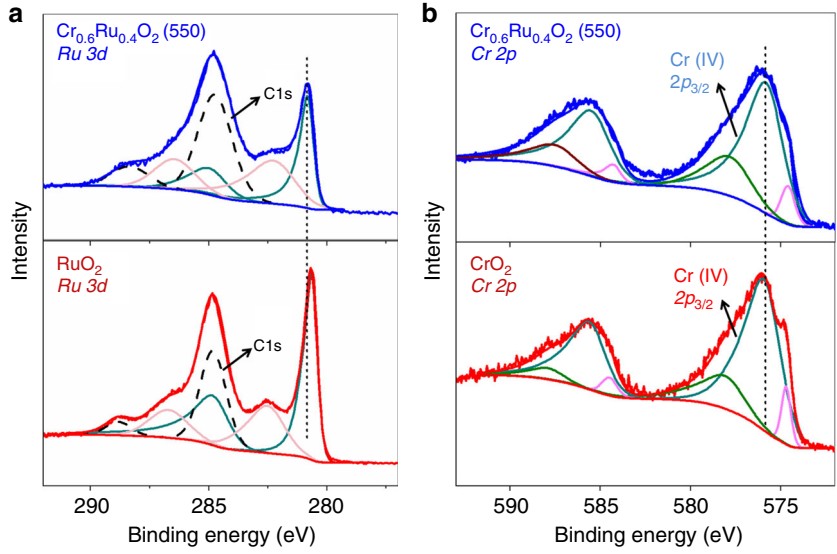

**Fig. 6** XPS of $Cr_{0.6}Ru_{0.4}O_2$ (550) for Ru 3d and Cr 2p. **a** XPS of $Cr_{0.6}Ru_{0.4}O_2$ (550) and $RuO_2$ for Ru 3d regions. **b** XPS of $Cr_{0.6}Ru_{0.4}O_2$ (550) and $CrO_2$ for Cr 2p regions. The blue and red smoothing lines are fitting results of the sum of individual components. For Ru 3d, color codes are used to distinguish the different spin-orbit components, dark cyan for primary Ru $3d_{3/2}$ and 3d $_{5/2}$ spin states, and light magenta for satellite Ru $3d_{3/2}$ and 3d $_{5/2}$ spin states

Finally, we carried out DFT calculations in order to understand the promoted OER performance of $CrO_2–RuO_2$ electrocatalyst. Here we constructed a simulation model of $Cr_5Ru_3O_{16}$, which has a composition close to the experimentally measured value. We assumed that Cr and Ru are distributed as evenly as possible in the rutile-like crystal. The simulated PXRD pattern of the relaxed structure was in good agreement with experiments. Based on the Bader charge analysis, the partial charge of Ru in bulk $RuO_2$ was

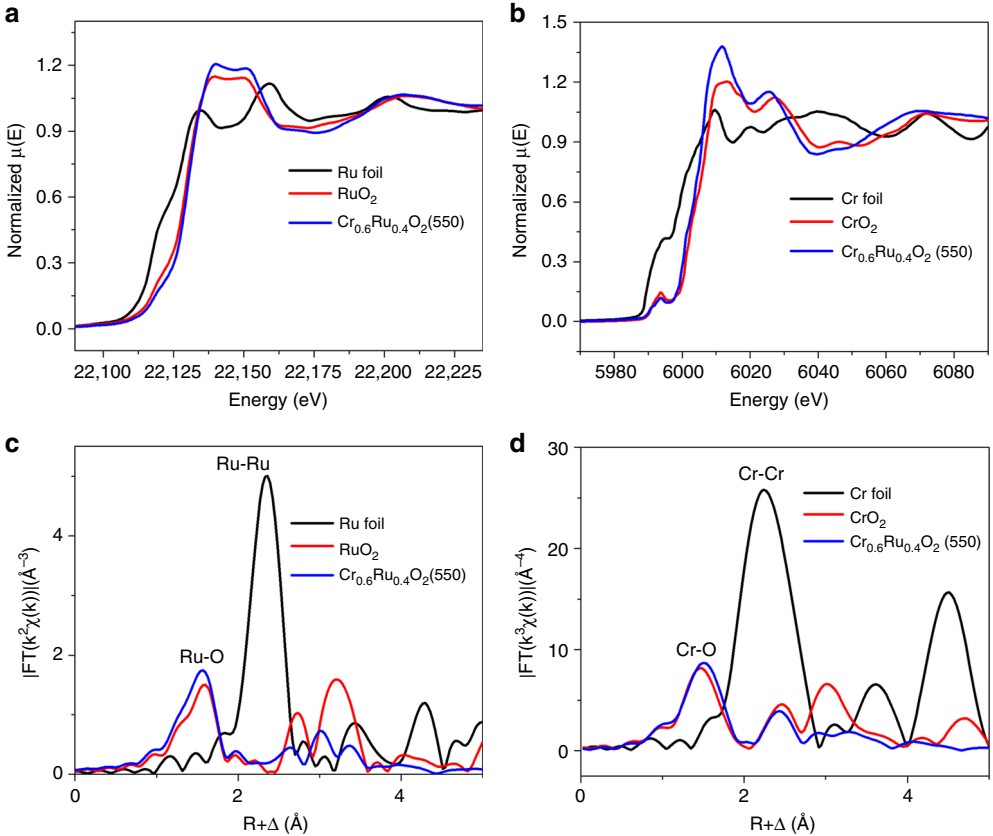

**Fig. 7** XAS analysis of $Cr_{0.6}Ru_{0.4}O_2$ (550) electrocatalyst. **a** Normalized Ru K-edge XANES spectra of $Cr_{0.6}Ru_{0.4}O_2$ (550), Ru foil and commercial $RuO_2$. **b** Normalized Cr K-edge XANES spectra of $Cr_{0.6}Ru_{0.4}O_2$ (550), Cr foil, and commercial $CrO_2$. **c** Fourier transformed EXAFS spectra of Ru edge for $Cr_{0.6}Ru_{0.4}O_2$ (550), Ru foil and commercial $RuO_2$. **d** Fourier transformed EXAFS spectra of Cr edge for $Cr_{0.6}Ru_{0.4}O_2$ (550), Cr foil, and commercial $CrO_2$

calculated to be $+1.73|e|$. For comparison, the Ru cation in $Cr_5Ru_3O_{16}$ possesses a higher positive charge of $+1.92|e|$. Accordingly, the partial charge on the neighboring Cr cation decreases from $+1.89|e|$ in $CrO_2$ to $+1.84|e|$ in $Cr_5Ru_3O_{16}$. Clearly, the electron transfer from Ru to Cr is consistent with the XANES results. More positively charged Ru cations lead to smaller cation radius, and the corresponding peaks in EXAFS slightly shift to the left. Moreover, the highly oxidized state of Ru implies the improved ability for the oxidation of water to oxygen, namely OER. We further plotted the density of states (DOS) to discern the nature of the electronic structures. As shown in Fig. 8a, the incorporation of Cr apparently altered the DOS of nonmagnetic $RuO_2$. The occupation at the Fermi level decreases from 2.01 states/(eV*cell*spin) in $RuO_2$ to 1.07 states/(eV*cell*spin) in the solid solution, indicating the stabilization of crystalline structure[57]. Owing to the localized nature of O $p$-band, its band center is known as an effective descriptor to predict intrinsic OER activity of oxides[58]. Considering the less electron number of $Cr^{4+}$ ($2e^-$) than that of $Ru^{4+}$ ($4e^-$), the Fermi level of solid solution is shifted downward due to the band filling effect. Correspondingly, the O $p$-band center moves closer to the Fermi level in $Cr_5Ru_3O_{16}$ ($-2.48$ eV), compared with the value of $-2.91$ eV in $RuO_2$. Clearly, the upshift O $p$-band suggests the enhanced activity for OER. The detailed projected DOS of $Cr_5Ru_3O_{16}$ are displayed in Supplementary Figures 32–33, also consistent with previous theoretical study on $CrO_2$–$RuO_2$ structures[59]. The oxygen 2$p$ (O-$p$) states below ~1.5 eV overlap with part of the metal $d$-bands. Metal $t_{2g}$ orbitals show a unique spread and strong peak at the edge of valence band, especially in Ru $d$-orbitals. Interestingly, the O-$p$ orbital (the major

component is below $-2$ eV) and Ru-$d$ bands at higher energy state are well separated in $RuO_2$ (Fig. 8a). In contrast, the relatively low energy Cr $t_{2g}$ orbitals can enhance the hybridization of O-$p$ orbital, thus further push O $p$-center closer to Fermi level in the solid solution. Note that the empty $e_g$ orbitals of Cr intensely strengthen the DOS ranging from about 2 to 6 eV and the increased DOS related to σ antibonding state suggests a weak Cr–O binding strength.

On the other hand, we also calculated the free energy profiles of OER to directly compare the OER activities of $RuO_2$ and $CrO_2$–$RuO_2$ solid solution. A slab model containing 32 O and 16 metal atoms was employed, in which the Cr/Ru ratio was kept as 5:3. Here we considered a four-step OER mechanism with CHE model to provide a general view[60–62]. We first focused on the (110) facet as the surface model, because it has been identified as the most stable from calculation of surface energy (Supplementary Table 8). We constructed surfaces of both $RuO_2$ and solid solution for comparison. Five different configurations of solid solution surface were further modeled to average the calculated energy barriers. As shown in Fig. 8b, the five-coordinated surface Ru was identified as the reactive adsorption site. Interestingly, under the oxidation condition in water, the Ru site could readily adsorb OH to form *OH. For all the models, the formation of *OOH was found to be the rate determining step (RDS). On the $Cr_5Ru_3O_{16}$ surface, the free energy change of RDS at the Ru site was calculated to be 1.87 eV, which is approximately 0.1 eV lower than that on $RuO_2$ surface (2.02 eV) (Fig. 8c). It is consistent with the decreased overpotential of 100 mV measured in experiments. To further corroborate the synergistic effect of Cr ions on the enhanced OER activity, we considered two more cases with

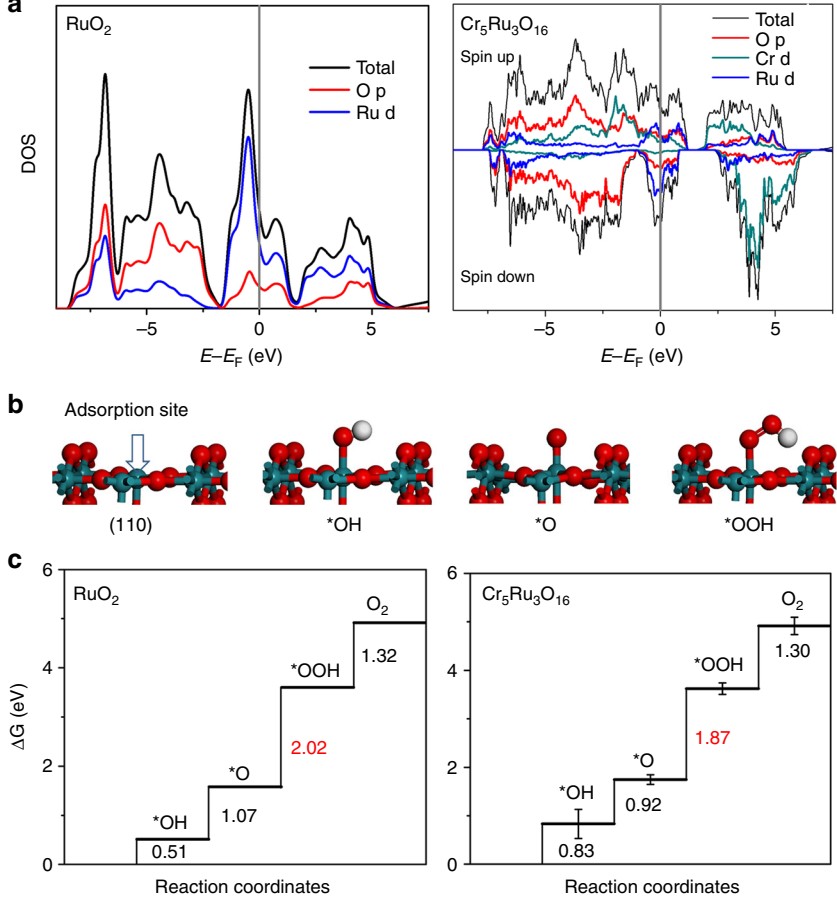

**Fig. 8** DFT calculations on the studied $CrO_2$-$RuO_2$ solid solution system. **a** DOS of $RuO_2$ and $Cr_5Ru_3O_{16}$; **b** The four-step OER process; **c** The calculated free energy diagrams for $RuO_2$ and $Cr_5Ru_3O_{16}$

different Ru concentration and structures, as shown in Supplementary Figure 34. In the first one, there are only surface Ru and sub-surface Cr ions, in which the energy barrier for RDS was calculated to be 1.81 eV. In the second case, one Cr cation on $CrO_2$(110) surface was replaced by Ru, yielding a reduced RDS barrier of 1.75 eV, which is lower than that of original RuO2 surface. On the other hand, we also investigated two relatively less stable surfaces, namely (200) and (101) facets, which were observed in HR-TEM image (Fig. 3d). In a previous experimental study[63], these two surfaces of $RuO_2$ were found to be more active. Our DFT results (Supplementary Figure 35) are consistent with this work, and the corresponding energy barriers of RDS on (200) and (101) surfaces were calculated to be 1.79 and 1.96 eV, respectively. In $CrO_2$–$RuO_2$ system, these two barriers were further decreased by 0.04 and 0.02 eV, respectively.

## Discussion

In summary, by using Cr-based MOF, we have developed a cost effective rutile $Cr_{0.6}Ru_{0.4}O_2$ electrocatalyst with superior OER activity and stability in acidic media. Our experimental results and DFT calculations revealed the profound influence of Cr on the OER performance. The enhanced stability is related to the lower occupation at the Fermi level, while the higher activity results from the altered electronic structures. The calculated free energy diagrams for OER further demonstrates a lower energy barrier for the formation of *OOH, which is the RDS. On the other hand, Ru plays the key role to induce the formation of ruile-structured $CrO_2$ and thus $CrO_2$–$RuO_2$ solid solutions because $RuO_2$ and $CrO_2$

share the same rutile structure and have similar lattice constants. Note that direct annealing of the MIL-101(Cr) precursor only led to an inactive product of $Cr_2O_3$. These findings and results open a route to design highly active, stable and relatively low-cost electrocatalysts for OER in acidic media. To shed light on the further optimization, we have investigated and screened a series of possible rutile-like $MO_2$–$RuO_2$ systems, in which M is a tetravalent cation. In light of the altered electronic structures in $CrO_2$–$RuO_2$, we propose that the electron withdrawing on Ru ions can facilitate water oxidation and oxygen evolution. The calculated parameters of the cells and partial charges on Ru are summarized in Supplementary Table 9. It is noteworthy that the partial charge on Ru ions in $CrO_2$–$RuO_2$ is found to be the most positive. Interestingly, we found that $MnO_2$ can also form solid solution with $RuO_2$ and possess good OER performance, owing to the similar cell parameter with that of $RuO_2$ and the highly positive partial charge on Ru (1.88 e). In contrast, we found that the reductive tetravalent cations, such as Nb and W, lead to electron accumulation on Ru. As a result, their corresponding solid solutions are expected to have lower OER activity. In addition to the formation of solid solution, we also anticipate that doping $RuO_2$ with highly oxidizing metal ions, such as $Ce^{4+}$, is another viable strategy to improve the OER activity. On the other hand, our preparation method for chromium ruthenium oxides solid solution electrocatalyst can be extended to prepare other rutile-structured electrocatalysts, such as the potentially active manganese ruthenium oxide and vanadium ruthenium oxide, or even non-Pt-group metal materials, such as chromium manganese oxide.

## Methods

**Materials**. All chemicals were obtained from commercial suppliers at analytical grade and used as received without further purification. The commercial $RuO_2$ and $CrO_2$ were purchased from Sigma-Aldrich.

**MIL-101(Cr) synthesis**. MIL-101(Cr) was prepared by a hydrothermal reaction following the procedure reported in our previously work[30]. The prepared MIL-101 (Cr) was activated at 150 °C under vacuum for 12 h for future use.

**Preparation of $RuCl_3$-MIL-101 (Cr)**. A series of $RuCl_3$-MIL-101 (Cr) with different $RuCl_3$ loading were prepared by mixing the desired amount of MIL-101 (Cr) and $RuCl_3$ in THF solution. Typically, 0.2 g $RuCl_3$ was dissolved in 30 ml tetrahydrofuran (THF) under stirring for 5 min. After that, 0.2 g MIL-101 (Cr) was slowly added into the $RuCl_3$ solution under stirring. To ensure the uniform distribution of $RuCl_3$ within the pores and thus the formation of homogeneous Cr–Ru oxide solid solution phase upon heating, the resulting mixture was further kept stirring at room temperature for 18 h to allow the compete loading of $RuCl_3$ into MIL-101 (Cr) pores. After impregnation, the product was recovered by centrifugation, and washed four times with THF to remove $RuCl_3$ remained on the outer surface of MIL-101 (Cr) particles. Finally, the resulting $RuCl_3$-MIL-101 (Cr) was dried at 80 °C in air for 6 h. For the other $RuCl_3$-MIL-101 (Cr) with lower $RuCl_3$ loading, the amount of $RuCl_3$ used was decreased to 0.15 g, 0.1 g, 0.05 g, and 0.025 g, respectively.

**Preparation of $Cr_{1-x}Ru_xO_2$**. Fifty microgram of $RuCl_3$-MIL-101 (Cr) powder was placed in muffle furnace and heated to $T$ ($T$ = 450, 500, 550, 600, 650 °C) at a heating rate of 5 °C/min and held for 4 h. After cooling down to room temperatures, the resulting black products were collected, and denoted as $Cr_{1-x}Ru_xO_2$ (T).

**Electrochemical measurements**. In a typical procedure, 4 mg of $Cr_{1-x}Ru_xO_2$ was added to 1 ml of water/ethanol (3:1, v/v) containing 15 μl Nafion aqueous solution (5%, Sigma-Aldrich), and dispersed by sonication for 30 min to generate a homogeneous black ink. Five microliter of the catalyst ink was drop-cast on a glassy carbon electrode (surface area: $0.07065\ cm^2$) and dried in air at room temperature to form a thin film working electrode. For the mixed $CrO_2$-$RuO_2$/C ink preparation, 4 mg carbon black (commercial acetylene black that has high conductivity) was added. A three-electrode cell was employed to measure the OER electrochemical performance. The cell contained the glassy carbon working electrode, a counter electrode made of platinum wire (diameter: 0.5 nm), and a saturated $Hg/Hg_2SO_4$ reference electrode. All measurements were performed in 0.5 M $H_2SO_4$ acidic solution after purging with $O_2$ (99.999%) for at least 30 min. The $Hg/Hg_2SO_4$ reference electrode was calibrated with a Pt wire electrode in $H_2$-saturated 0.5 M $H_2SO_4$ solution. The potential difference between the $Hg/Hg_2SO_4$ reference electrode and reversible hydrogen electrode is 0.645 V. Cyclic voltammograms (CVs) tests were collected at a scan rate of 100 mV/s typically between 1.2 and 1.6 V. Linear sweep voltammetry (LSV) curves were recorded at a scan rate of 5 mV/s typically between 0.8 and 1.6 V. Chronopotentiometric measurements were conducted by applying constant current (10 mA $cm^{-2}$) for up to 10 h. Electrochemical impedance spectroscopy (EIS) were performed at 0.75 V. The EIS results were presented in the form of Nyquist plot and fitted using ZView software with a representative equivalent electrical circuit.

**Electrochemically active surface area (ECSAs)**. The ECSAs were estimated from the electrochemical double-layer capacitance of the catalytic surface. The double layer capacitance ($C_{DL}$) was determined by measuring the non-Faradaic capacitive current charging from the scan-rate dependence of CVs. The potential window of CVs was 1.21–1.31 V vs. RHE (0.1 V potential window centered at the open-circuit potential of the system). The $C_{DL}$ was given by the following equation:

$$i_c = \nu C_{DL}, \tag{1}$$

where $\nu$ is the scan rate. The slope of the plot of $i_c$ as a function of $\nu$ is equal to $C_{DL}$.

The ECSA is calculated from the double layer capacitance according to:

$$ECSA = C_{DL}/C_s, \tag{2}$$

where $C_s$ is the specific capacitance of the sample. We use general specific capacitances of $C_s = 0.035$ mF $cm^{-2}$ based on typical reported values. The roughness factor (RF) is then calculated by dividing ECSA by $0.07065\ cm^2$, the geometric area of the electrode.

**Material characterization**. Power X-ray diffractions (PXRD) patterns of the samples were collected on a D8-Advance Bruker AXS diffractometer with $Cu_{k\alpha}$ ($\lambda$ = 1.5418 Å) radiation at room temperature. In order to obtain high quality data for $Cr_{0.6}Ru_{0.4}O_2$ (550), a very slow scan measurement was performed with a scan interval of 0.005° per step and a scan rate of 3 s per step. Structure analysis was conducted on Jade 2004. The lattice parameters were refined using GSAS software [A. C. Larson and R.B. von Dreele, Los Alamos, 1994]. Inductively coupled plasma-mass spectroscopy (ICP-MS) measurements were carried on NexION 300 (Perkin-

Elmer). For the leaching measurements, the loading amounts of catalysts varied from 20 to 60 μg and the volume of the electrolyte was 100 ml. After 10,000 cycles, the electrolyte was concentrated to a final volume of ~10 ml for ICP-MS analysis. The samples morphologies were examined using a field emission scanning electron microscope (SEM) (Hitachi, S-4800). SEM specimens were prepared by depositing sample powders on carbon adhesive tape on a SEM holder. Transmission electron microscopy (TEM) and high-resolution TEM (HR-TEM) images were recorded on Tecnai F20 microscope, and high-angle annular dark-field scanning transmission electron microscopy (HAADF-STEM) images were carefully recorded on Talos F200X and JEM-ARM200F. Atomic solution HAADF-STEM images were carefully recorded on JEM-ARM200F. For TEM specimen preparation, sample powders were firstly dispersed in ethanol by sonication, followed by dripping onto a carbon-coated copper grid. Nitrogen adsorption/desorption isotherms were measured on ASAP2020M apparatus at 77 K. The Brunauer−Emmett−Teller (BET) surface area was calculated over the range of relative pressures between 0.05 and 0.2. Before the measurements were performed, the samples were outgassed under vacuum at 160 °C for 12 h. X-ray photoelectron spectroscopy (XPS) spectra were recorded on the AXIS ULTRA using $Al_{K\alpha}$ radiation. The X-ray absorption data (XAS) at the Ru-K edge and the Cr K edge of the samples, which were mixed with LiF to reach 50 mg, were recorded at room temperature in transmission mode using ion chambers using the BL14W1 beam line of the Shanghai Synchrotron Radiation Facility (SSRF), China. The station was operated with a Si (111) double crystal mono-chromator. The electron beam energy of the storage ring was 3.5 GeV and the maximum stored current was ~210 mA. The energy calibrations were performed using a Ru foil (22117 eV) or Cr foil (5989 eV). For Ru K-edge XAS, The extracted EXAFS signal, $\chi(k)$, was weighted by $k^2$ in k-range from 3.8 to 15.6 $Å^{-1}$ to obtain the magnitude. For Cr K-edge XAS, The extracted EXAFS signal, $\chi(k)$, was weighted by $k^3$ in k-range from 3 to 12 $Å^{-1}$ to obtain the magnitude.

**Turnover frequency calculation (TOF)**. TOF was calculated based on the method reported in previous works[27,40]. This calculation assumes 100% Faradaic efficiency:

$$TOF = N_{O_2}/N_{Ru} \tag{3}$$

where $N_{O_2}$ is the number of $O_2$ turnovers, calculated using the following formula:

$$N_{O_2} = (j\ mA\ cm^{-2}) \times (A\ cm^2_{oxide}) \times (1\ Cs^{-1}/1000\ mA) \times$$
$$(1\ mol\ e^-/96485\ C) \times (1\ mol\ O_2/4\ mol\ e^-) \times N_A. \tag{4}$$

where $j$ is the measured current density, $A$ is the surface area of electrode, and $N_A$ is Avogadro constant ($6.02 \times 10^{23}\ mol^{-1}$).

The number of Ru sites ($N_{Ru}$) is calculated using the formula: $(0.4 \times (20 \times 10^{-6}$ g) $\times N_A$/ molecular weight of $Cr_{0.6}Ru_{0.4}O_2$) for $Cr_{0.6}Ru_{0.4}O_2$, and $((20 \times 10^{-6}$ g) $\times N_A$/molecular weight $RuO_2$) for $RuO_2$, respectively.

**Density functional theory (DFT) calculations**. The DFT calculations were performed using the Vienna Ab-initio Simulation Package (VASP)[64]. The Perdew-Burke-Ernzerhof (PBE) functional of the generalized gradient approximation (GGA)[65] was employed with projector augmented wave (PAW)[66] method. The valence electronic configurations were O (2s, 2p), Ru (4p, 4d, 5s), Cr (3p, 3d, 4s), and H (1s). In particular, the $U_{eff}$ of 3.7 eV was utilized for 3d orbital of Cr. Spin polarization was also considered. The energy cutoff for plane wave was set to 500 eV. The thresholds for electronic structure iteration and geometry relaxation were $10^{-5}$ eV and 0.03 eV/Å in force, respectively. Due to the conducting nature, the first order Methfessel-Paxton method with smearing of 0.1 eV was applied for optimization and tetrahedron method with Blöchl corrections was further used for the density of states (DOS) calculation. The lattices of $RuO_2$ and Cr and Ru oxides solid solution were relaxed based on fixed rutile symmetry. $9 \times 9 \times 13$ Monkhorst-Pack k-point grid was used to sample the Brillouin zone. Then four layered (110) facet was cleaved with the vacuum slab height of 20 Å. A $2 \times 1$ supercell containing 32 O and 16 metal atoms were studied with $5 \times 5 \times 1$ Monkhorst-Pack k-point grid. To describe vdW interaction, empirical Grimme's D3 correction was adopted.

The free energy of each species was calculated based on the following formula:

$$G = E_{dft} + E_{zpe} - T\Delta S \tag{5}$$

The zero-point energy and entropy correction were obtained from standard vibrational calculation, whereas the free energy of $O_2$ was derived according to experimental standard formation energy of liquid water:

$$G(O_2) = 4.92\ eV + 2G(H_2O) - 2G(H_2) \tag{6}$$

Moreover, we have also attempted to screen a series of potential solid solutions for further prediction, which are composed of $RuO_2$ and other rutile-like oxides, including $TiO_2$, $VO_2$, $CrO_2$, $MnO_2$, $GeO_2$, $NbO_2$, $MoO_2$, $RhO_2$, $SnO_2$, $WO_2$, and $PbO_2$. The cell sizes of the bulk models were allowed to relax in the calculations at the aforementioned level. The calculated theoretical lattice parameters are listed in Supplementary Table 9. Ideally, the closer cell parameters for the two $MO_2$ crystals, the higher possibility the solid solution can be formed. Besides, the atomic charges

on Ru atoms in these solid solution systems were calculated based on the Bader charge analysis. Here the higher positive partial charge compared with Ru in bulk $RuO_2$ indicates that the Ru ion in solid solution would donate electrons to other metals, and accordingly its oxidizing ability is strengthened to promote OER performance. For comparison, the number of valence electron and electronegativity of various metals are labeled in Supplementary Table 9. However, it seems that they have trivial influence on the electronic distribution on Ru.

## Data availability

The authors declare that all the published data supporting the findings of this study are available within the article and its supplementary information files.

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

## Acknowledgements

We thank Dr. Minghui Yang and Dr. Minghao Zhang for the help on the Rietveld refinement of $Cr_{0.6}Ru_{0.4}O_2$ structure. We thank Dr. Haifeng Zhao for very helpful discussions on EXAFS results. This work was financially supported by National Science Foundation of China (Nos. 51472255 and 51602320), the aided program for science and technology innovative research team of Ningbo municipality (No. 2015B11002). This research used computational resources of the High-Performance Computing Center of Collaborative Innovation Center of Advanced Microstructures, Nanjing University.

## Author contributions

L.C. and Y.L. designed the project and wrote the manuscript; Y.L. carried out the experiments; Z.T. carried out DFT calculations and wrote the computational part of manuscript; L.Z., J.M., and Z.J. performed the XANES and EXAFS experiments and data analysis. B.J.D. and R.G. provided helpful suggestions and polished the manuscript. All authors discussed the results and commented on the manuscript.

## Additional information

**Competing interests:** The authors declare no competing interests.

