## [Peer Review File · Nature Communications]

Reviewers' Comments:

Reviewer #1:

Remarks to the Author:

In the submitted manuscript, the authors report a Cr_{0.6}Ru_{0.4}O₂ material that shows promising activity and stability for OER in acidic solution. This material is constructed from the thermal decomposition of a high-surface area MOF material. The stability of the material was tested with both 10-h controlled current electrolysis and 10,000-cycle stability measurements. The Cr_{0.6}Ru_{0.4}O₂ catalyst shows promising activity per geometric area and TOFs of 0.1 s⁻¹ Ru⁻¹.

This manuscript has sufficient novelty and impact for publication in Nature Communications. Developing acid-stable OER electrocatalysts is crucial for enabling acidic water electrolysis, which is preferable to alkaline water electrolysis primarily due to the lower cost and higher ion conductivity of proton exchange membranes compared to anion exchange membranes. The conclusions presented here will be of interest to the broader water-splitting community.

There are only a few minor points the authors should address before publication.

- 1) The authors report TOF values but do not specify how these were calculated in the manuscript. Instead, the authors mention that the TOF was calculated using literature procedures and provide the references. Even though the authors are using literature procedures, the authors should still show exactly how TOF is calculated and, importantly, how the surface area is determined for TOF calculations (SA is not addressed anywhere in the manuscript). The authors should also provide the surface areas of both RuO₂ and Cr_{0.6}Ru_{0.4}O₂ in the manuscript.
- 2) The commercial RuO₂ chosen for this study shows particularly poor stability compared to other reported RuO₂ materials. This suggests that the RuO₂ chosen may not be an accurate representation of RuO₂ activity—not all commercial RuO₂ sources are equivalent. The authors must report the commercial source for the RuO₂ in the experimental section (it is not sufficient to say it is from a commercial source). In addition, the authors should comment on whether the RuO₂ stability and activity are typical of that reported in the literature (especially the full loss of catalytic activity in less than 2 hours constant electrolysis), and if not the authors should also compare their Cr_{0.6}Ru_{0.4}O₂ activity and stability to other reported RuO₂ materials in acidic solution.
- 3) In table S3, the authors should include specific activity and/or mass activity when comparing their results to previously reported materials if possible. Both specific activity and mass activity are better indications of intrinsic catalyst activity than overpotential at 10 mA cm⁻² (Boettcher et al, Chem Mater, 2017, 29, 120-140), and mass activity can be easily calculated from reported data and mass loadings.

Reviewer #2:

Remarks to the Author:

In this manuscript, the authors developed chromium-ruthenium oxide derived from metal-organic framework as an electrocatalyst for oxygen evolution reaction (OER). This catalyst exhibited a record-high activity and good stability, but the reviewer was not convinced by the evidence the authors provided. They showed that one catalyst Cr_{0.6}Ru_{0.4}O₂ possessed overwhelming activity and it was simply due to the electron withdrawing effect of Cr(IV) and the shift of O p-band center. The only variable in this manuscript was the annealing temperature, which merely influenced the catalytic properties. Specifically, the overpotential of Cr_{0.6}Ru_{0.4}O₂ was over 100 mV lower than that of commercial RuO₂; but the difference of overpotential due to annealing temperature was only 20 mV. It meant that the role of Cr in this system was much more important than annealing temperature. However, the authors did not discuss the properties of the catalysts with different Cr

composition. The reviewer agreed that the presence of Cr ions in RuO₂ had the positive effect on the catalysis, but the authors did not identify the role of Cr ions in the lattice and how they affected the catalytic properties. That is to say, the reviewer just read a manuscript showing some catalyst with high activity but lacking scientific discussion, which is crucial to the development of science. Moreover, the authors were not familiar with the electrochemistry and some discussion was not appropriate (listed below). In general, the results in this manuscript were too preliminary to publish in high quality journal Nature Communications.

Some detailed comments were listed below:

1. The authors claimed that the morphologies of Cr_{0.6}Ru_{0.4}O₂ powders were generally preserved after annealing in air. However, the morphology in Figure 3a was different from that in Figure 2b, which was not consistent with the argument the authors mentioned.
2. The authors claimed that the pore sizes of Cr_{0.6}Ru_{0.4}O₂ particle were larger than 10 nm and increase with the annealing temperature. However, it was not observed the pores of 10 nm from the result of TEM (the pore size of 10-100 nm should be identified by microscopy), so it was also not consistent with the description the authors disclosed.
3. in (3e) HAADF-STEM, the resolution was poor. It definitely had no subnanometer resolution.
4. Electrochemical impedance spectroscopy (EIS) were performed at 0.75 V. However, the OER reaction did not begin at that voltage, so the measured R_{ct} did not represent the charge transfer of OER. Cr_{0.6}Ru_{0.4}O₂(450) showed the best activity but R_{ct}, which is the crucial characteristics for catalysis, was obviously higher than the other two. R_{ct}, which refers to the charge transfer between the interface, is irrelevant to crystallinity. In addition, the results of R_{ct} in this manuscript were much higher than other OER electrocatalysts.
5. RuO₂ showed the R_{ct} of over 4000 Ω, indicating that the catalytic property was extremely poor, but it should be one of the best electrocatalysts. It showed some contradiction.
6. The authors claimed that low stability of Cr_{0.6}Ru_{0.4}O₂(450) electrocatalyst was caused by the poor crystallinity. Without further evidence, these two results should be the independent events.
7. The high stability should be identified by the variation of Cr/Ru composition during the reaction rather than simply comparing the morphology change, which was hard to determine in the porous structure.

Reviewer #3:

Remarks to the Author:

In this manuscript the authors have reported the OER catalytic activity of mixed phase Ru-Cr-oxide derived from Cr-modified MOF in acidic electrolyte. Designing efficient OER electrocatalysts has recently been in the center of attraction owing to the kinetically sluggish OER process and the challenges in reducing overpotential. A lot of research activities has been focused on designing catalysts for OER in alkaline medium because that is the more practical medium for water electrolysis. Contrary to what the authors claim, acidic medium electrolysis is not the ideal solution due to the highly corrosive nature of the acids, not only with respect to the catalyst, but also corrosion of the walls of reaction vessel, electrodes and so on. Hence, this research will have very limited novelty in the OER research field. Apart from that there are several major technical issues regarding less than desired characterization, missing analysis as well weak data interpretation that leads me not to recommend this manuscript in its present form to be published in Nature Communications.

Major issues:

1. The p_xrd patterns are especially weak. Also, they do not show any signature for RuCl₃ - why?
2. What is the mechanism of attachment of RuCl₃ within the MOF framework?
3. Instead of ICP-MS, which assumes an average relative composition of the elements to be

uniform over the entire sample composition, the authors should rather try to estimate the elemental ratio from the SEM-EDS or the STEM-EDS modes that can provide better evidence for local elemental composition for different regions of the sample. ICP-MS is a bulk characterization technique, while EDS is more microscopic technique and is more desirable in the present case.

4. Acidic medium typically leads to leaching of these metal ions into the electrolyte. The authors should characterize the electrolyte solution and check for any leached metal ions. If the metal ions are indeed leaching into the solution, that would lead to corrosion and corrosion current will show a similar LSV as OER.

5. It is not clear what kind of structure the authors are proposing for Cr-Ru-oxide. Do they mean solid solution as in CrO₂ and RuO₂ phases intimately mixed together or do they imply it is a Cr-doped RuO₂. While electron transfer between the metal centers can be envisioned in the later case, such mechanism of charge transfer will be very challenging (if not barely possible) in the former case. Cr-doped RuO₂ has been reported before and shows slightly different xrd pattern. The author needs to clear up the structural identification.

6. In the DFT calculations, the authors should show both the spin up and spin down plots and show their convergence to build confidence in their data. The d-orbital splitting is not very apparent in the DOS plots. In fact, the profile for O 2p and metal d- seems to be very similar both in terms of shape distribution as well as peak positions. This is highly unlikely. Even if the eg orbitals may overlap with O 2p and show delocalization below the Fermi level at the edge of the valence band, the t_{2g} orbitals should show a unique spread between the valence and conduction band, which is not visible in any of the DOS plots. This doesn't seem to be appropriate and the authors need to either repeat the DOS studies or provide an explanation for the observed discrepancy.

7. Is it possible to vary the relative ratio of Cr:Ru and if so how does the activity vary?

8. It is strange to see the solution resistance changing so much as has been calculated by the authors from the EIS data. Typically the solution resistance should stay the same in the same electrolyte. For such a changing value of solution resistance, how did they estimate the iR drop?

9. The degradation of the LSV (as apparent from the supporting document) is very concerning and again points to possible corrosion and leaching of the catalyst film. The authors should characterize the electrolyte solution, as well as measure elemental composition of the film after prolonged activity.

10. For the surface modeling, did the authors retain the Cr:Ru relative ratio in their model construct?

Response to referee 1

In the submitted manuscript, the authors report a $\text{Cr}_{0.6}\text{Ru}_{0.4}\text{O}_2$ material that shows promising activity and stability for OER in acidic solution. This material is constructed from the thermal decomposition of a high-surface area MOF material. The stability of the material was tested with both 10-h controlled current electrolysis and 10,000-cycle stability measurements. The $\text{Cr}_{0.6}\text{Ru}_{0.4}\text{O}_2$ catalyst shows promising activity per geometric area and TOFs of $0.1 \text{ s}^{-1} \text{ Ru}^{-1}$.

This manuscript has sufficient novelty and impact for publication in Nature Communications. Developing acid-stable OER electrocatalysts is crucial for enabling acidic water electrolysis, which is preferable to alkaline water electrolysis primarily due to the lower cost and higher ion conductivity of proton exchange membranes compared to anion exchange membranes. The conclusions presented here will be of interest to the broader water-splitting community.

There are only a few minor points the authors should address before publication.

Q1: The authors report TOF values but do not specify how these were calculated in the manuscript. Instead, the authors mention that the TOF was calculated using literature procedures and provide the references. Even though the authors are using literature procedures, the authors should still show exactly how TOF is calculated and, importantly, how the surface area is determined for TOF calculations (SA is not addressed anywhere in the manuscript). The authors should also provide the surface areas of both RuO_2 and $\text{Cr}_{0.6}\text{Ru}_{0.4}\text{O}_2$ in the manuscript.

Our response: We really thank the reviewer for this valuable suggestion. We have thus added the TOF calculation (*Nat. Energy* **1**, 16053 (2016), *J. Am. Chem. Soc.* **139**, 12076-12083 (2017)) in the Method section. In this calculation, all Ru atoms including a substantial portion in the bulk were treated as surface sites, thus underestimate the true TOF values. The BET surface areas of RuO_2 and $\text{Cr}_{0.6}\text{Ru}_{0.4}\text{O}_2$ have been added in the supporting information (Table S2).

Q2: The commercial RuO_2 chosen for this study shows particularly poor stability compared to other reported RuO_2 materials. This suggests that the RuO_2 chosen may not be an accurate representation of RuO_2 activity—not all commercial RuO_2 sources are equivalent. The authors must report the commercial source for the RuO_2 in the experimental section (it is not sufficient to say it is from a commercial source). In addition, the authors should comment on whether the

RuO₂ stability and activity are typical of that reported in the literature (especially the full loss of catalytic activity in less than 2 hours constant electrolysis), and if not the authors should also compare their Cr_{0.6}Ru_{0.4}O₂ activity and stability to other reported RuO₂ materials in acidic solution.

Our response: The RuO₂ was purchased from Sigma-Aldrich. We have added this information in the Method section. After a careful comparison with those reported RuO₂ for acidic solution, we find that the performance of RuO₂ used in our work is comparable to those reported in the literatures. We have added the summarization in Table S5.

Table S5. The OER activity of RuO₂ reported in literatures.

Catalyst	substrate	Electrolyte	Overpotential at specific current	Stability	Reference
RuO ₂	GCE	0.5 M H ₂ SO ₄	297@10 mA cm ⁻² 240@1 mA cm ⁻²	Chronopotentiometry @10 mA cm ⁻² : full loss activity in 1 hours	This work
RuO ₂	GCE	0.1 M HClO ₄	320 @ 1 mA cm ⁻²	Chronopotentiometry @ 1 mA cm ⁻² : full loss activity in 1.5 hours	J. Am. Chem. Soc. 2017, 139, 12076.
RuO ₂	GCE	0.1 M HClO ₄	430@10 mA cm ⁻²	--	J. Phys. Chem. Lett. 2012, 3, 399.
RuO ₂	Ti	0.5 M H ₂ SO ₄	240@1 mA cm ⁻²	--	Electrochimica Acta. 1998, 44, 1515.
RuO ₂	Au	0.5 M H ₂ SO ₄	230@1 mA cm ⁻²	CV measurements: activity dramatically decreased in 1000 cycles	J. Phys. Chem. C , 2016, 120, 2562–2573.
RuO ₂	GCE	0.5 M H ₂ SO ₄	289@10 mA cm ⁻²	CV measurements: activity dramatically decreased in 2000 cycles	J. Mater. Chem. A 2017, 5, 17221.

Q3: In table S3, the authors should include specific activity and/or mass activity when comparing their results to previously reported materials if possible. Both specific activity and mass activity are better indications of intrinsic catalyst activity than overpotential at 10 mA cm⁻² (Boettcher et al, Chem Mater, 2017, 29, 120-140), and mass activity can be easily calculated from reported data and mass loadings.

Our response: As suggested, we have added the mass activity at specific overpotential (270 mV) in table S6 (some of the reported literatures didn't provide the catalyst mass used, or the current density is too low to read from the LSV plot at the specific overpotential). It shows that the mass activity at 270 mV of $\text{Cr}_{0.6}\text{Ru}_{0.4}\text{O}_2$ (229 A g^{-1}) is much higher than those reported in literatures.

Response to referee 2

In this manuscript, the authors developed chromium-ruthenium oxide derived from metal-organic framework as an electrocatalyst for oxygen evolution reaction (OER). This catalyst exhibited a record-high activity and good stability, but the reviewer was not convinced by the evidence the authors provided. They showed that one catalyst $\text{Cr}_{0.6}\text{Ru}_{0.4}\text{O}_2$ possessed overwhelming activity and it was simply due to the electron withdrawing effect of Cr(IV) and the shift of O p-band center. The only variable in this manuscript was the annealing temperature, which merely influenced the catalytic properties. Specifically, the overpotential of $\text{Cr}_{0.6}\text{Ru}_{0.4}\text{O}_2$ was over 100 mV lower than that of commercial RuO_2 ; but the difference of overpotential due to annealing temperature was only 20 mV. It meant that the role of Cr in this system was much more important than annealing temperature. However, the authors did not discuss the properties of the catalysts with different Cr composition. The reviewer agreed that the presence of Cr ions in RuO_2 had the positive effect on the catalysis, but the authors did not identify the role of Cr ions in the lattice and how they affected the catalytic properties. That is to say, the reviewer just read a manuscript showing some catalyst with high activity but lacking scientific discussion, which is crucial to the development of science. Moreover, the authors were not familiar with the electrochemistry and some discussion was not appropriate (listed below). In general, the results in this manuscript were too preliminary to publish in high quality journal Nature Communications.

Q1: The only variable in this manuscript was the annealing temperature, which merely influenced the catalytic properties. Specifically, the overpotential of $\text{Cr}_{0.6}\text{Ru}_{0.4}\text{O}_2$ was over 100 mV lower than that of commercial RuO_2 ; but the difference of overpotential due to annealing temperature was only 20 mV. It meant that the role of Cr in this system was much more important than annealing temperature. However, the authors did not discuss the properties of the catalysts with different Cr composition.

Our response: We really thank the reviewer for this valuable comment. We have thus prepared additional samples with different Cr/Ru ratios (including $\text{Cr}_{0.91}\text{Ru}_{0.09}\text{O}_{2-\delta}$, $\text{Cr}_{0.83}\text{Ru}_{0.17}\text{O}_{2-\delta}$, $\text{Cr}_{0.72}\text{Ru}_{0.28}\text{O}_{2-\delta}$ and $\text{Cr}_{0.67}\text{Ru}_{0.33}\text{O}_2$), here δ was used to balance the valance of Cr^{3+} for the powders with mixed phase of Cr_2O_3 and CrO_2 - RuO_2 solid solution, and added this section in the manuscript. The revision is as follows:

We further synthesized a series of Cr-Ru oxides with different Cr ratios to investigate the Cr role on the catalytic property. By varying the mass of RuCl_3 in THF solution, we prepared MIL-101- RuCl_3 precursors with different RuCl_3 loading, and then obtained Cr-Ru oxides with Cr/Ru ratios varying from 9:1 to 6:4 after annealing (Cr/Ru ratios were determined by ICP-MS measurements). Figure 5 shows the corresponding morphology and structure evolution of Cr-Ru oxides. Directly annealing MIL-101(Cr) without loading RuCl_3 , we only obtained Cr_2O_3 nanoparticles with high crystallinity (Figure S18). After loading RuCl_3 into MIL-101 (Cr), CrO_2 - RuO_2 solid solution phase started to emerge after annealing. This is because RuO_2 and CrO_2 share the same rutile structure and have similar lattice constants, and the presence of Ru would induce the formation of RuO_2 - CrO_2 solid solution. For $\text{Cr}_{0.91}\text{Ru}_{0.09}\text{O}_{2-\delta}$ and $\text{Cr}_{0.83}\text{Ru}_{0.17}\text{O}_{2-\delta}$ (δ was used to balance the valance of Cr^{3+} for the powders with mixed phase of Cr_2O_3 and CrO_2 - RuO_2 solid solution) with low Ru content, the major phase is still Cr_2O_3 , which can be clearly observed from the PXRD patterns in Figure 5e. In contrast, for $\text{Cr}_{0.72}\text{Ru}_{0.28}\text{O}_{2-\delta}$ with higher Ru content, the CrO_2 - RuO_2 solid solution turn to be the major phase, and for $\text{Cr}_{0.67}\text{Ru}_{0.33}\text{O}_2$ and $\text{Cr}_{0.6}\text{Ru}_{0.4}\text{O}_2$, pure phase of CrO_2 - RuO_2 solid solution was formed. Note that, due to the saturation adsorption limit, we are unable to prepare Cr-Ru oxides with Cr/Ru ratio lower than 0.6/0.4. LSV results show that the OER performance of Cr-Ru oxides is highly correlated to the Ru/Cr ratio. $\text{Cr}_{0.91}\text{Ru}_{0.09}\text{O}_{2-\delta}$ and $\text{Cr}_{0.83}\text{Ru}_{0.17}\text{O}_{2-\delta}$ show moderate performance due to the high content of inactive Cr_2O_3 phase. With increasing Ru composition, the OER activity can be dramatically improved because the CrO_2 - RuO_2 solid solution evolved as the major phase or even pure phase (Figure 5f). However, it is noteworthy that the OER performance of $\text{Cr}_{0.91}\text{Ru}_{0.09}\text{O}_2$ with small amount of CrO_2 - RuO_2 solid solution phase is still higher than that of RuO_2 , highlighting the crucial role of Cr ions on the improved activity towards OER.

Figure 5. Evolution of chromium-ruthenium oxides with different Cr/Ru ratios: (a-d) TEM images of $\text{Cr}_{0.91}\text{Ru}_{0.09}\text{O}_{2-\delta}$, $\text{Cr}_{0.83}\text{Ru}_{0.17}\text{O}_{2-\delta}$, $\text{Cr}_{0.72}\text{Ru}_{0.28}\text{O}_{2-\delta}$ and $\text{Cr}_{0.67}\text{Ru}_{0.33}\text{O}_2$, respectively; (e) HR-TEM image of $\text{Cr}_{0.83}\text{Ru}_{0.17}\text{O}_{2-\delta}$; (f) PXRD patterns, Cr_2O_3 powder was obtained from directly annealing pure MIL-101 (Cr) ; (g) LSV results.

Figure S18. The PXRD pattern of the product by annealing pure MIL-101 (Cr) without loading RuCl₃. The reference PXRD pattern of Cr₂O₃ is obtained from Jade 2004 (JCPDS No. 06-0504)

Q2: The reviewer agreed that the presence of Cr ions in RuO₂ had the positive effect on the catalysis, but the authors did not identify the role of Cr ions in the lattice and how they affected the catalytic properties.

Our response: As suggested, we have performed X-ray absorption near edge structure (XANES) and extended X-ray absorption fine structure (EXAFS) analysis in Shanghai Synchrotron Radiation Facility (SSRF) to elucidate the structure at atomic level. In addition, we have solved the structure of Cr_{0.6}Ru_{0.4}O₂, and added the structure description in the revised manuscript. The combination of XANES and EXAFS clearly show the deliberate change to the local electronic environments of Ru and Cr ions, associated Ru-O and Cr-O bonds in the presence of both lattice Cr and Ru within CrO₂-RuO₂ solid solution. To corroborate the XANES and EXAFS results and unravel the critical role of Cr ions on the improved catalytic activity, we also carried out DFT calculations and analyzed the partial charge transfer. The revision is as follows:

Page 4, line 10-14; Figure 2d and Figure S3

The structure of Cr_{0.6}Ru_{0.4}O₂ is refined by Rietveld refinement (Figure S3). As shown in Figure 2d, Cr and Ru atoms are randomly distributed in the metal sites of the Cr_{0.6}Ru_{0.4}O₂ lattice. These metal atoms are edge-sharing and octahedrally coordinated to form chains along the [0 0 1] direction. Each chain is connected to four neighboring chains by shared corners. The MO₆ octahedra are tetragonally distorted, thus these M–O bond distances are not equal.

Figure S3. Experimental PXRD data vs. simulated $\text{Cr}_{0.6}\text{Ru}_{0.4}\text{O}_2$ structure. The PXRD is acquired by a very slow scan with a scan step of 0.005° and a scan rate of 4 second per step.

Figure 2. (a, b) PXRD patterns and SEM images of MIL-101 (Cr) before and after loading RuCl_3 ; (c) PXRD patterns of $\text{Cr}_{0.6}\text{Ru}_{0.4}\text{O}_2$ powders annealed at different temperatures. The reference patterns of CrO_2 and RuO_2 were obtained from Jade 2004 (JCPDS No.43-1040 and 43-1027); (d) Crystal structure of $\text{Cr}_{0.6}\text{Ru}_{0.4}\text{O}_2$: (left) packing image, (middle) unit cell, (right) corner

shared octahedral MO₆ structure. Color code: blue (60% Ru, 40% Cr), red (O). (e) 77 K N₂ adsorption/desorption isotherms of Cr_{0.6}Ru_{0.4}O₂ powders annealed at different temperatures.

Page 8, the second paragraph; Figure 7

To elucidate the atomic structure of Cr_{0.6}Ru_{0.4}O₂(550), X-ray absorption spectroscopy (XAS) characterization was further employed. Figure 7a shows the X-ray absorption near edge structure (XANES) of Ru K-edge region of the rutile-type Cr_{0.6}Ru_{0.4}O₂(550). Pure Ru metal foil and RuO₂ powder were also measured as reference. The shoulder near the absorption threshold of Ru foil corresponds to the heteroatomic interaction between the 5p states of the central Ru atom and the neighboring atoms.⁵² For RuO₂ and Cr_{0.6}Ru_{0.4}O₂(550), this shoulder was shifted and even merged into the near edge region, indicating the increased transition energy from 1s to the outmost shell orbitals on Ru atoms. The higher transition energy can be attributed to the formation of Ru-O bonds, which push up the empty state of 5d orbitals of Ru atoms.⁵² We further analyzed the absorption energy (E₀, determined from the first maximum in the first-order derivative), which is proportional to the oxidation state of transition metals.^{53,54} We found that the absorption energy for Cr_{0.6}Ru_{0.4}O₂(550) (E₀= 22129.9 eV) was similar with the value of RuO₂ (E₀= 22129.5 eV), implying that the oxidation state of Ru in Cr_{0.6}Ru_{0.4}O₂(550) is close to +4. The slightly higher absorption energy can be attributed to the electron withdrawing effect of the neighboring lattice Cr⁴⁺ ion, consistent with the XPS analysis results. Ru K-edge extended X-ray absorption fine structure (EXAFS) analysis was used to reveal the initial information on the Ru-O and Ru-Ru bonds. The corresponding Fourier transformed (FT) radial structure based on the k²-weighted EXAFS is displayed in Figure 7b. The peak at 1.59 Å for RuO₂ is associated with the back scattering of Ru-O in the first shell.²⁷ In contrast, the Ru-O bond length in Cr_{0.6}Ru_{0.4}O₂(550) is slightly shortened to 1.55 Å, in line with the slightly higher absorption energy for Cr_{0.6}Ru_{0.4}O₂(550). The peaks at 2.73 and 3.21 Å for RuO₂ arise from the back scatterings of Ru-Ru in the second and third shell.⁵⁵ These peaks for Cr_{0.6}Ru_{0.4}O₂(550) are assigned to the back scatterings Ru-Ru and Ru-Cr. The decreased intensity (i.e., vibrational amplitude) should be ascribed to the extremely small particle size (less than 15 nm).^{56,57} Furthermore, these peaks are also shifted to the left. Clearly, the presence of Cr can profoundly alter the local electronic structures of Ru and the associated Ru-O bonding, which directly determine the OER activity. Accordingly, Cr K-edge XANES and EXAFS were also used to examine the Cr oxide state, and

Cr-O bond in $\text{Cr}_{0.6}\text{Ru}_{0.4}\text{O}_2(550)$ (Figures 7c, d). The absorption energy of $\text{Cr}_{0.6}\text{Ru}_{0.4}\text{O}_2(550)$ ($E_0=6006.7$ eV) is higher than that of Cr metal ($E_0=5989.0$ eV), but consistent with the value of CrO_2 ($E_0=6006.8$ eV). In addition, the peak in the region of pre-edge absorption is also a characteristic of the formation of Cr^{4+} , corresponding to the 1s to 3d transition.⁵⁸ As shown in Figure 7d, the Cr-O length for CrO_2 is 1.47 Å. In $\text{Cr}_{0.6}\text{Ru}_{0.4}\text{O}_2(550)$, the Cr-O is slightly elongated to 1.50 Å, in accordance with the EXAFS result of Ru K-edge.

Indeed, the electron transfer from Ru to Cr and associated changes to the bond lengths were corroborated by the DFT calculations. Based on the Bader charge analysis, the partial charge of Ru in bulk RuO_2 is calculated to be +1.73|e|. While in $\text{CrO}_2\text{-RuO}_2$ solid solution, the Ru cation possesses charge of +1.92|e|. Simultaneously, the partial charge on the neighboring Cr cation decreases from +1.89|e| in CrO_2 to +1.84|e| in solid solution. Clearly, more positively charged Ru cation results in smaller cation radius, and the corresponding peaks in EXAFS slightly shift to the left. More importantly, it implies that highly oxidized state of Ru may significantly promote the oxidation of water to oxygen, i.e., OER reaction.

Figure 7. (a) Normalized Ru K-edge XANES spectra of $\text{Cr}_{0.6}\text{Ru}_{0.4}\text{O}_2$ (550), Ru foil and commercial RuO_2 . (b) Normalized Ru K-edge XANES spectra of $\text{Cr}_{0.6}\text{Ru}_{0.4}\text{O}_2$ (550), Ru foil and commercial RuO_2 . (c) Fourier transformed EXAFS spectra of Ru edge for $\text{Cr}_{0.6}\text{Ru}_{0.4}\text{O}_2$ (550), Ru foil and commercial RuO_2 . (d) Fourier transformed EXAFS spectra of Cr edge for $\text{Cr}_{0.6}\text{Ru}_{0.4}\text{O}_2$ (550), Cr foil and commercial CrO_2 .

Other detailed comments:

Q1: The authors claimed that the morphologies of $\text{Cr}_{0.6}\text{Ru}_{0.4}\text{O}_2$ powders were generally preserved after annealing in air. However, the morphology in Figure 3a was different from that in Figure 2b, which was not consistent with the argument the authors mentioned.

Our response: We apologize that our previous claim was not accurate. We have revised the related statement.

Page 4, line 15-16

SEM images show that the morphologies of $\text{Cr}_{0.6}\text{Ru}_{0.4}\text{O}_2$ powders became smaller, and their surface became much rougher after annealing (Figure 3a and S4).

Q2: The authors claimed that the pore sizes of $\text{Cr}_{0.6}\text{Ru}_{0.4}\text{O}_2$ particle were larger than 10 nm and increase with the annealing temperature. However, it was not observed the pores of 10 nm from the result of TEM (the pore size of 10-100 nm should be identified by microscopy), so it was also not consistent with the description the authors disclosed.

Our response: We apologize that we did not describe the pore structures accurately in the previous manuscript. As shown in the SEM and TEM images, every single $\text{Cr}_{0.6}\text{Ru}_{0.4}\text{O}_2$ particle is composed of much smaller nanocrystals (~ 15 nm). Actually, the mesopores were generated from the aggregation of these small nanocrystals, as shown in the TEM images. We have made it clear in the manuscript as follows:

Page 4, line 21-23

Barrett-Joyner-Halenda (BJH) pore size analysis revealed that the pore sizes of $\text{Cr}_{0.6}\text{Ru}_{0.4}\text{O}_2$ particle are larger than 10 nm (generated from the aggregation of nanocrystals in an individual particle as shown in TEM images) and increase with the annealing temperature (Figure S9).

Q3: In (3e) HAADF-STEM, the resolution was poor. It definitely had no subnanometer resolution.

Our response: We are very sorry for the misleading statements. We have thus deleted the related statements “...with subnanometer resolution”. Furthermore, we have also added elemental mapping images on a wider region (Figure S10) for comparison.

Figure S10. HAADF-STEM image and element mapping of $\text{Cr}_{0.6}\text{Ru}_{0.4}\text{O}_2(550)$.

Q4: Electrochemical impedance spectroscopy (EIS) were performed at 0.75 V. However, the OER reaction did not begin at that voltage, so the measured R_{ct} did not represent the charge transfer of OER. $\text{Cr}_{0.6}\text{Ru}_{0.4}\text{O}_2$ (450) showed the best activity but R_{ct} , which is the crucial characteristics for catalysis, was obviously higher than the other two. R_{ct} , which refers to the charge transfer between the interface, is irrelevant to crystallinity. In addition, the results of R_{ct} in this manuscript were much higher than other OER electrocatalysts.

Our response: We apologize that we missed the phrase “vs. $\text{Hg}/\text{Hg}_2\text{SO}_4$ ” in the previous manuscript so that the information was misleading. Actually, the electrochemical impedance spectroscopy (EIS) were performed at 0.75 V vs. $\text{Hg}/\text{Hg}_2\text{SO}_4$, i.e., 1.395 V vs. RHE. We have

also remeasured the EIS of $\text{Cr}_{0.6}\text{Ru}_{0.4}\text{O}_2$ (450), indeed the Rct is the smallest. However, we noted that the EIS of $\text{Cr}_{0.6}\text{Ru}_{0.4}\text{O}_2$ (450) is different before and after the first LSV measurement due to its unstable OER performance. The used EIS data of $\text{Cr}_{0.6}\text{Ru}_{0.4}\text{O}_2$ (450) in our previous manuscript was after the LSV measurement. Note that the other samples are relatively stable so that no big difference was observed before and after the first LSV measurements. We have thus revised it in the manuscript.

The EIS results are highly dependant on the voltage used, and generally higher voltage applied would lead to lower Rct. The EIS in our manuscript was measured at 1.395 V vs. RHE, in which the OER reaction on $\text{Cr}_{0.6}\text{Ru}_{0.4}\text{O}_2$ catalysts already took place. Note that no other literatures reported EIS data of OER electrocatalyst for acid condition at this voltage. For comparison, we measured the EIS of our catalyst at the voltages (1.42, 1.45 and 1.47V) used in literatures. Furthermore, we are aware that many studies used the area-specific impedance ($\Omega \text{ cm}^2$) because the impedance is also dependant on the surface area of electrode used. Therefore, we converted our EIS data to area-specific impedance for comparison. It shows that the Rct of $\text{Cr}_{0.6}\text{Ru}_{0.4}\text{O}_2$ (550) is rather small, with Rct of 45.1 Ω and area-specific Rct of 3.2 $\Omega \text{ cm}^2$ at 1.47 V, and 10.6 Ω or 0.7 $\Omega \text{ cm}^2$ at 1.42 V, which is smaller than those of reported in literatures (22.4 $\Omega \text{ cm}^2$ for $\text{Y}_2\text{Ru}_2\text{O}_{7-\delta}$ at 1.42 V (*J. Am. Chem. Soc.* **139**, 12076(2017)); 55.7 $\Omega \text{ cm}^2$ for $\text{IrO}_2@\text{RuO}_2$ at 1.47 V (*J. Phys. Chem. C*, **120**, 2562(2016))).

The revision is as follows:

Page 5, line 16-17; Figure 4b

The charge transfer resistance of $\text{Cr}_{0.6}\text{Ru}_{0.4}\text{O}_2$ electrocatalyst generally increases with the applied annealing temperature.

Figure 4. (a) Electrocatalytic oxygen evolution (OER) activities of $\text{Cr}_{0.6}\text{Ru}_{0.4}\text{O}_2$ (450-650) nanoparticles; (b) Nyquist plots at 1.395 V. Solid curves are the fitting results by using the equivalent circuit shown in the inset; (c) LSVs of $\text{Cr}_{0.6}\text{Ru}_{0.4}\text{O}_2$ (550) and commercial RuO_2 for the first and 10,000th cycle. Inset shows the comparison of overpotentials for $\text{Cr}_{0.6}\text{Ru}_{0.4}\text{O}_2$ (550) and RuO_2 at the current density of 10 mA cm^{-2} at the first and 10,000th cycle. For RuO_2 after 10,000 cycles, the overpotential is corresponded to 3.5 mA cm^{-2} which is the maximum current density of its LSV curve; (d) Tafel plots of $\text{Cr}_{0.6}\text{Ru}_{0.4}\text{O}_2$ (550) and RuO_2 at first and 10,000th cycle; (e) Chronopotentiometry performance under constant current density of 10 mA cm^{-2} up to 10 h. (f) TOF results of $\text{Cr}_{0.6}\text{Ru}_{0.4}\text{O}_2$ (550) and RuO_2 at the first and 10,000th cycle.

Page 5, line 25-27; Figure S14

In addition, we also measured the EIS spectra of $\text{Cr}_{0.6}\text{Ru}_{0.4}\text{O}_2$ (550) at higher voltages (Figure S14). The results show that $\text{Cr}_{0.6}\text{Ru}_{0.4}\text{O}_2$ (550) has rather small R_{ct} , with a value of $10.6 \ \Omega$ at 1.47 V, corresponding to the area-specific R_{ct} value of $0.7 \ \Omega \text{ cm}^2$.

Figure S14. (Left) Nyquist plots of Cr_{0.6}Ru_{0.4}O₂(550) at a series of voltages, (Right) The area-specific impedance (the electrode surface area is 0.07065 cm²).

Q5: RuO₂ showed the Rct of over 4000 Ω, indicating that the catalytic property was extremely poor, but it should be one of the best electrocatalysts. It showed some contradiction.

Our response: We are sorry that we did not provide sufficient measurements for comparison in the previous submission. In fact, the EIS results are highly dependant on the voltage used, and generally higher voltage applied would lead to lower Rct. The OER reaction of RuO₂ didnot begin at 1.395 V vs. RHE. We have thus measured the EIS spectra of RuO₂ at other voltages (1.45, 1.47, 1.50, 1.53 and 1.55 V) used in literatures. Many studies used the area-specific impedance (Ω cm²) because the impedance also depends on the surface area of electrode used. So we converted the EIS spectra to area-specific impedance for comparison. The area-specific Rct at 1.55 V is 3.2 Ω cm², comparable to the reported 2.0 Ω cm² (*J. Mater. Chem. A* **5**, 17221(2017)) and 6.3 Ω cm² (*J. Phys. Chem. C*, **120**, 2562(2016)). In addition, after a careful comparison with those reported RuO₂ for acidic solution, we find that the OER performance of RuO₂ used in our work is comparable to those reported in the literatures. We have added the summarization in Table S5.

The revision is as follows:

Here the high Rct of commercial RuO₂ is due to the fact that the OER reactions on RuO₂ catalyst do not occur at 1.395 V. We thus measured the EIS spectra of RuO₂ at higher voltages (Figure S13). It shows that the Rct of RuO₂ dramatically decreased with the increasing voltage applied. At 1.55 V, the Rct is 45.6 Ω, and the corresponding area-specific Rct is 3.2 Ω cm², comparable to those of literatures reported³⁸.

Figure S13. (Left) Nyquist plots of RuO₂ at a series of voltages; (Right) The area-specific impedance (the electrode surface area is 0.07065 cm²).

Page 6, line 12; Table S5

.....which are consistent with those reported in literatures (Table S5).

Table S5. The OER activity of RuO₂ reported in literatures.

Catalyst	substrate	Electrolyte	Overpotential specific current	at	Stability	Reference
RuO ₂	GCE	0.5 M H ₂ SO ₄	297@10 mA cm ⁻² 240@1 mA cm ⁻²		Chronopotentiometry @10 mA cm ⁻² : full loss activity in 1 hours	This work
RuO ₂	GCE	0.1 M HClO ₄	320 @ 1 mA cm ⁻²		Chronopotentiometry @ 1 mA cm ⁻² : full loss activity in 1.5 hours	J. Am. Chem. Soc. 2017, 139, 12076.

RuO ₂	GCE	0.1 M HClO ₄	430@10 mA cm ⁻²	--	J. Phys. Chem. Lett. 2012, 3, 399.
RuO ₂	Ti	0.5 M H ₂ SO ₄	240@1 mA cm ⁻²	--	Electrochimica Acta. 1998, 44, 1515.
RuO ₂	Au	0.5 M H ₂ SO ₄	230@1 mA cm ⁻²	CV measurements: activity dramatically decreased in 1000 cycles	J. Phys. Chem. C, 2016, 120, 2562–2573.
RuO ₂	GCE	0.5 M H ₂ SO ₄	289@10 mA cm ⁻²	CV measurements: activity dramatically decreased in 2000 cycles	J. Mater. Chem. A 2017, 5, 17221.

Q6. The authors claimed that low stability of Cr_{0.6}Ru_{0.4}O₂(450), electrocatalyst was caused by the poor crystallinity. Without further evidence, these two results should be the independent events.

Our response: We really appreciate this important comment. We agree with this reviewer that further evidence should be provided to support the relevance of poor crystallinity and low stability. For this purpose, we conducted XANES characterization of Cr_{0.6}Ru_{0.4}O₂(450). Cr K-edge XANES show the intensity of the pre-edge peak of Cr_{0.6}Ru_{0.4}O₂ (450) is above that of Cr_{0.6}Ru_{0.4}O₂ (550), indicating a lower symmetry environment of the Cr atoms in Cr_{0.6}Ru_{0.4}O₂ (450). This difference confirmed the fine structure difference of Cr_{0.6}Ru_{0.4}O₂(450) and Cr_{0.6}Ru_{0.4}O₂(550). We further carried out EDS measurement of Cr_{0.6}Ru_{0.4}O₂(450) after 10,000 CV cycles. The result (Figure R1) revealed a dramatic change to the Cr/Ru composition of Cr_{0.6}Ru_{0.4}O₂ (450) after OER reactions (from 0.56:0.44 to 0.43:0.57 after 10,000 cycles), which implies the substantial leaching of Cr and Ru metal during the reactions. Accordingly, the OER performance decayed. It is noteworthy that the leaching of metal in the samples annealed at higher temperature is much smaller, which will be shown in the response to Q7 later.

Nevertheless, we agree that the above two results still might be the independent events, although the latter can explain the degradation of performance of Cr_{0.6}Ru_{0.4}O₂(450). Therefore, to be safe, we have deleted the claim “... was caused by the poor crystallinity” and added the discussion on the unstable structures under acidic conditions. The revision is as follows:

For $\text{Cr}_{0.6}\text{Ru}_{0.4}\text{O}_2$ electrocatalyst annealed at $450\text{ }^\circ\text{C}$, the overpotential (at 10 mA cm^{-2}) dramatically decreased from 177 mV at the first cycle to 242 mV at the $10,000^{\text{th}}$ cycle (Figure S15) due to the relatively unstable structures under the acidic solutions.

Figure R1. EDS spectra of $\text{Cr}_{0.6}\text{Ru}_{0.4}\text{O}_2$ (450) after 10,000 CV cycles.

Page 9, line 13-17; Figure S22

For comparison, we also measured the Cr K-edge XANES of $\text{Cr}_{0.6}\text{Ru}_{0.4}\text{O}_2$ (450). As shown in Figure S22, the intensity of the pre-edge peak of $\text{Cr}_{0.6}\text{Ru}_{0.4}\text{O}_2$ (450) is above that of $\text{Cr}_{0.6}\text{Ru}_{0.4}\text{O}_2$ (550), indicating a lower symmetry environment of the Cr atoms in $\text{Cr}_{0.6}\text{Ru}_{0.4}\text{O}_2$ (450).⁵⁸ It confirms that the fine structure of $\text{Cr}_{0.6}\text{Ru}_{0.4}\text{O}_2$ (450) is different from that of $\text{Cr}_{0.6}\text{Ru}_{0.4}\text{O}_2$ (550), i.e., the pure phase rutile Cr-Ru oxide has not been well formed.

Figure S22. (left) Normalized Cr K-edge XANES spectra for Cr_{0.6}Ru_{0.4}O₂ (450), Cr_{0.6}Ru_{0.4}O₂ (550) and reference CrO₂. (Right) Shows the amplified pre-edge region and the inset diagram shows the energy level of possible transitions.

Q7: The high stability should be identified by the variation of Cr/Ru composition during the reaction rather than simply comparing the morphology change, which was hard to determine in the porous structure.

Our response: As suggested, we have performed ICP-MS experiments to measure the amount of Ru and Cr dissolved after 10,000 CV cycles. The results indicate that less than 2.5% Ru and less than 8% Cr of Cr_{0.6}Ru_{0.4}O₂ (550) were dissolved in the acidic electrolyte solution. Correspondingly, the Cr/Ru composition slightly changed from 0.6/0.4 to 0.58/0.42. Note that such leaching content is smaller than those of excellent OER catalysts for acidic condition (for example: 3.33% Ru leaching for Y₂Ru₂O_{7- δ} after 10,000 CV cycles reported in *J. Am. Chem. Soc.* **139**, 12076(2017), the leaching of Y was not reported; 12.7% Ba and 9.5% Pr leaching for in Ba₂PrIrO₆ after electrolysis experiments of 1 h at constant electrode potential of 1.45 V reported in *Nat. Commun.* **7**, 12363(2016); ~80% Co and ~70% Cu leaching of Co-IrCu ONC after 2,000 CV cycles reported in *Adv. Func. Mater.* **27**, 1604688(2017))

The revision is as follows:

In addition, the ICP-MS experiments (Table S4) show that less than 2.5% Ru and 8% Cr of $\text{Cr}_{0.6}\text{Ru}_{0.4}\text{O}_2$ (550) were dissolved in the acidic electrolyte solution after 10,000 cycles. Note that such leaching content is smaller than those of recently reported excellent OER catalysts for acidic condition.^{23, 27}

Table S4. ICP-MS analysis of dissolved Ru and Cr ions after 10,000 CV cycles in 0.5 M H_2SO_4 .

Sample amount	20 ug	20 ug	40 ug	60 ug
Concentration of Ru ion (ppb)	1.92	2.15	2.6	5.04
Concentration of Cr ion (ppb)	4.33	5.08	9.02	16.12
Loss of mass (Ru)	2.47%	2.76%	1.66%	2.16%
Loss of mass (Cr)	7.19%	8.42%	7.44%	7.02%
Average mass loss (Ru)	2.26%			
Average mass loss (Cr)	7.51%			

Response to referee 3

In this manuscript the authors have reported the OER catalytic activity of mixed phase Ru-Cr-oxide derived from Cr-modified MOF in acidic electrolyte. Designing efficient OER electrocatalysts has recently been in the center of attraction owing to the kinetically sluggish OER process and the challenges in reducing overpotential. A lot of research activities has been focused on designing catalysts for OER in alkaline medium because that is the more practical medium for water electrolysis. Contrary to what the authors claim, acidic medium electrolysis is not the ideal solution due to the highly corrosive nature of the acids, not only with respect to the catalyst, but also corrosion of the walls of reaction vessel, electrodes and so on. Hence, this research will have very limited novelty in the OER research field. Apart from that there are several major technical issues regarding less than desired characterization, missing analysis as well weak data interpretation that leads me not to recommend this manuscript in its present form to be published in Nature Communications.

Our response: We appreciate the valuable comments from the reviewer. We agree with the reviewer that OER in alkaline medium is particular important because it is the more practical medium for water electrolysis, while acidic medium electrolysis has some severe issues related to corrosion. Nevertheless, the development of efficient and stable OER catalysts that operate under acidic pH regimes is also becoming important and has received increasing attention more recently (for example, please see *Nat. Chem.* **10**, 24 (2018); *Adv. Mater.* **29**, 1703798(2017); *J. Am. Chem. Soc.* **139**, 12076(2017); *Science* **353**, 1011(2016); *Nat. Commun.* **7**, 12363(2016)). Compared with alkaline conditions, OER catalysis under acidic conditions also has its own advantages because acidic electrolyte has higher ionic conductivity and fewer unfavorable reactions. Indeed, the state-of-the-art water electrolysis assemblies are often based on proton exchange membranes (PEM), e.g., Nafion, that create acidic local environment during operation. Therefore, we believe that it can appeal a broad readership in the research community of chemistry, materials and energy.

Q1 and Q2: The pXRD patterns are especially weak. Also, they do not show any signature for RuCl₃ - why? What is the mechanism of attachment of RuCl₃ within the MOF framework?

Our response: We have remeasured the PXRD of Cr-MIL-101-RuCl₃ (Figure 2a). The PXRD pattern and intensity of MIL-101 (Cr) are typical and consistent with those reported in literatures

(for example: *Science* **309**, 2040-2042 (2005), *Sci. Rep.* **3**, 1859 (2013)). RuCl₃ did not crystallize in the MIL-101-RuCl₃ pores, but were adsorbed on the pore surface. It is known that MIL-101(Cr) is highly porous, possesses ultrahigh surface area (> 3000 m² g⁻¹), thus can readily adsorb RuCl₃. In addition, MIL-101 (Cr) possesses open metal sites and –OH group, which could also serve as the adsorption sites for RuCl₃. Indeed, MOFs have been used to remove heavy metal ions from water (for example: *J. Solid State Chem.* **262**, 135 (2018), *Sep. Purif. Tech.* **194**, 462(2018)). In addition, we present the digital photographs of MIL-101 (Cr) before and after loading RuCl₃. The color of MIL-101 (Cr) changed from light green to brown (the color of RuCl₃) after loading RuCl₃, visually indicating the successful loading.

The revision is as follows:

Page 3, line 12-14 and 22-23; Figure 2a; Figure S1

.....After loading RuCl₃, the color of MIL-101 (Cr) changed from light green to brown (the color of RuCl₃) (Figure S1), visually indicating the successful loading of RuCl₃.

.....There was no peak for RuCl₃, indicating that RuCl₃ did not crystallize in the pores of MIL-101 (Cr) but was adsorbed on the pore surface.....

Figure S1. Digital photographs of MIL-101(Cr) before (left) and after (right) loading RuCl₃.

Figure 2. (a, b) PXRD patterns and SEM images of MIL-101 (Cr) before and after loading RuCl₃; (c) PXRD patterns of Cr_{0.6}Ru_{0.4}O₂ powders annealed at different temperatures. The reference patterns of CrO₂ and RuO₂ were obtained from Jade 2004 (JCPDS No.43-1040 and 43-1027); (d) Crystal structure of Cr_{0.6}Ru_{0.4}O₂ (550): (left) packing image, (middle) unit cell, (right) corner shared octahedral MO₆ structure. Color code: blue (60% Ru, 40% Cr), red (O). (e) 77 K N₂ adsorption/desorption isotherms of Cr_{0.6}Ru_{0.4}O₂ powders annealed at different temperatures.

Q3: Instead of ICP-MS, which assumes an average relative composition of the elements to be uniform over the entire sample composition, the authors should rather try to estimate the elemental ration from the SEM-EDS of the STEM-EDS modes that can provide better evidence for local elemental composition for different regions of the sample. ICP-MS is a bulk characterization technique, while EDS is more microscopic technique and is more desirable in the present case.

Our response: As suggested, we have conducted STEM-EDS analysis for the local elemental composition. In addition, the elemental mapping images for wider region are also provided in the revised manuscript. Overall, the EDS result (Cr/Ru ratio is 0.56:0.44) is consistent with the composition Cr_{0.6}Ru_{0.4}O₂ identified in ICP-MS. The revision is as follows:

.....(mapping images for a wider region are shown in Figure S10). In addition, the EDS analysis indicates that Cr/Ru ratio is 0.56:0.44, generally consistent with the ICP-MS result (Figure S11).

Figure S10. HAADF-STEM image and elemental mapping of $\text{Cr}_{0.6}\text{Ru}_{0.4}\text{O}_2(550)$.

Figure S11. EDS results of $\text{Cr}_{0.6}\text{Ru}_{0.4}\text{O}_2(550)$, the element of Cu comes from carbon-coated copper grid.

Q4: Acidic medium typically leads to leaching of these metal ions into the electrolyte. The authors should characterize the electrolyte solution and check for any leached metal ions. If the metal ions are indeed leaching into the solution, that would lead to corrosion and corrosion current will show a similar LSV as OER.

Our response: As suggested, we have thus measured the amount of Ru and Cr dissolved from the $\text{Cr}_{0.6}\text{Ru}_{0.4}\text{O}_2$ (550) sample after 10,000 CV cycles by means of ICP-MS. The result shows that the amount of Ru dissolved was below 2.5% and the amount of Cr dissolved was below 8% in the acidic electrolyte solution. Such leaching amount is smaller than those of recently reported excellent OER catalysts for acidic condition reported recently (for example: 3.33% Ru leaching for $\text{Y}_2\text{Ru}_2\text{O}_{7-\delta}$ after 10,000 CV cycles reported in *J. Am. Chem. Soc.* **139**, 12076(2017), the leaching of Y was not reported; 12.7% Ba and 9.5% Pr leaching for in $\text{Ba}_2\text{PrIrO}_6$ after electrolysis experiments of 1 h at constant electrode potential of 1.45 V reported in *Nat. Commun.* **7**, 12363(2016);); ~80% Co and ~70% Cu leaching of Co-IrCu ONC after 2,000 CV cycles reported in *Adv. Func. Mater.* **27**, 1604688(2017)). We agree with the reviewer that corrosion current would contribute to the LSV plot, however, this contribution is very limited in our case. For comparison, we have measured the LSV of the acid unstable Cr_2O_3 (i.e., with more leaching metal ions) under the same condition (Figure 5g). The result shows that even the corrosion current of the acid unstable Cr_2O_3 is negligible compared to the LSV results of $\text{Cr}_{0.6}\text{Ru}_{0.4}\text{O}_2$ (550).

The revision is as follows:

Page 6, line 2-5; Table S4

In addition, the ICP-MS experiments (Table S4) show that less than 2.5% Ru and 8% Cr of $\text{Cr}_{0.6}\text{Ru}_{0.4}\text{O}_2$ (550) were dissolved in the acidic electrolyte solution after 10,000 cycles. Note that such leaching content is smaller than those of recently reported excellent OER catalysts for acidic condition.^{23, 27}

Table S4 ICP-MS analysis of dissolved Ru and Cr ions after 10,000 CV cycles in 0.5 M H_2SO_4 .

Sample amount	20 ug	20 ug	40 ug	60 ug
---------------	-------	-------	-------	-------

Concentration of Ru ion (ppb)	1.92	2.15	2.6	5.04
Concentration of Cr ion (ppb)	4.33	5.08	9.02	16.12
Loss of mass (Ru)	2.47%	2.76%	1.66%	2.16%
Loss of mass (Cr)	7.19%	8.42%	7.44%	7.02%
Average mass loss (Ru)	2.26%			
Average mass loss (Cr)	7.51%			

Q5: It is not clear what kind of structure the authors are proposing for Cr-Ru-oxide. Do they mean solid solution as in CrO_2 and RuO_2 phases intimately mixed together or do they imply it is a Cr-doped RuO_2 . While electron transfer between the metal centers can be envisioned in the later case, such mechanism of charge transfer will be very challenging (if not barely possible) in the former case. Cr-doped RuO_2 has been reported before and shows slightly different pxd pattern. The author needs to clear up the structural identification.

Our response: The Cr-Ru oxide herein is a CrO_2 and RuO_2 solid solution, which is a purely single phase material. The lattice constants of our Cr-Ru oxide are between that of CrO_2 and RuO_2 which share the same rutile structure and have similar lattice constants. We have solved the structure by Rietveld refinement, and added the structure description in our revised manuscript. In addition, we have performed X-ray absorption near edge structure (XANES) and X-ray absorption fine structure (EXAFS) analysis in Shanghai Synchrotron Radiation Facility (SSRF) to elucidate the structure at atomic level. The combination of XANES and EXAFS clearly show the deliberate change to the local electronic environments of Ru and Cr ions, associated Ru-O and Cr-O bonds in the presence of both lattice Cr and Ru within CrO_2 - RuO_2 solid solution. The revision is as follows:

Page 4, line 10-16; Figure 2d; Figure S3

The structure of $\text{Cr}_{0.6}\text{Ru}_{0.4}\text{O}_2$ is refined by Rietveld refinement (Figure S3). As shown in Figure 2d, Cr and Ru atoms are randomly distributed in the metal sites of the $\text{Cr}_{0.6}\text{Ru}_{0.4}\text{O}_2$ lattice. These metal atoms are edge-sharing and octahedrally coordinated to form chains along the [0 0 1] direction. Each chain is connected to four neighboring chains by shared corners. The MO_6

octahedra are tetragonally distorted, thus these M–O bond distances are not equal. SEM images show that the morphologies of $\text{Cr}_{0.6}\text{Ru}_{0.4}\text{O}_2$ powders became smaller, and their surface became much rougher after annealing (Figures 3a and S4).

Figure S3. Experimental PXRD data vs. simulated $\text{Cr}_{0.6}\text{Ru}_{0.4}\text{O}_2$ structure. The PXRD is acquired by a very slow scan with a scan step of 0.005° and a scan rate of 4 second per step.

Figure 2. (a, b) PXRD patterns and SEM images of MIL-101 (Cr) before and after loading RuCl₃; (c) PXRD patterns of Cr_{0.6}Ru_{0.4}O₂ powders annealed at different temperatures. The reference patterns of CrO₂ and RuO₂ were obtained from Jade 2004 (JCPDS No.43-1040 and 43-1027); (d) Crystal structure of Cr_{0.6}Ru_{0.4}O₂: (left) packing image, (middle) unit cell, (right) corner shared octahedral MO₆ structure. Color code: blue (60% Ru, 40% Cr), red (O). (e) 77 K N₂ adsorption/desorption isotherms of Cr_{0.6}Ru_{0.4}O₂ powders annealed at different temperatures.

Page 8, the second paragraph; Figure 7

To elucidate the atomic structure of Cr_{0.6}Ru_{0.4}O₂(550), X-ray absorption spectroscopy (XAS) characterization was further employed. Figure 7a shows the X-ray absorption near edge structure (XANES) of Ru K-edge region of the rutile-type Cr_{0.6}Ru_{0.4}O₂(550). Pure Ru metal foil and RuO₂ powder were also measured as reference. The shoulder near the absorption threshold of Ru foil corresponds to the heteroatomic interaction between the 5p states of the central Ru atom and the neighboring atoms.⁵² For RuO₂ and Cr_{0.6}Ru_{0.4}O₂(550), this shoulder was shifted and even merged into the near edge region, indicating the increased transition energy from 1s to the outmost shell orbitals on Ru atoms. The higher transition energy can be attributed to the formation of Ru-O bonds, which push up the empty state of 5d orbitals of Ru atoms.⁵² We further analyzed the absorption energy (E_0 , determined from the first maximum in the first-order derivative), which is proportional to the oxidation state of transition metals.^{53,54} We found that the absorption energy for Cr_{0.6}Ru_{0.4}O₂(550) ($E_0 = 22129.9$ eV) was similar with the value of RuO₂ ($E_0 = 22129.5$ eV), implying that the oxidation state of Ru in Cr_{0.6}Ru_{0.4}O₂(550) is close to +4. The slightly higher absorption energy can be attributed to the electron withdrawing effect of the neighboring lattice Cr⁴⁺ ion, consistent with the XPS analysis results. Ru K-edge extended X-ray absorption fine structure (EXAFS) analysis was used to reveal the initial information on the Ru-O and Ru-Ru bonds. The corresponding Fourier transformed (FT) radial structure based on the k^2 -weighted EXAFS is displayed in Figure 7b. The peak at 1.59 Å for RuO₂ is associated with the back scattering of Ru-O in the first shell.²⁷ In contrast, the Ru-O bond length in Cr_{0.6}Ru_{0.4}O₂(550) is slightly shortened to 1.55 Å, in line with the slightly higher absorption energy for Cr_{0.6}Ru_{0.4}O₂(550). The peaks at 2.73 and 3.21 Å for RuO₂ arise from the back scatterings of Ru-Ru in the second and third shell.⁵⁵ These peaks for Cr_{0.6}Ru_{0.4}O₂(550) are assigned to the back scatterings Ru-Ru and Ru-Cr. The decreased intensity (i.e., vibrational amplitude) should be

ascribed to the extremely small particle size (less than 15 nm).^{56,57} Furthermore, these peaks are also shifted to the left. Clearly, the presence of Cr can profoundly alter the local electronic structures of Ru and the associated Ru-O bonding, which directly determine the OER activity. Accordingly, Cr K-edge XANES and EXAFS were also used to examine the Cr oxide state, and Cr-O bond in $\text{Cr}_{0.6}\text{Ru}_{0.4}\text{O}_2(550)$ (Figures 7c, d). The absorption energy of $\text{Cr}_{0.6}\text{Ru}_{0.4}\text{O}_2(550)$ ($E_0=6006.7$ eV) is higher than that of Cr metal ($E_0=5989.0$ eV), but consistent with the value of CrO_2 ($E_0=6006.8$ eV). In addition, the peak in the region of pre-edge absorption is also a characteristic of the formation of Cr^{4+} , corresponding to the 1s to 3d transition.⁵⁸ As shown in Figure 7d, the Cr-O length for CrO_2 is 1.47 Å. In $\text{Cr}_{0.6}\text{Ru}_{0.4}\text{O}_2(550)$, the Cr-O is slightly elongated to 1.50 Å, in accordance with the EXAFS result of Ru K-edge.

Figure 7. (a) Normalized Ru K-edge XANES spectra of $\text{Cr}_{0.6}\text{Ru}_{0.4}\text{O}_2(550)$, Ru foil and commercial RuO_2 . (b) Normalized Cr K-edge XANES spectra of $\text{Cr}_{0.6}\text{Ru}_{0.4}\text{O}_2(550)$, Ru foil and commercial RuO_2 . (c) Fourier transformed EXAFS spectra of Ru edge for $\text{Cr}_{0.6}\text{Ru}_{0.4}\text{O}_2(550)$, Ru foil and commercial RuO_2 . (d) Fourier transformed EXAFS spectra of Cr edge for $\text{Cr}_{0.6}\text{Ru}_{0.4}\text{O}_2(550)$, Cr foil and commercial CrO_2 .

Ru foil and commercial RuO₂. (d) Fourier transformed EXAFS spectra of Cr edge for Cr_{0.6}Ru_{0.4}O₂ (550), Cr foil and commercial CrO₂.

Q6: In the DFT calculations, the authors should show both the spin up and spin down plots and show their convergence to build confidence in their data. The d-orbital splitting is not very apparent in the DOS plots. In fact, the profile for O 2p and metal d- seems to be very similar both in terms of shape distribution as well as peak positions. This is highly unlikely. Even if the eg orbitals may overlap with O 2p and show delocalization below the Fermi level at the edge of the valence band, the t_{2g} orbitals should show a unique spread between the valence and conduction band, which is not visible in any of the DOS plots. This doesn't seem to be appropriate and the authors need to either repeat the DOS studies or provide an explanation for the observed discrepancy.

Our response: Thanks for the helpful comments and suggestions. We thus redid and double checked the DOS calculations, and also added the DOS with both spin up and spin down and detailed pDOS in Figure 8a and Figure S23-24, which is consistent with the recent DFT calculations results of IrO₂ (*Phys. Rev. Lett.* **112**, 117601(2014)), IrO₂, CrO₂ and IrO₂-CrO₂ interfaces (*Phys. Rev. B*, **84**, 054438 (2011)). Note that the spin up and spin down channels of RuO₂ are identical because of the nonmagnetic nature. We added the discussions of electronic structures as follows:

Page 10, the first paragraph; Figure 8a; Figure S23-24

Considering the less electron number of Cr⁴⁺ (2e⁻) than that of Ru⁴⁺ (4e⁻), the Fermi level of solid solution is shifted downward due to the band filling effect. Correspondingly, the O *p*-band center moves closer to the Fermi level in Cr₅Ru₃O₁₆ (-2.48 eV), compared with the value of -2.91 eV in RuO₂. Clearly, the upshift O *p*-band suggests the enhanced activity for OER. The detailed projected DOS of Cr₅Ru₃O₁₆ are displayed in Figure S23-24, also consistent with previous theoretical study on CrO₂-RuO₂ structures.⁶¹ The oxygen 2*p* (O-*p*) states below ~1.5 eV overlap with part of the metal *d*-bands. Metal t_{2g} orbitals show a unique spread and strong peak at the edge of valence band, especially in Ru *d*-orbitals. Interestingly, the O-*p* orbital (the major component is below -2 eV) and Ru-*d* bands at higher energy state are well separated in RuO₂

(Figure 8a). In contrast, the relatively low energy Cr t_{2g} orbitals can enhance the hybridization of O- p orbital, thus further push O p -center closer to Fermi level in the solid solution. Note that the empty e_g orbitals of Cr intensely strengthen the DOS ranging from about 2 to 6 eV and the increased DOS related to σ antibonding state suggests a weak Cr-O binding strength.

Figure 8. (a) Density of states of RuO_2 and $\text{Cr}_5\text{Ru}_3\text{O}_{16}$; (b) The four-step OER process; (c) The calculated free energy diagrams for RuO_2 and $\text{Cr}_5\text{Ru}_3\text{O}_{16}$.

Figure S23. (Left) Projected density of states of Cr atom of $\text{Cr}_5\text{Ru}_3\text{O}_{16}$; (Right) Projected density of states of Cr t_{2g} and e_g of $\text{Cr}_5\text{Ru}_3\text{O}_{16}$.

Figure S24. (Left) Projected density of states of Ru atom of $\text{Cr}_5\text{Ru}_3\text{O}_{16}$; (Right) Projected density of states of Ru t_{2g} and e_g of $\text{Cr}_5\text{Ru}_3\text{O}_{16}$.

Q7: Is it possible to vary the relative ratio of Cr:Ru and if so how does the activity vary?

Our response: We have thus prepared additional samples with different Cr/Ru ratios (including $\text{Cr}_{0.91}\text{Ru}_{0.09}\text{O}_{2-\delta}$, $\text{Cr}_{0.83}\text{Ru}_{0.17}\text{O}_{2-\delta}$, $\text{Cr}_{0.72}\text{Ru}_{0.28}\text{O}_{2-\delta}$ and $\text{Cr}_{0.67}\text{Ru}_{0.33}\text{O}_2$), here δ was used to balance the valance of Cr^{3+} for the powders with mixed phase of Cr_2O_3 and $\text{CrO}_2\text{-RuO}_2$ solid solution, and added this section in the manuscript. The revision is as follows:

Page 7, the second paragraph; Figure 5; Figure S18

We further synthesized a series of Cr-Ru oxides with different Cr ratios to investigate the Cr role on the catalytic property. By varying the mass of RuCl_3 in THF solution, we prepared MIL-101- RuCl_3 precursors with different RuCl_3 loading, and then obtained Cr-Ru oxides with Cr/Ru ratios varying from 9:1 to 6:4 after annealing (Cr/Ru ratios were determined by ICP-MS measurements). Figure 5 shows the corresponding morphology and structure evolution of Cr-Ru oxides. Directly annealing MIL-101(Cr) without loading RuCl_3 , we only obtained Cr_2O_3 nanoparticles with high crystallinity (Figure S18). After loading RuCl_3 into MIL-101 (Cr), $\text{CrO}_2\text{-RuO}_2$ solid solution phase started to emerge after annealing. This is because RuO_2 and CrO_2 share the same rutile structure and have similar lattice constants, and the presence of Ru would induce the formation of $\text{RuO}_2\text{-CrO}_2$ solid solution. For $\text{Cr}_{0.91}\text{Ru}_{0.09}\text{O}_{2-\delta}$ and $\text{Cr}_{0.83}\text{Ru}_{0.17}\text{O}_{2-\delta}$ (δ was used to balance the valance of Cr^{3+} for the powders with mixed phase of Cr_2O_3 and $\text{CrO}_2\text{-RuO}_2$ solid solution) with low Ru content, the major phase is still Cr_2O_3 , which can be clearly observed from the PXRD patterns in Figure 5e. In contrast, for $\text{Cr}_{0.72}\text{Ru}_{0.28}\text{O}_{2-\delta}$ with higher Ru content, the $\text{CrO}_2\text{-RuO}_2$ solid solution turn to be the major phase, and for $\text{Cr}_{0.67}\text{Ru}_{0.33}\text{O}_2$ and $\text{Cr}_{0.6}\text{Ru}_{0.4}\text{O}_2$, pure phase of $\text{CrO}_2\text{-RuO}_2$ solid solution was formed. Note that, due to the saturation adsorption limit, we are unable to prepare Cr-Ru oxides with Cr/Ru ratio lower than 0.6/0.4. LSV results show that the OER performance of Cr-Ru oxides is highly correlated to the Ru/Cr ratio. $\text{Cr}_{0.91}\text{Ru}_{0.09}\text{O}_{2-\delta}$ and $\text{Cr}_{0.83}\text{Ru}_{0.17}\text{O}_{2-\delta}$ show moderate performance due to the high content of inactive Cr_2O_3 phase. With increasing Ru composition, the OER activity can be dramatically improved because the $\text{CrO}_2\text{-RuO}_2$ solid solution evolved as the major phase or even pure phase (Figure 5f). However, it is noteworthy that the OER performance of $\text{Cr}_{0.91}\text{Ru}_{0.09}\text{O}_2$

with small amount of $\text{CrO}_2\text{-RuO}_2$ solid solution phase is still higher than that of RuO_2 , highlighting the crucial role of Cr ions on the improved activity towards OER.

Figure 5. Evolution of chromium-ruthenium oxides with different Cr/Ru ratios: (a-d) TEM images of $\text{Cr}_{0.91}\text{Ru}_{0.09}\text{O}_{2-\delta}$, $\text{Cr}_{0.83}\text{Ru}_{0.17}\text{O}_{2-\delta}$, $\text{Cr}_{0.72}\text{Ru}_{0.28}\text{O}_{2-\delta}$ and $\text{Cr}_{0.67}\text{Ru}_{0.33}\text{O}_2$, respectively; (e) HR-TEM image of $\text{Cr}_{0.83}\text{Ru}_{0.17}\text{O}_{2-\delta}$; (f) PXRD patterns, Cr_2O_3 powder was obtained from directly annealing pure MIL-101 (Cr) ; (g) LSV results.

Figure S18. The PXRD pattern of the product by annealing pure MIL-101 (Cr) without loading RuCl_3 . The reference PXRD pattern of Cr_2O_3 is obtained from Jade 2004 (JCPDS No. 06-0504)

Q8: It is strange to see the solution resistance changing so much as has been calculated by the authors from the EIS data. Typically the solution resistance should stay the same in the same electrolyte. For such a changing value of solution resistance, how did they estimate the iR drop?

Our response: In Table S3, the R_{sol} values for different catalysts are 9.4, 10.8, 7.9, 9.3, 9.4 and 9.6 Ω . This deviation is comparable to those of literatures reported. In addition, at 10 mA cm^{-2} , the difference of iR drop using 7.9 Ω and 10.8 Ω is only 2 mV (the electrode surface area is 0.07065 cm^2). We used the average solution resistance for the iR drop calculation.

Q9: The degradation of the LSV (as apparent from the supporting document) is very concerning and again points to possible corrosion and leaching of the catalyst film. The authors should characterize the electrolyte solution, as well as measure elemental composition of the film after prolonged activity.

Our response: We really appreciate this important comment and agree that the degradation of the LSV should be related to corrosion and leaching of the catalyst film. As suggested, we have

measured the amount of Ru and Cr dissolved from the $\text{Cr}_{0.6}\text{Ru}_{0.4}\text{O}_2$ (550) sample after 10,000 CV cycles by means of ICP-MS. The result shows that the amount of Ru dissolved was below 2.5% and the amount of Cr dissolved was below 8% in the acidic electrolyte solution. Such leaching amount is smaller than those of recently reported excellent OER catalysts for acidic condition reported recently (for example: 3.33% Ru leaching for $\text{Y}_2\text{Ru}_2\text{O}_{7-\delta}$ after 10,000 CV cycles reported in *J. Am. Chem. Soc.* **139**, 12076(2017), the leaching of Y was not reported; 12.7% Ba and 9.5% Pr leaching for in $\text{Ba}_2\text{PrIrO}_6$ after electrolysis experiments of 1 h at constant electrode potential of 1.45 V reported in *Nat. Commun.* **7**, 12363(2016);); ~80% Co and ~70% Cu leaching of Co-IrCu octahedral nanocages after 2000 CV cycles reported in *Adv. Func. Mater.* **27**, 1604688(2017)). We further measured the EDS of $\text{Cr}_{0.6}\text{Ru}_{0.4}\text{O}_2$ (550) after 10,000 CVs (Figure R2). The EDS spectra also show a relatively small difference in the Cr/Ru composition before (0.56:0.44) and after (0.5:0.5) 10,000 cycles.

Figure R2. EDS spectra of $\text{Cr}_{0.6}\text{Ru}_{0.4}\text{O}_2$ (550) after 10,000 CV cycles.

The revision is as follows:

Page 6, line 2-5; Table S4

In addition, the ICP-MS experiments (Table S4) show that less than 2.5% Ru and 8% Cr of $\text{Cr}_{0.6}\text{Ru}_{0.4}\text{O}_2$ (550) were dissolved in the acidic electrolyte solution after 10,000 cycles, which

results in the slight degradation of OER performance. Note that such leaching content is smaller than those of recently reported excellent OER catalysts for acidic condition.^{23,27}

Table S4 ICP-MS analysis of dissolved Ru and Cr ions after 10,000 CV cycles in 0.5 M H₂SO₄.

Sample amount	20 ug	20 ug	40 ug	60 ug
Concentration of Ru ion (ppb)	1.92	2.15	2.6	5.04
Concentration of Cr ion (ppb)	4.33	5.08	9.02	16.12
Loss of mass (Ru)	2.47%	2.76%	1.66%	2.16%
Loss of mass (Cr)	7.19%	8.42%	7.44%	7.02%
Average mass loss (Ru)	2.26%			
Average mass loss (Cr)	7.51%			

Q10: For the surface modeling, did the authors retain the Cr:Ru relative ratio in their model construct?

Our response: Yes, we retained the Cr:Ru relative ratio in our surface model. For clarification, we have reorganized the discussions and Figure 8c:

Page 9, the second paragraph; Page 10, the first and second paragraph

On the other hand, we also calculated the free energy profiles of OER to directly compare the OER activities of RuO₂ and CrO₂-RuO₂ solid solution. A slab model containing 32 O and 16 metal atoms was employed, in which the Cr/Ru ratio was kept as 5:3. Here we considered the a four-step OER mechanism with CHE model to provide a general view.⁶²⁻⁶⁴ The (110) facet was chosen as the surface model, because it has been identified as the most stable. We constructed surfaces of both RuO₂ and solid solution for comparison. Five different configurations of solid solution surface were further modeled to average the calculated energy barriers. As shown in Figure 8b, the five-coordinated surface Ru was identified as the reactive adsorption site. Interestingly, under the oxidation condition in water, the Ru site could readily adsorb OH to form *OH. For all the models, the formation of *OOH was found to be the rate determining step

(RDS). On the $\text{Cr}_5\text{Ru}_3\text{O}_{16}$ surface, the free energy change of RDS at the Ru site was calculated to be 1.87 eV, which is approximately 0.1 eV lower than that on RuO_2 surface (2.02 eV) (Figure 8c). It is consistent with the decreased overpotential of 100 mV measured in experiments

Reviewers' Comments:

Reviewer #1:

Remarks to the Author:

Chen et al. have reported a Cr_{0.6}Ru_{0.4}O₂ material that shows promising activity and stability for water oxidation in acidic solution. Acidic water electrolysis is emerging as an important and active area of research in the water splitting community. Although alkaline conditions are more common and perhaps less corrosive than acidic solutions, there are still advantages to acidic water splitting when one considers large-scale electrolyzer implementation including: better ionic conductivity through proton-exchange membranes (as compared to more resistive anion-exchange membranes) and faster kinetics at Pt-based cathodes in acidic solutions. Companies such as Proton Onsite actually have developed commercial PEM water electrolysis systems that work in acidic medium, although they often rely on proprietary iridium-based catalysts which further highlights the desire for cheaper acid-stable anode materials.

I believe that the authors have appropriately addressed most reviewer concerns and included new experimental data to better support their conclusions. In particular, the authors have included XANES and EXAFS data that show a very slight electron-withdrawing effect on the Ru (as evidenced by a very small positive shift in the XANES Ru K-edge features and a slight shortening of the Ru-O bond via EXAFS measurements) which is consistent with XPS data and DFT calculations. In addition, the authors have explored a variety of Cr/Ru oxides with varying compositions to further address the role of Cr incorporation into the RuO₂ lattice on OER activity. They have also clarified their EIS and stability measurements and included ICP-MS analysis of their electrolyte after their stability measurements to quantify Ru and Cr leaching, and have included mass activity comparisons to other acid-OER systems.

I believe this manuscript has adequately addressed reviewer comments and been suitably revised for publication in Nature Communications.

Reviewer #2:

Remarks to the Author:

According to the updated information the authors provided, the reviewer still suggested to reject this manuscript because the data was not consistent with the results the authors claimed and the authors still did not provide the conclusive role of Cr ions. The detailed concerns were shown below:

1. The authors prepared the samples with different Cr/Ru ratios and claimed the sample formed CrO₂-RuO₂ solid solution in most samples. However, the XRD patterns in Figure 5f did not prove the characteristics of the solid solution, which the peak position would shift with the composition, especially the peak angle around 35. Thus, their samples should be composed of some separated phases, possibly dominated by RuO₂. Most importantly, if the proposed solid solution was not confirmed, the calculation and the built crystal were not convinced.
2. The fixed peak positions of their samples were not consistent with the results in XAS (Figure 7c-d), in which the bond distance of Ru-O and Cr-O varied.
3. Again, the authors were unfamiliar with the fundamental concept of XAS. The shoulder near absorption threshold should be the pre-edge feature, which belongs to the excitation from 1s to 4d, and the lattice symmetry determined the peak intensity. The energy transition of XANES was the excitation from 1s to 5p rather than 5d orbitals.
4. The oxidation status of Ru and Cr ions mutually influenced each other in Figure 7. However, the data in Figure 6 showed the chemical state of Ru ions changed between RuO₂ and CrRuO₂ but

unvaried between CrO₂ and CrRuO₂. It meant that the XAS result was not consistent with that of XPS.

5. The authors also did not elucidate the specific role of Cr ions in their system in the revised manuscript. It showed that Ru dominated the catalytic abilities because the activities increased with the amount of Ru in these samples, especially the tiny Ru in Cr_{0.91}Ru_{0.09}O would exhibit higher activity than pristine RuO₂. Therefore, the reviewer believed that some synergistic effect of Cr ions should improve this system. Unfortunately, the authors did not mention it.

Reviewer #3:

Remarks to the Author:

In this revised manuscript concerning the enhanced OER catalytic activity in a solid solution of CrO₂-RuO₂ system derived from transition metal doped open-framework solids, the authors have tried to address the previous concerns of the reviewers. However, in spite of the revisions done by the authors, the manuscript remains sub-par and unsuitable to be published in Nature Communications. I must also mention that not all of the reviewers' concerns has been addressed satisfactorily.

1. The authors claim that the presence of even minute amounts of Cr⁴⁺ in the solid solution has a profound effect in the OER activity compared to pure RuO₂, although pure Cr₂O₃ is relatively inactive. First of all, comparing CrO₂ and Cr₂O₃ catalytic activities is not justified, since the oxidation states of Cr in these two compounds is very different, and OER depends a lot on the oxidation state and chemical potential of the metal ions. Secondly, this argument would be more convincing if the authors can compare their catalytic activity with ball-milled or mixed powders of CrO₂ and RuO₂ with their annealed powders. Such comparison will throw better insight into the role of Cr⁴⁺ as a participating lattice ion (as in a solid solution) or presence in the matrix (assisting in charge transport between the catalytic grains).

2. The particle size in the annealed powders are shown to be very small. Catalytic activity is significantly influenced by particle size. Have the authors compared the activity with RuO₂ of similar particle sizes (i.e. <15 nm). As had been mentioned in the previous reviews, the RuO₂ performance shown in this article is significantly lower than those reported in the literature. Although the authors have selected few RuO₂ publications to show that they are comparable, they have left out majority of recent reports of highly active RuO₂ catalysts with nanostructured grains which show much better activity.

3. For the slab model to describe the OH adsorption and further OER steps in RuO₂ and Cr₅Ru₃O₁₆, the authors have considered the 110 plane, since "it has been identified as the most stable". However, this can be very misleading too, since the catalytic activity can depend strongly on the nature, composition, and surface energy of the specific lattice plane. The authors may try to compare other the energetics of the catalytic reaction on other lattice planes with higher and lower concentration of Ru ions, as well as only surface Ru and sub-surface Cr ions, respectively to build confidence in their claim.

4. There is a pre-oxidation peak appearing around 1.3 V at higher Ru content. What does that peak correspond to?

5. Since their particle sizes are small and the powder is derived from porous framework materials, the authors should also report the ECSA, roughness factor, and preferably report current density with respect to the ECSA (specific current density).

6. The discussion part of the manuscript is rather short and unsatisfactory. Except for mentioning that inclusion of Cr reduces e-density around Ru and increases the oxidation state, and enhancing

the catalytic activity of a Cr-rich Ru-poor compound, it doesn't really go into the details of how this can be further used in a predictive way to design better catalysts. Ruthenates are well known stoichiometric compounds, and such a discussion can help the readers gain knowledge to advance further in this field.

Response to referee 2

Q1 and Q2. The authors prepared the samples with different Cr/Ru ratios and claimed the sample formed $\text{CrO}_2\text{-RuO}_2$ solid solution in most samples. However, the XRD patterns in Figure 5f did not prove the characteristics of the solid solution, which the peak position would shift with the composition, especially the peak angle around 35° . Thus, their samples should be composed of some separated phases, possibly dominated by RuO_2 . Most importantly, if the proposed solid solution was not confirmed, the calculation and the built crystal were not convinced. The fixed peak positions of their samples were not consistent with the results in XAS (Figure 7c-d), in which the bond distance of Ru-O and Cr-O varied.

Our response: To clearly show the peak shift, we have added a straight line for those samples with $\text{CrO}_2\text{-RuO}_2$ solid solution as the major phase or pure phase in Figure 5f (Figure R1). Clearly, the peak around 35° shifts to the left side as increasing the Ru content, which is a characteristic of $\text{RuO}_2\text{-CrO}_2$ solid solution. In addition, as we have presented in our manuscript, the HAADF-STEM EDS mapping images indicate that Cr, Ru and O atoms distribute uniformly in each individual nanocrystal. HR-TEM image (Figure 3e) and the corresponding fast Fourier transform (FFT, Figure 3f) indicate that these nanocrystals are indeed single crystals. Furthermore, we performed atomic-resolution HAADF-STEM (Figure 3h-j) and EELS mapping characterization (Figure 3k). The atomic-resolution HAADF-STEM images clearly confirm the well crystallized single nanocrystals. EELS analysis of a randomly selected region in a single nanocrystal confirmed the coexistence of Ru and Cr atoms (the inset in Figure 3i). EELS elemental mapping (Figure 3k) showed a uniform uncorrelated spatial distribution of Cr, Ru and O. These results definitely demonstrate that the product is $\text{CrO}_2\text{-RuO}_2$ solid-solution, instead of a combination of separated phases.

It should be noted that, as shown in our manuscript, directly annealing MIL-101(Cr) only led to the formation of Cr_2O_3 . The presence of Ru plays the key role to induce the formation of rutile-structured CrO_2 . This is because RuO_2 and CrO_2 share the same rutile structure and have similar lattice constants, and thus the presence of Ru would induce the formation of $\text{RuO}_2\text{-CrO}_2$ solid solution. The revision is as follows:

Page 7, the last line; Page 8, the first line

Note that, the peak around 35° shifts to the left side as the Ru content increases, which is a characteristic of $\text{RuO}_2\text{-CrO}_2$ solid solution.

Figure 5. Evolution of chromium-ruthenium oxides with different Cr/Ru ratios: (a-d) TEM images of $\text{Cr}_{0.91}\text{Ru}_{0.09}\text{O}_{2-\delta}$, $\text{Cr}_{0.83}\text{Ru}_{0.17}\text{O}_{2-\delta}$, $\text{Cr}_{0.72}\text{Ru}_{0.28}\text{O}_{2-\delta}$ and $\text{Cr}_{0.67}\text{Ru}_{0.33}\text{O}_2$, respectively; (e) HR-TEM image of $\text{Cr}_{0.83}\text{Ru}_{0.17}\text{O}_{2-\delta}$; (f) PXRD patterns, Cr_2O_3 powder was obtained from directly annealing pure MIL-101 (Cr); (g) LSV results.

Page 4, line 18-19

High resolution TEM (HR-TEM) image (Figure 3e) and the corresponding fast Fourier transform (FFT, Figure 3f) indicate that these nanocrystals are indeed single-crystalline.

Page 4, the last two lines; Page 5, line 1-4

Furthermore, we performed atomic-resolution HAADF-STEM and electron energy loss spectroscopy (EELS) mapping characterization. As shown in Figure 3h-j, the atomic-resolution HAADF-STEM images clearly demonstrate the well crystallized single nanocrystals. EELS analysis of a randomly selected region

in a single nanocrystal confirmed the coexistence of Ru and Cr atoms. EELS elemental mapping with subnanometer resolution (Figure 3k) showed a uniform uncorrelated spatial distribution of Cr, Ru and O.

Figure 3. Morphology and elemental mapping of $\text{Cr}_{0.6}\text{Ru}_{0.4}\text{O}_2$ (550). (a) SEM image; (b) Dark field TEM image; (c) TEM image; (d) HR-TEM image; (h) HR-TEM image of a single nanocrystal; (i) The corresponding FFT image; (g) HAADF-STEM image, corresponding EDS element mapping showing the

distribution of Cr, Ru and O; (h-j) atomic-resolution HAADF-STEM images and EELS analysis (inset of i); (k) EELS maps.

Q3. Again, the authors were unfamiliar with the fundamental concept of XAS. The shoulder near absorption threshold should be the pre-edge feature, which belongs to the excitation from 1s to 4d, and the lattice symmetry determined the peak intensity. The energy transition of XANES was the excitation from 1s to 5p rather than 5d orbitals.

Our response: We agree with the reviewer that the shoulder near absorption threshold is the pre-edge feature, and the intensity feature of the electric dipole forbidden Ru 1s to 4d transitions is dependent on the 5p mixing into 4d orbitals (lattice symmetry determine the mixing). Thus, for clearness, we have rewritten the statement “The shoulder near the absorption threshold of Ru foil corresponds to the heteroatomic interaction between the 5p states of the central Ru atom and 4d/5p orbitals of the neighboring atoms.” in the revised manuscript. Note that similar statement has been reported by other researchers recently (for example, see *J. Am. Chem. Soc.* **139**, 12076, 2017).

We agree that the expression “The higher transition energy can be attributed to the formation of Ru-O bonds, which push up the empty state of 5d orbitals of Ru atoms” is misleading and not accurate. After discussed with some experts in Shanghai synchrotron center and the authors who firstly used this expression (*J. Mater. Chem. A* **3**, 1518-1529 (2015)), we have corrected the statements. The revision is as follows:

Page 8, the last two lines; Page 9, line 1-4

The shoulder near the adsorption threshold of Ru foil corresponds to the 1s to 4d transition. For RuO₂ and Cr_{0.6}Ru_{0.4}O₂(550), this shoulder is weaker, because the increased lattice symmetry prevents the mixing of 4d and 5p orbitals. The observed transition energy of XANES (corresponding to the 1s to 5p transition) for RuO₂ and Cr_{0.6}Ru_{0.4}O₂ is higher than that for Ru. This can be attributed to the formation of Ru-O bonds, which pushes up the empty state of 5p orbitals of Ru atoms.⁵²

Q4. The oxidation status of Ru and Cr ions mutually influenced each other in Figure 7. However, the data in Figure 6 showed the chemical state of Ru ions changed between RuO₂ and CrRuO₂ but unvaried between CrO₂ and CrRuO₂. It meant that the XAS result was not consistent with that of XPS.

Our response : As suggested, we have carefully remeasured the Cr XPS spectrum of CrO₂-RuO₂ sample.

As shown in Figure 6b, a slight shift to lower binding energy can be observed compared to CrO₂. This small shift is consistent with the slight lower adsorption energy (0.1 eV) for Cr atoms in CrO₂-RuO₂ solid

solution compared to CrO_2 , as shown in Cr K-edge XANES (page 9, line 22-24). On the other hand, the relative shift of Ru state in CrRuO_2 and RuO_2 measured in XPS and XAS are also consistent with each other. The revision is as follows:

Page 25, Figure 6b

Figure 6. (a) XPS of $\text{Cr}_{0.6}\text{Ru}_{0.4}\text{O}_2$ (550) and RuO_2 for Ru 3d regions. (b) XPS of $\text{Cr}_{0.6}\text{Ru}_{0.4}\text{O}_2$ (550) and CrO_2 for Cr 2p regions. The blue and red smoothing lines are fitting results of the sum of individual components. For Ru 3d, color codes are used to distinguish the different spin-orbit components, dark cyan for primary Ru $3d_{3/2}$ and $3d_{5/2}$ spin states, and light magenta for satellite Ru $3d_{3/2}$ and $3d_{5/2}$ spin states.

Page 9 line 25-26

The slightly lower absorption energy can be attributed to the electron withdrawing effect of Cr^{4+} ion, in agreement with the XPS results and Ru oxidation analysis.

Q5. The authors also did not elucidate the specific role of Cr ions in their system in the revised manuscript. It showed that Ru dominated the catalytic abilities because the activities increased with the amount of Ru in these samples, especially the tiny Ru in $\text{Cr}_{0.91}\text{Ru}_{0.09}\text{O}_2$ would exhibit higher activity than

pristine RuO₂. Therefore, the reviewer believed that some synergistic effect of Cr ions should improve this system. Unfortunately, the authors did not mention it.

Our response: Actually, we have emphasized the synergistic effect of Cr ions in our manuscript (please see Page 7, the last two lines of the second paragraph in our last version). In addition, in our manuscript, we have found that Ru atoms are the active sites for the OER, and the Cr ion in the lattice affects the electron and valance of Ru. A combination of XAS, XPS experiments, together with DFT calculations were performed to elucidate the specific role of Cr ions. All results revealed that the Cr ions in the lattice can profoundly alter the electronic structures of Ru and thus improve the OER activity. It should be noted that, compared with the enhancement for the samples with Cr₂O₃ as the major phase, the enhancement of OER for the samples with CrO₂-RuO₂ solid solution as the major or the pure phase is very slight, indicating the importance of Cr. Although we cannot prepare CrO₂-RuO₂ solid solution samples with higher Ru content due to the adsorption limit, it is reasonable to conclude that there is a point where the OER performance will decrease with the increase of Ru due to the poor OER performance of the pure RuO₂. To further verify the important role of Cr in the lattice, we prepared mixed CrO₂ and RuO₂ by ball-milling, the result showed that the OER performance is even lower than that of pure RuO₂ (Figure 5g), that is, physically mixing the RuO₂ with CrO₂ (i.e., separated phases) cannot enhance the OER performance.

The Cr_{0.91}Ru_{0.09}O_{2-δ} with only a small amount of CrO₂-RuO₂ solid solution (Cr₂O₃ is the major phase) exhibit higher activity than pristine RuO₂, also confirm the importance of Cr role in lattice of the CrO₂-RuO₂ solid solution. Based on our DFT calculation and EXAFS results, we conclude that the specific role of Cr in CrO₂-RuO₂ solid solution lattice is the electron-withdrawing effect which could affect the electron density and valance of Ru that could largely affect the OER performance (note that the mechanism of OER is an electron transfer process).

The revision is as follows:

Page 8, line 8-10

Notably, the mixed RuO₂ and CrO₂ sample shows very poor OER performance, even much lower than that of pristine RuO₂, indicating the synergistic effect of Cr⁴⁺ role as a participating lattice ion.

Figure 5. Evolution of chromium-ruthenium oxides with different Cr/Ru ratios: (a-d) TEM images of $\text{Cr}_{0.91}\text{Ru}_{0.09}\text{O}_{2-\delta}$, $\text{Cr}_{0.83}\text{Ru}_{0.17}\text{O}_{2-\delta}$, $\text{Cr}_{0.72}\text{Ru}_{0.28}\text{O}_{2-\delta}$ and $\text{Cr}_{0.67}\text{Ru}_{0.33}\text{O}_2$, respectively; (e) HR-TEM image of $\text{Cr}_{0.83}\text{Ru}_{0.17}\text{O}_{2-\delta}$; (f) PXRD patterns, Cr_2O_3 powder was obtained from directly annealing pure MIL-101 (Cr); (g) LSV results.

Response to referee 3

Q1. The authors claim that the presence of even minute amounts of Cr^{4+} in the solid solution has a profound effect in the OER activity compared to pure RuO_2 , although pure Cr_2O_3 is relatively inactive. First of all, comparing CrO_2 and Cr_2O_3 catalytic activities is not justified, since the oxidation states of Cr in these two compounds is very different, and OER depends a lot on the oxidation state and chemical potential of the metal ions. Secondly, this argument would be more convincing if the authors can compare their catalytic activity with ball-milled or mixed powders of CrO_2 and RuO_2 with their annealed powders. Such comparison will throw better insight into the role of Cr^{4+} as a participating lattice ion (as in a solid solution) or presence in the matrix (assisting in charge transport between the catalytic grains).

Our response: We agree with the reviewer that the oxidation states of Cr in these two compounds are very different, and OER depends a lot on the oxidation state and chemical potential of the metal ions. Here we presented the OER performances of Cr_2O_3 , CrO_2 and $\text{CrO}_2\text{-RuO}_2$ solid solutions in Figure 5g not to compare the CrO_2 and Cr_2O_3 catalytic activities. Our purpose is just to show that Cr_2O_3 , the product of pristine MIL-101 precursor, is inactive for OER.

We thank the reviewer for this valuable suggestion. As suggested, we prepared mixed powders of CrO_2 and RuO_2 by ball-milling, and performed the OER measurement. As shown in Figure 5g, the OER performance of mixed CrO_2 and RuO_2 powder is even lower than that of pristine RuO_2 , indicating the importance of Cr^{4+} role as a participating lattice ion. The revision is as follows:

Page 8, line 8-10

Notably, the mixed RuO_2 and CrO_2 sample shows very poor OER performance, even much lower than that of pristine RuO_2 , indicating the important synergistic effect of Cr^{4+} role as a participating lattice ion.

Q2. The particle size in the annealed powders are shown to be very small. Catalytic activity is significantly influenced by particle size. Have the authors compared the activity with RuO_2 of similar particle sizes (i.e. <15 nm). As had been mentioned in the previous reviews, the RuO_2 performance shown in this article is significantly lower than those reported in the literature. Although the authors have selected few RuO_2 publications to show that they are comparable, they have left out majority of recent reports of highly active RuO_2 catalysts with nanostructured grains which show much better activity.

Our response: We are aware that although there are quite a few reports on RuO_2 for OER in basic condition, the reports on RuO_2 for OER in acidic condition are relatively rare (please see Chem. Soc. Rev.

46, 337, 2017), mainly due to its instability and relative lower OER activity in acidic condition than that in basic condition. Among these reports focused on acidic condition, we note that Lee et al. (J. Phys. Chem. Lett. 3, 399, 2012) reported the OER performance of nano-sized RuO₂ with size of ~6 nm, but the OER performance is still very low, with overpotential of ~275 mV at **1 mA cm⁻²**, not 10 mA cm⁻², the benchmark condition in our work. In addition, we also characterized the morphology of the commercial RuO₂ used in our work. The TEM images showed that the particle size of commercial RuO₂ is ~30 nm (Figure S17).

Furthermore, as suggested by the reviewer, to compare the OER performance, we analyzed the ECSA and roughness factors of the samples, and calculated the current density with respect to the ECSA (see Figure S18-20, Table S6). The results show that the enhanced activity of OER performance of CrO₂-RuO₂ solid solution is not just increased by ECSA, and the intrinsic activity arising from the lattice Cr ions plays a more important role.

The revision is as follows:

Figure R2 and Figure S17

Figure R2. SEM images of the commercial RuO₂.

Figure S17. TEM image of the commercial RuO_2 nanoparticles.

Page 7, line 6-10; Figure S18-20; Table S12

In addition, we further calculated the electrochemically active surface area (ECSA), roughness factor of $\text{Cr}_{0.6}\text{Ru}_{0.4}\text{O}_2$ (550) and RuO_2 electrode, and plotted the LSVs with respect to the ECSA (Figure S18-20, Table S6). The results show that the enhanced activity of OER performance of CrO_2 - RuO_2 solid solution is not just increased by ECSA, the intrinsic activity arising from the Cr ions plays a more important role.

Figure S18. (Left) CVs of $\text{Cr}_{0.6}\text{Ru}_{0.4}\text{O}_2$ collected at various scan rates (20, 60, 100, 140 and 180 mV s^{-1}); (Right) Capacitive current at 1.26 V (vs. RHE) against the scan rate and the corresponding C_{DL} value.

Figure S19. (Left) CVs of RuO_2 collected at various scan rates (20, 60, 100, 140 and 180 mV s^{-1}); (Right) Capacitive current at 1.26 V (vs. RHE) against the scan rates and the corresponding C_{DL} value.

Figure S20. ECAS based LSVs of $\text{Cr}_{0.6}\text{Ru}_{0.4}\text{O}_2(550)$ and RuO_2 .

Table S6. ECSA parameters for $\text{Cr}_{0.6}\text{Ru}_{0.4}\text{O}_2(550)$ and RuO_2 .

Catalyst	C_{DL}	ECSA	RF
$\text{Cr}_{0.6}\text{Ru}_{0.4}\text{O}_2(550)$	2.58 mF	73.7 cm^2	1043
RuO_2	0.32 mF	9.1 cm^2	129

Q3. For the slab model to describe the OH adsorption and further OER steps in RuO_2 and $\text{Cr}_5\text{Ru}_3\text{O}_{16}$, the authors have considered the 110 plane, since "it has been identified as the most stable". However, this can be very misleading too, since the catalytic activity can depend strongly on the nature, composition, and surface energy of the specific lattice plane. The authors may try to compare other the energetics of the catalytic reaction on other lattice planes with higher and lower concentration of Ru ions, as well as only surface Ru and sub-surface Cr ions, respectively to build confidence in their claim.

Our response: As suggested, we have investigated two more planes, (101) and (200) surfaces, which have slightly higher surface energy (the calculated surface energies are shown in Table S8) and were observed in our HR-TEM images. Moreover, we also supplemented two cases with extremely low Ru concentration and layered structure as the reviewer suggested. All these results indicate that the formation of solid solution with CrO₂ has a positive impact on the catalytic performance. The revision is as follows:

Page 11, line 8-16; Figure S29 and 30

To further corroborate the synergistic effect of Cr ions on the enhanced OER activity, we considered two more cases with different Ru concentration and structures, as shown in Figure S29. In the first one, there are only surface Ru and sub-surface Cr ions, in which the energy barrier for RDS was calculated to be 1.81 eV. In the other case, one Cr cation on CrO₂(110) surface was replaced by Ru, yielding a reduced RDS barrier of 1.75 eV, which is lower than that of original RuO₂ surface. On the other hand, we also investigated two relatively less stable surfaces, namely (200) and (101) facets, which were observed in HR-TEM image (Figure 3d). In a previous experimental study,⁶⁵ these two surfaces of RuO₂ were found to be more active. Our DFT results (Figure S30) are consistent with this work, and the corresponding energy barriers of RDS on (200) and (101) surfaces were calculated to be 1.79 and 1.96 eV, respectively. In CrO₂-RuO₂ system, these two barriers were further decreased by 0.04 and 0.02 eV, respectively.

Figure S29. Energy profiles of OER processes on CrO₂(110) surfaces with (a) coated RuO₂ layer and (b) doped Ru atom in surface. The simulated models are depicted in insets. Color code: red, O; cyan, Ru; gray, Cr.

Figure S30. The slab models of (a) (200) and (b) (101) facets of rutile crystal. Energy profiles of OER processes on (c) RuO₂(200), (d) RuO₂-CrO₂(200), (e) RuO₂(101) and (f) RuO₂-CrO₂(101) surfaces.

Table S8. Surface energy of a series of facets of RuO₂.

Facet	Surface energy / eV
(110)	1.41

(101)	1.50
(200)	1.67
(001)	1.79
(111)	1.99
(210)	2.34
(211)	2.48

Q4. There is a pre-oxidation peak appearing around 1.3 V at higher Ru content. What does that peak correspond to?

Our response: This corresponds to the pre-oxidation of Cr in the lattice of CrO₂-RuO₂ solid solution, because the chemical states of Cr is slightly shifted to lower valance due to the electron-withdrawing effect of Cr in the lattice of RuO₂-CrO₂ solid solution as demonstrated by the EXAFS, XPS experiments and DFT calculations. We thank the reviewer for pointing out this peak. Actually, this is another evidence of the formation of solid solution, instead of separated phases as claimed by the 2nd reviewer.

Q5. Since their particle sizes are small and the powder is derived from porous framework materials, the authors should also report the ECSA, roughness factor, and preferably report current density with respect to the ECSA (specific current density).

Our response: As suggested, we calculated the ECSA, roughness factor and plotted the LSV with respect to the ECSA (Figure S18-20, Table S6). The results show that the enhanced activity of OER performance of CrO₂-RuO₂ solid solution is not just increased by ECSA, and the intrinsic activity arising from the lattice Cr ions plays a more important role. The revision is as follows:

Page 7, line 6-10

In addition, we further calculated the electrochemically active surface area (ECSA), roughness factor of Cr_{0.6}Ru_{0.4}O₂ (550) and RuO₂ electrode, and plotted the LSVs with respect to the ECSA (Figure S18-20, Table S6). The results show that the enhanced activity of OER performance of CrO₂-RuO₂ solid solution is not just increased by ECSA, the intrinsic activity arising from the Cr ions plays a more important role.

Figure S18. (Left) CVs of $\text{Cr}_{0.6}\text{Ru}_{0.4}\text{O}_2$ collected at various scan rates (20, 60, 100, 140 and 180 mV s^{-1}); (Right) Capacitive current at 1.26 V (vs. RHE) against the scan rate and the corresponding C_{DL} value.

Figure S19. (Left) CVs of RuO_2 collected at various scan rates (20, 60, 100, 140 and 180 mV s^{-1}); (Right) Capacitive current at 1.26 V (vs. RHE) against the scan rates and the corresponding C_{DL} value.

Figure S20. ECAS based LSVs of $\text{Cr}_{0.6}\text{Ru}_{0.4}\text{O}_2(550)$ and RuO_2 .

Table S6. ECSA parameters for $\text{Cr}_{0.6}\text{Ru}_{0.4}\text{O}_2(550)$ and RuO_2 .

Catalyst	C_{DL}	ECSA	RF
$\text{Cr}_{0.6}\text{Ru}_{0.4}\text{O}_2(550)$	2.58 mF	73.7 cm^2	1043
RuO_2	0.32 mF	9.1 cm^2	129

Q6. The discussion part of the manuscript is rather short and unsatisfactory. Except for mentioning that inclusion of Cr reduces e-density around Ru and increases the oxidation state, and enhancing the catalytic activity of a Cr-rich Ru-poor compound, it doesn't really go into the details of how this can be further used in a predictive way to design better catalysts. Ruthenates are well known stoichiometric compounds, and such a discussion can help the readers gain knowledge to advance further in this field.

Our response: Thanks for your valuable suggestion. We have thus rewritten the discussion and conclusion part. The presence of Ru plays the key role to induce the formation of rutile-structured CrO_2 . This is because RuO_2 and CrO_2 share the same rutile structure and have similar lattice constants, and thus the presence of Ru would induce the formation of $\text{RuO}_2\text{-CrO}_2$ solid solution. In light of the present study,

we proposed that the partial charge on Ru might be a descriptor to predict the performance of OER of solid solution. Herein we have investigated and screened other possible MO₂ materials in rutile structure, and found that MnO₂ can also enhance the catalytic performance of RuO₂. CrO₂-RuO₂ should be one of the best combinations for this kind of solid solutions. In contrast, the reductive tetravalent cations, such as Nb and W, lead to electron accumulation on Ru and their corresponding solid solutions are expected to have lower OER activity. In addition to the formation of solid solution, we also anticipate that doping RuO₂ with highly oxidizing metal ions, such as Ce⁴⁺, is another viable strategy to improve the OER activity.

The revision is as follows:

Page 11, the last paragraph; Page 12, line 1-4; Table S9

In summary, by using Cr-based MOF, we have developed a cost effective rutile Cr_{0.6}Ru_{0.4}O₂ electrocatalyst with superior OER activity and stability in acidic media. Our experimental results and DFT calculations revealed the profound influence of Cr on the OER performance. The enhanced stability is related to the lower occupation at the Fermi level, while the higher activity results from the altered electronic structures. The calculated free energy diagrams for OER further demonstrates a lower energy barrier for the formation of *OOH, which is the RDS. On the other hand, Ru plays the key role to induce the formation of rutile-structured CrO₂ and thus CrO₂-RuO₂ solid solutions because RuO₂ and CrO₂ share the same rutile structure and have similar lattice constants. Note that direct annealing of the MIL-101(Cr) precursor only led to an inactive product of Cr₂O₃. These findings and results open a new route to design highly active, stable and relatively low-cost electrocatalysts for OER in acidic media. To shed light on the further optimization, we have investigated and screened a series of possible rutile-like MO₂-RuO₂ systems (based on the M₃Ru₃O₁₆ model), in which M is a tetravalent cation. In light of the altered electronic structures in CrO₂-RuO₂, we propose that the electron withdrawing on Ru ions can facilitate water oxidation and oxygen evolution. The calculated parameters of the cells and partial charges on Ru are summarized in Table S9. It is noteworthy that the partial charge on Ru ions in CrO₂-RuO₂ is found to be the most positive. Interestingly, we found that MnO₂ can also form solid solution with RuO₂ and possess good OER performance, owing to the similar cell parameter with that of RuO₂ and the highly positive partial charge on Ru (1.88 e). In contrast, we found that the reductive tetravalent cations, such as Nb and W, lead to electron accumulation on Ru. As a result, their corresponding solid solutions are expected to have lower OER activity. In addition to the formation of solid solution, we also anticipate that doping RuO₂ with highly oxidizing metal ions, such as Ce⁴⁺, is another viable strategy to improve the OER activity. On the other hand, our preparation method for chromium ruthenium oxides solid solution

electrocatalyst can be extended to prepare other rutile-structured electrocatalysts, such as the potentially active manganese ruthenium oxide and vanadium ruthenium oxide, or even non-Pt-group metal materials, such as chromium manganese oxide.

Table S9. Screening of rutile-like RuO₂-MO₂ systems.

	n(valence e)	a / Å	c / Å	Bader charge / e
RuO ₂	-	4.497	3.115	1.73
M=				
Ti	4	4.643	2.961	1.79
V	5	4.521	3.053	1.77
Cr	6	4.535	3.009	1.92
Mn	7	4.495	3.087	1.88
Ge	4	4.496	3.026	1.71
Nb	5	4.680	3.118	1.60
Mo	6	4.606	3.152	1.71
Rh	9	4.501	3.109	1.82
Sn	4	4.650	3.211	1.78
W	6	4.592	3.198	1.52
Pb	6	4.702	3.245	1.76

Reviewers' Comments:

Reviewer #1:

Remarks to the Author:

I believe the authors have attempted to address the reviewer comments adequately, but there are a few outstanding technical concerns the authors should address. Of perhaps most technical importance is the concerns raised by reviewers concerning the evidence for the existence of a simple substitutional solid solution. This is of key importance to the paper, so the authors should definitively address these concerns and explicitly address any evidence of mixed-phase systems.

Please find a few detailed comments below:

1. I agree with previous reviewers that the evidence for solid solution formation was not adequately addressed by the authors. The authors demonstrate that there is a shift in a single peak position with changing Ru loading—while that is certainly expected for a solid solution, we would actually expect ALL peaks to shift by an appropriate amount in accordance with Vegard's law. However, when looking at the data in Figure 5f, it actually appears that all peak positions shifted to lower 2θ values (and thus higher interplanar spacing) as the crystal went from low to higher Ru loading to $\text{Cr}_{0.6}\text{Ru}_{0.4}\text{O}_2$, not just one. I think the authors highlighted only one, but it seemed to me that they were all shifting which is consistent with a lattice change. It would be easy for the authors to demonstrate if this was true or not plotting the calculated lattice parameter calculated from multiple peaks in the pXRD (perhaps the peaks at $2\theta \approx 27^\circ$, $2\theta \approx 36^\circ$ and $2\theta \approx 55^\circ$) vs the x in $\text{Cr}_{1-x}\text{Ru}_x\text{O}_2$ and fitting the data to a straight line. The authors could then overlay this with a model with Vegard's law with the endpoints being the calculated lattice parameters for the CrO_2 and RuO_2 lattice parameters. If the Vegard's law model data overlays the author's data, then they have a simple substitutional solid solution and everything's good. If not, then the authors would need to reconsider their manuscript. The authors should also tabulate their pXRD data in the SI, including peak positions and the Miller indices of the corresponding lattice planes, so that it was clearer that there was a shift in peak position for every peak AND so other reviewers could corroborate their fits.

2. More concerning to me is the presence of Cr_2O_3 peak at low loading that slowly disappears at high loading. This very much suggests that, at least at low loadings, there is a mixed-phase system with an appreciable amount of Cr_2O_3 . $\text{Cr}_x\text{Ru}_{1-x}\text{O}_2$ should be isostructural with RuO_2 with all corresponding peaks shifting in 2θ according to Vegard's law – however, the disappearance of the peak at $2\theta \approx 33^\circ$, for instance, would not be expected and suggests appreciable Cr_2O_3 present. The authors should explicitly address the likelihood that a mixed phase exists in the low Ru-loading materials based on the pXRD patterns. Also, adding a RuO_2 pure phase spectra in Figure 5 would be helpful to determine which peaks arise due to possible contamination with a Cr_2O_3 phase and which peaks would be expected for the $\text{Cr}_x\text{Ru}_{1-x}\text{O}_2$ solid solution.

3. I agree with previous reviewers that the mapping analysis is not sufficient to show solid solution formation

4. The authors included a ball-milled mixture of $\text{RuO}_2/\text{Cr}_2\text{O}_3$ in response to an initial reviewer comment and use it to compare catalytic activity. However, the use of the ball-milled mixture introduces new concerns about conductivity. Ascertaining the conductivity of materials is important because conductivity will play a large role in relative activity—however, it is also very difficult to ascertain the conductivity of a material under catalytic conditions since the conductivity can change under applied potential. Boettcher and coworkers in particular have determined very creative ways of measuring conductivity for thin film OER catalysts using interdigitated arrays, but these techniques do not work for physically deposited nanoparticle/binder composite films. Historically, adding carbon sources into the ball-milled mixture could increase the resulting film's activity and therefore activity, but this would only raise additional questions concerning the type of carbon used and how this affects the activity measurements. In addition, it's unclear that the

surface area and number of active sites will be consistent between the nanoparticle films and the ball-milled mixture.

I actually think it was sufficient previously for the authors to show that their mixed phase outperformed the Cr₂O₃ and RuO₂ single phase systems, and I would not have encouraged them to look at a mixed-phase ball-milled system due to the complexity in evaluating the operational conductivity and intrinsic activity of such systems. However, since the authors have included this new ball-milled data, they should explicitly address the limitations of this approach (in terms of conductivity and surface area) and acknowledge that is not necessarily a good comparison to their chromium-ruthenium oxides. The authors might look to some of the seminal work of Boettcher et al (among others) discussion the importance of conductivity in OER measurements. I think such a discussion of the shortcomings of the ball-milling experiments is sufficient for the manuscript.

5. I think additional explanation of Table S9, and in particular further discussion of how the parameters in Table S9 were determined, is warranted given the importance of these calculated partial charges to the discussion section.

Reviewer #2:

Remarks to the Author:

I have studied the revision and conclude that the authors have satisfactorily answered my concerns

Reviewer #3:

Remarks to the Author:

In this revised version of the manuscript, although the authors have tried to address the previous reviewers' comments, the original concerns still remain and in some cases they have become more contradictory.

- It seems that that the authors are not very well-versed with the concepts of solid solution and doping etc. The kind of experimental evidence they provide to substantiate their claims of CrO₂-RuO₂ solid solution formation is not the right one nor is it conclusive. For example, how does the pXRD peak "shifting to the left" confirm that it has RuO₂ in solid solution? A solid solution of RuO₂-CrO₂ is domains of the individual oxides mixed homogeneously in a single grain of of the composite product. This will not typically lead to shift of only one pXRD peak.

- Drawing a line and saying that the pXRD peak has shifted because of RuO₂ without even mentioning the hkl value of the peak and how that shift signifies presence of RuO₂, is neither insightful nor correct.

- The mapping analysis with extremely poor resolution and showing a very small region of the sample (<5 nm ROI) is not very helpful in ascertaining formation of solid solution.

- If the pre-oxidation peak is due to Cr, why is it not visible as Cr₂O₃ which has lower oxidation state?

- How does the pre-oxidation peak behave with respect to cycling, i.e. does it shift or lose intensity?

- In the ball-milled mixture of RuO₂ and CrO₂ did the authors also use a carbonaceous additive since that would also cause a difference. In MOF-derived samples, there is inherently a carbonaceous matrix created which will lead to better electron transport and better activity. In the

absence of such an additive, it can be expected that the ball-milled powder will show less activity.

- It is not clear how did the authors generate the data reported in Table S7. Did they consider the neutral metal atoms (assuming from the number of valence electrons reported in the table)? Have they actually compared the electronegativity values of the metal ions in consideration with that of Ru to validate their claims? Even within the rutile structures, distortions and other variations in their crystal structure will lead to various effects in the bond-valence sum calculations, and the data reported in Table S7 is highly dubious.

- Overall, this is a manuscript which raises more questions than answers with each round of revision, which is indicative of the fact that more research is needed into the underlying problem. The authors need to stand back and look at the validity of their claims from actual experimental evidence and not from reported literature.

Response to reviewer 1#

Q1: I agree with previous reviewers that the evidence for solid solution formation was not adequately addressed by the authors. The authors demonstrate that there is a shift in a single peak position with changing Ru loading—while that is certainly expected for a solid solution, we would actually expect ALL peaks to shift by an appropriate amount in accordance with Vegard's law. However, when looking at the data in Figure 5f, it actually appears that all peak positions shifted to lower 2-theta values (and thus higher interplanar spacing) as the crystal went from low to higher Ru loading to $\text{Cr}_{0.6}\text{Ru}_{0.4}\text{O}_2$, not just one. I think the authors highlighted only one, but it seemed to me that they were all shifting which is consistent with a lattice change. It would be easy for the authors to demonstrate if this was true or not by plotting the calculated lattice parameter calculated from multiple peaks in the pXRD (perhaps the peaks at $2\theta \approx 27^\circ$, $2\theta \approx 36^\circ$, $2\theta \approx 55^\circ$) vs the x in $\text{Cr}_{1-x}\text{Ru}_x\text{O}_2$ and fitting the data to a straight line. The authors could then overlay this with a model with Vegard's law with the endpoints being the calculated lattice parameters for the CrO_2 and RuO_2 lattice parameters. If the Vegard's law model data overlays the author's data, then they have a simple substitutional solid solution and everything's good. If not, then the authors would need to reconsider their manuscript. The authors should also tabulate their pXRD data in the SI, including peak positions and the Miller indices of the corresponding lattice planes, so that it was clearer that there was a shift in peak position for every peak AND so other reviewers could corroborate their fits.

Our response: Thanks a lot for the valuable suggestion. Firstly, for clearance, we provided a direct comparison of $\text{Cr}_{0.6}\text{Ru}_{0.4}\text{O}_2$ pattern with standard CrO_2 and RuO_2 (Figure S3a), which clearly demonstrates that the all the peaks are shifted and the positions are located between those of CrO_2 and RuO_2 . Therefore, as CrO_2 and RuO_2 share the same rutile structure and have similar lattice constants (Table R1), they can readily generate a solid solution (Cr and Ru atoms are uniformly distributed in the crystal lattice) in both theory and experiment.

Table R1. Lattice parameters of standard CrO_2 and RuO_2 (JCPDS No. 09-0332 and 43-1027).

Formula	RuO_2	CrO_2
Crystal system	tetragonal	tetragonal
Space group	P 42/m n m	P 42/m n m
a/Å	4.499	4.421

b/Å	4.499	4.421
c/Å	3.107	2.916
α	90°	90°
β	90°	90°
γ	90°	90°

Figure S3. Experimental PXR D data vs. (a) standard CrO₂ and RuO₂ (JCPDS No. 09-0332 and 43-1027) and (b) simulated Cr_{0.6}Ru_{0.4}O₂ structure. The PXR D is acquired by a very slow scan with a scan step of 0.005° and a scan rate of 4 second per step.

Furthermore, as suggested, we calculated the lattice parameters of Cr_{1-x}Ru_xO₂ with solid solution as the major or pure phase (i.e. Cr_{0.72}Ru_{0.28}O_{2- δ} , Cr_{0.67}Ru_{0.33}O₂ and Cr_{0.6}Ru_{0.4}O₂), to check whether the results obey the Vegard's law or not. The PXR D data were provided in the supporting information, and the positions of peaks (the peaks at ~27° and ~36°) used for the calculation were highlighted. The calculation results are as follows:

$$\text{Cr}_{0.72}\text{Ru}_{0.28}\text{O}_{2-\delta}, a=b=4.483 \text{ \AA}, c=2.955 \text{ \AA}$$

$$\text{Cr}_{0.67}\text{Ru}_{0.33}\text{O}_2, a=b=4.495 \text{ \AA}, c=2.974 \text{ \AA}$$

Cr_{0.6}Ru_{0.4}O₂, a=b=4.497 Å, c=2.983 Å. For this sample, as we have presented in our manuscript, the lattice parameters by Rietveld refinement based on a high quality PXR D with a very slow scan were: a=b=4.495 Å, c=2.994 Å. We also added this point in the following plot

(lattice parameter c vs. Ru content), but for consistence with other samples, we didn't use this data for the linear fitting.

The standard parameters of RuO_2 and CrO_2 :

RuO_2 , $a=b=4.499 \text{ \AA}$, $c=3.107 \text{ \AA}$

CrO_2 , $a=b=4.421 \text{ \AA}$, $c=2.916 \text{ \AA}$

Thus, we plot the lattice parameter c vs. Ru content (x in $\text{Cr}_{1-x}\text{Ru}_x\text{O}_2$):

Figure S23. Relationship between the lattice parameter c and Ru content. The red point was the data of $\text{Cr}_{0.6}\text{Ru}_{0.4}\text{O}_2$ obtained by Rietveld refinement based on a high quality PXRD with a very slow scan. For consistent with other samples, this point wasn't used for the linear fitting.

This quasi-linear relationship is in good agreement with Vegard's law. We note that, although the shift of the a parameter shows the same trend as the c parameter when the Ru content increases, there is a deviation for the a parameter according to the Vegard's law. This deviation was possibly due to the little difference of a parameter between RuO_2 ($a=4.499$) and CrO_2 ($a=4.421$), and/or the existence of defects in the lattice along a axis (*Appl. Phys. A* **2010**, 99,

189-195; *J. Cryst. Growth* **2006**, 287, 134–138). In addition, we note that, for non-cubic crystal system, in some cases (for example, *Int J Mater. Res.* **2007**, 98, 776-779; *J. Mater. Chem. A* **2018**, 6, 15170-15181), not all lattice parameters of a solid solution vary linearly with the composition. The above Figures and some of the discussion were included and highlighted in the main text and supporting information.

Q2: More concerning to me is the presence of Cr_2O_3 peak at low loading that slowly disappears at high loading. This very much suggests that, at least at low loadings, there is a mixed-phase system with an appreciable amount of Cr_2O_3 . $\text{Cr}_x\text{Ru}_{1-x}\text{O}_2$ should be isostructural with RuO_2 with all corresponding peaks shifting in 2θ according to Vegard's law. However, the disappearance of the peak at $2\theta \approx 33^\circ$, for instance, would not be expected and suggests appreciable Cr_2O_3 present. The authors should explicitly address the likelihood that a mixed phase exists in the low Ru-loading materials based on the pXRD patterns. Also, adding a RuO_2 pure phase spectra in Figure 5 would be helpful to determine which peaks arise due to possible contamination with a Cr_2O_3 phase and which peaks would be expected for the $\text{Cr}_x\text{Ru}_{1-x}\text{O}_2$ solid solution.

Our response: Thanks for your comment. We have thus added the pXRD pattern of pure RuO_2 as a reference. The peak at $2\theta \approx 33^\circ$ is a characteristic of Cr_2O_3 . As presented in page 7, the last paragraph, we have emphasized the existence of Cr_2O_3 in the samples with low Ru content. For $\text{Cr}_{0.72}\text{Ru}_{0.28}\text{O}_{2-\delta}$ with higher Ru content, the CrO_2 - RuO_2 solid solution turn to be the major phase, and for $\text{Cr}_{0.67}\text{Ru}_{0.33}\text{O}_2$ and $\text{Cr}_{0.6}\text{Ru}_{0.4}\text{O}_2$, pure phase of CrO_2 - RuO_2 solid solution was formed.

Q3: I agree with previous reviewers that the mapping analysis is not sufficient to show solid solution formation.

Our response: We agree that these mapping data (EELS and EDS maps) in a small region cannot directly prove the formation of solid solution. Therefore, we also presented the EDS mapping image in a much larger region (Figure S10, ~ 100 nm), which can clearly show that Cr and Ru elements are uniformly distributed in the as-synthesized nanoparticles. We believe this can serve as an important side evidence for our conclusions.

Figure S10. HAADF-STEM image and EDS mapping of $\text{Cr}_{0.6}\text{Ru}_{0.4}\text{O}_2(550)$.

Q4: The authors included a ball-milled mixture of $\text{RuO}_2/\text{CrO}_2$ in response to an initial reviewer comment and use it to compare catalytic activity. However, the use of the ball-milled mixture introduces new concerns about conductivity. Ascertaining the conductivity of materials is important because conductivity will play a large role in relative activity—however, it is also very difficult to ascertain the conductivity of a material under catalytic conditions since the conductivity can change under applied potential. Boettcher and coworkers in particular have determined very creative ways of measuring conductivity for thin film OER catalysts using interdigitated arrays, but these techniques do not work for physically deposited nanoparticle/binder composite films. Historically, adding carbon sources into the ball-milled mixture could increase the resulting film’s activity and therefore activity, but this would only raise additional questions concerning the type of carbon used and how this affects the activity measurements. In addition, it’s unclear that the surface area and number of active sites will be consistent between the nanoparticle films and the ball-milled mixture. Outperformed the Cr_2O_3 and RuO_2 single phase systems, and I would not have encouraged them to look at a mixed-phase ball-milled system due to the complexity in evaluating the operational conductivity and intrinsic activity of such systems. However, since the authors have included this new ball-milled data, they should explicitly address the limitations of this approach (in terms of conductivity and surface area) and acknowledge that is not necessarily a good comparison to their chromium-ruthenium oxides. The authors might look to some of the seminal work of Boettcher et al (among

others) discussion the importance of conductivity in OER measurements. I think such a discussion of the shortcomings of the ball-milling experiments is sufficient for the manuscript.

Our response: Thanks for your valuable suggestion. As suggested, we have added the discussion of the importance of conductivity in our manuscript. We do agree that it will not be a good comparison to our chromium-ruthenium oxides if some residual carbon species exist in our samples. Therefore, we have tried to detect whether there is some residual carbon in our products by Raman analysis. As shown in Figure S26, no visible signal of carbon can be observed. In addition, herein, we further performed thermogravimetric (TG) measurement of $\text{Cr}_{0.6}\text{Ru}_{0.4}\text{O}_2$ (450) (450 °C was the lowest annealing temperature used for $\text{Cr}_{1-x}\text{Ru}_x\text{O}_2$ samples) under air atmosphere. As shown in Figure S27, no visible weight loss due to carbon-loss was observed. So, we are confident that there is negligible residual carbon in our sample. Nevertheless, we added some carbonaceous additive (commercial acetylene black that has high conductivity) to mixed $\text{CrO}_2\text{-RuO}_2$, denoted as mixed $\text{CrO}_2\text{-RuO}_2/\text{C}$. For the mixed $\text{CrO}_2\text{-RuO}_2/\text{C}$ ink preparation, 4 mg of carbon black additive was added into the ink mixture (the mass of $\text{CrO}_2\text{-RuO}_2$ was 4 mg). As shown in Figure S28, the OER activity of mixed $\text{CrO}_2\text{-RuO}_2/\text{C}$ was enhanced after the addition of carbon black, but still lower than that of pure RuO_2 .

Figure S26. Raman spectroscopy of $\text{Cr}_{0.6}\text{Ru}_{0.4}\text{O}_2$ (450), $\text{Cr}_{0.6}\text{Ru}_{0.4}\text{O}_2$ (550) and $\text{Cr}_{0.6}\text{Ru}_{0.4}\text{O}_2$ (650).

Figure S27. TG profile of $\text{Cr}_{0.6}\text{Ru}_{0.4}\text{O}_2$ (450) in air with a heating rate of $5^\circ\text{C}/\text{min}$.

Figure S28. LSV curves of pure RuO_2 , mixed $\text{CrO}_2\text{-RuO}_2$ and $\text{CrO}_2\text{-RuO}_2/\text{C}$.

The above Figures (Figure S27 and 28) were also included and highlighted in supporting information. The discussion added in main text is as follows:

Page 8, the last 9 lines

Note that the conductivity plays an important role in the OER process, and it might not be a good comparison to the chromium-ruthenium oxides if some residual carbon species inherent

from MOF precursor exist in our samples. Therefore, we further preformed Raman and thermogravimetric (TG) measurement of the samples to detect the residual carbon. As shown in Figures S26 and 27, no signal of the residual carbon can be observed. Nevertheless, we added carbonaceous additive (commercial acetylene black that has high conductivity) to the mixed $\text{CrO}_2\text{-RuO}_2$, denoted as mixed $\text{CrO}_2\text{-RuO}_2/\text{C}$. As shown in Figure S28, the OER activity of mixed $\text{CrO}_2\text{-RuO}_2/\text{C}$ was enhanced after the addition of carbon black, but still lower than that of pure RuO_2 , indicating the important synergistic effect of Cr^{4+} role as a participating lattice ion.

Q5: I think additional explanation of Table S9, and in particular further discussion of how the parameters in Table S9 were determined, is warranted given the importance of these calculated partial charges to the discussion section.

Our response: We have added the explanation of Table S9, we also added a column of electronegativity in Table S9. The revision is as follows:

Page 15, the last paragraph

Moreover, we have also attempted to screen a series of potential solid solutions for further prediction, which are composed of RuO_2 and other rutile-like oxides, including TiO_2 , VO_2 , CrO_2 , MnO_2 , GeO_2 , NbO_2 , MoO_2 , RhO_2 , SnO_2 , WO_2 and PbO_2 . The cell sizes of the bulk models were allowed to relax in the calculations at the aforementioned level. The calculated theoretical lattice parameters are listed in Table S9. Ideally, the closer cell parameters for the two MO_2 crystals, the higher possibility the solid solution can be formed. Besides, the atomic charges on Ru atoms in these solid solution systems were calculated based on the Bader charge analysis. Here the more positive partial charge compared with Ru in bulk RuO_2 indicates electron transfer from Ru to other metals as well as strengthened oxidizing ability to promote OER performance. For comparison, the number of valence electron and electronegativity of various metals are labeled in Table S9. However, it seems that they have little influence on the electronic distribution on Ru.

Response to reviewer 3#

Q1 : It seems that that the authors are not very well-versed with the concepts of solid solution and doping etc. The kind of experimental evidence they provide to substantiate their claims of CrO₂-RuO₂ solid solution formation is not the right one nor is it conclusive. For example, how does the pxd peak "shifting to the left" confirm that it has RuO₂ in solid solution? A solid solution of RuO₂-CrO₂ is domains of the individual oxides mixed homogeneously in a single grain of of the composite product. This will not typically lead to shift of only one pxd peak. Drawing a line and saying that the pxd peak has shifted because of RuO₂ without even mentioning the hkl value of the peak and how that shift signifies presence of RuO₂, is neither insightful nor correct.

Our response: Thanks for the question. We also appreciate the suggestion from Reviewer 1#. The slow scanned PXRD pattern as well as the refined lattice constant have been provided in Figure S3 and Table S1. The lattice constants of as-synthesized Cr_{0.6}Ru_{0.4}O₂ are between those of CrO₂ and RuO₂, which is an important characteristic of CrO₂ and RuO₂ solid solution. To further clarify our claim, we provide a direct comparison of Cr_{0.6}Ru_{0.4}O₂ pattern with standard CrO₂ and RuO₂ (Figure S3a), which clearly demonstrates that the all the peaks are shifted and the positions are located between those of CrO₂ and RuO₂. Therefore, as CrO₂ and RuO₂ share the same rutile structure and have similar lattice constants (Table R1), they can readily generate a solid solution (Cr and Ru atoms are uniformly distributed in the crystal lattice) in both theory and experiment.

Table R1. Lattice parameters of standard CrO₂ and RuO₂ (JCPDS No.09-0332 and 43-1027).

Formula	RuO ₂	CrO ₂
Crystal system	tetragonal	tetragonal
Space group	P 42/m n m	P 42/m n m
a/Å	4.499	4.421
b/Å	4.499	4.421
c/Å	3.107	2.916
α	90°	90°
β	90°	90°
γ	90°	90°

Figure S3. Experimental PXRD data vs. (a) standard CrO₂ and RuO₂ (JCPDS No. 09-0332 and 43-1027) and (b) simulated Cr_{0.6}Ru_{0.4}O₂ structure. The PXRD is acquired by a very slow scan with a scan step of 0.005° and a scan rate of 4 second per step.

We only highlighted the peak at ~36° because this is a suggestion from reviewer 2# in the second round revision (see Question 1 of reviewer 2# in the second round). Actually, as shown in Figure S3b, the PXRD peaks of CrO₂ and RuO₂ are very close but with a major difference at ~36°. We have added the hkl value of the peak in Figure 5f, and also highlighted the other two main peaks (101) and (211). The PXRD pattern of pure RuO₂ is also presented for clearance as suggested by reviewer 1#.

Figure 5f. PXRD patterns, Cr_2O_3 powder was obtained from directly annealing pure MIL-101 (Cr).

In addition, as kindly suggested by reviewer 1#, we calculated the lattice parameters of $\text{Cr}_{1-x}\text{Ru}_x\text{O}_2$ with solid solution as the major or pure phase (i.e., $\text{Cr}_{0.72}\text{Ru}_{0.28}\text{O}_{2-\delta}$, $\text{Cr}_{0.67}\text{Ru}_{0.33}\text{O}_2$ and $\text{Cr}_{0.6}\text{Ru}_{0.4}\text{O}_2$), to check whether the results obey the Vegard's law or not. The PXRD data were provided in supporting information, and the positions of peaks used for the calculation were highlighted. The calculation results are as follows:

$$\text{Cr}_{0.72}\text{Ru}_{0.28}\text{O}_{2-\delta}, a=b=4.483 \text{ \AA}, c=2.955 \text{ \AA}$$

$$\text{Cr}_{0.67}\text{Ru}_{0.33}\text{O}_2, a=b=4.495 \text{ \AA}, c=2.974 \text{ \AA}$$

$\text{Cr}_{0.6}\text{Ru}_{0.4}\text{O}_2$, $a=b=4.497 \text{ \AA}$, $c=2.983 \text{ \AA}$. For this sample, as we have presented in our manuscript, the lattice parameters by Retiveld refinement based on a high quality PXRD with a very slow scan were: $a=b=4.495 \text{ \AA}$, $c=2.994 \text{ \AA}$. We also added this point in the following plot (lattice parameter c vs. Ru content), but for consistent with other samples, we didn't use this data for the linear fitting.

The standard parameters of RuO_2 and CrO_2 :

$$\text{RuO}_2, a=b=4.499 \text{ \AA}, c=3.107 \text{ \AA}$$

$$\text{CrO}_2, a=b=4.421 \text{ \AA}, c=2.916 \text{ \AA}$$

Thus, we plot the lattice parameter c vs. Ru content (x in $\text{Cr}_{1-x}\text{Ru}_x\text{O}_2$):

Figure S23. Relationship between the lattice parameter c and Ru content. The red point was the data of $\text{Cr}_{0.6}\text{Ru}_{0.4}\text{O}_2$ obtained by Retiveld refinement based on a high quality PXRD with a very slow scan. For consistent with other samples, this point wasn't used for the linear fitting.

This quasi-linear relationship is in good agreement with Vegard's law. We note that, although the shift of the a parameter shows the same trend as the c parameter when the Ru content increases, there is a deviation for the a parameter according to the Vegard's law. This deviation was possibly due to the little difference of a parameter between RuO_2 ($a=4.499$) and CrO_2 ($a=4.421$), and/or the existence of defects in the lattice along a axis (*Appl. Phys. A* **2010**, 99, 189-195; *J. Cryst. Growth* **2006**, 287, 134-138). In addition, we note that, for non-cubic crystal system, not all lattice parameters of a solid solution vary linearly with the composition (for example, *In.t J Mater. Res.* **2007**, 98, 776-779; *J. Mater. Chem. A* **2018**, 6, 15170-15181).

The above mentioned Figures (Figure S3, 24 and 5f) were also included and highlighted in main text or supporting information. The discussion added in main text is as follows:

...and the standard PXRD patterns of CrO₂ and RuO₂ were shown in Figure S3a for comparison.

Page 8, Line 1-9

Note that, all the peaks shift slightly to the left side as the Ru content increases, which is a characteristic of RuO₂-CrO₂ solid solution. We further calculated the lattice parameters of Cr_{1-x}Ru_xO₂ with solid solution as the major or pure phase (i.e., Cr_{0.72}Ru_{0.28}O_{2-δ}, Cr_{0.67}Ru_{0.33}O₂ and Cr_{0.6}Ru_{0.4}O₂). As shown in Figure S23, the c parameter varies nearly linearly with the composition. This quasi-linear relationship is in good agreement with the Vegard's law. Although the shift of the a parameter shows the same trend as the c parameter when the Ru content increases, there is a deviation for the a parameter according to the Vegard's law. This deviation was possibly due to the little difference of a parameter between RuO₂ (a=4.499 Å) and CrO₂ (a=4.421 Å), and/or the existence of some defects in the lattice along the a axis.^{45,46}

Q2 : The mapping analysis with extremely poor resolution and showing a very small region of the sample (<5 nm ROI) is not very helpful in ascertaining formation of solid solution.

Our response: The resolution of our **EELS** mapping image is ~4 Å (note that the square is a mode of the signal of EELS mapping), which is comparable to those published in literatures (for example, *Science* 2016, 352, 333-337; *Nat. Mater.* 2011, 11, 49-52). We also presented the EDS mapping images in a much larger region (Figure S10, ~100 nm), which clearly show that Cr and Ru elements are uniformly distributed in the as-synthesized nanoparticles. Although we do agree that these mapping data cannot directly prove the formation of solid solution, we believe this is an important side evidence for our conclusions.

Figure S10. HAADF-STEM image and EDS mapping of $\text{Cr}_{0.6}\text{Ru}_{0.4}\text{O}_2(550)$.

Q3: If the pre-oxidation peak is due to Cr, why is it not visible on Cr_2O_3 which has lower oxidation state? How does the pre-oxidation peak behave with respect to cycling, i.e. does it shift or lose intensity?

Our response: The pre-oxidation peak of the solid solution is originated from the valance change of Cr. However, the unnoticeable pre-oxidation peak of Cr_2O_3 in LSV is caused by the large crystal size and less active surface area as revealed by the sharp PXRD patterns (Figure S22). We further directly observed the morphology by TEM. As shown in Figure S24a, the particle size of Cr_2O_3 annealed at 450 °C was above 100 nm, which is 5 times higher than of $\text{Cr}_{1-x}\text{Ru}_x\text{O}_2$. Thus, by means of lowering the annealing temperature, we prepared Cr_2O_3 particles with much smaller size (less than 5 nm), as shown in the TEM image (Figure S24b). The LSV of the resulting Cr_2O_3 indeed showed a weak pre-oxidation peak at ~1.3 V. Note that the pre-oxidation peak position of Cr_2O_3 was slightly higher than that of $\text{Cr}_{1-x}\text{Ru}_x\text{O}_2$, possibly due to the synergic effect of Cr and Ru in $\text{Cr}_{1-x}\text{Ru}_x\text{O}_2$.

Figure S22. The PXRD pattern of the product by annealing pure MIL-101 (Cr) without loading RuCl₃ at 300 and 450 °C. The reference PXRD pattern of Cr₂O₃ is obtained from Jade 2004 (JCPDS No. 06-0504).

Figure S24. TEM images of Cr₂O₃ by directly annealing MIL-101(Cr) at different temperatures: (a) 450 °C; (b) 300 °C.

As for the peak stability, we have tested for 10,000 cycles and found that the pre-oxidation peak lose intensity after cycling test (Figure 4c), which can be attributed to the leaching of

exposed Cr during the cycling. As we have mentioned in our manuscript (based on the ICP analysis), the surface Cr element would suffer a slight leaching (~8 %). However, although the surface atomic arrangement gradually changed, the OER performance nearly did not decay (Figure 4c).

Figure S25, LSV curve of Cr_2O_3 obtained from annealing MIL-101(Cr) at 300 °C.

The above Figures (Figure S22, 24 and 25) were also included and highlighted in supporting information. The discussion added in main text is as follows:

Page 8, line 9-17

It is noteworthy that there is a pre-oxidation peak of the solid solution samples, which can be ascribed to the pre-oxidation of Cr. No such pre-oxidation peak was observed on the Cr_2O_3 sample annealed at 450 °C, which can be attributed to its large particle size (Figure S24a) and relatively low active surface area that could cause low conductivity and activity. We thus prepared Cr_2O_3 with much smaller particle size (Figures S22 and 24b) by annealing MIL-101(Cr) at lower temperature (300 °C). Indeed, we also found this pre-oxidation peak on the Cr_2O_3 with less crystallinity (Figure S25), albeit the peak was weak. The position of pre-oxidation peak of Cr_2O_3 was slightly higher than that of $\text{Cr}_{1-x}\text{Ru}_x\text{O}_2$, which was possibly due to the synergistic effect of Cr and Ru in $\text{Cr}_{1-x}\text{Ru}_x\text{O}_2$.

Q4: In the ball-milled mixture of RuO₂ and CrO₂ did the authors also use a carbonaceous additive since that would also cause a difference. In MOF-derived samples, there is inherently a carbonaceous matrix created which will lead to better electron transport and better activity. In the absence of such an additive, it can be expected that the ball-milled powder will show less activity.

Our response: Thanks for your comment, we do agree conductivity plays an important role in the OER process, and it will be not a good comparison to our chromium-ruthenium oxides if some residual carbon species exist in our samples. Therefore, in our previous manuscript, we have tried to detect whether there is some residual carbon in our products by Raman analysis. As shown in Figure S26, we cannot find any visible signal of carbon. In addition, herein, we further performed thermogravimetric (TG) measurement of Cr_{0.6}Ru_{0.4}O₂ (450) (450 °C was the lowest annealing temperature used for Cr_{1-x}Ru_xO₂ samples) under air atmosphere. As shown in Figure 27, no visible weight loss due to carbon-loss was observed. So, we are confident that there is negligible residual carbon in our sample.

Figure S26. Raman spectroscopy of Cr_{0.6}Ru_{0.4}O₂ (450), Cr_{0.6}Ru_{0.4}O₂ (550) and Cr_{0.6}Ru_{0.4}O₂ (650).

Figure S27. TG profile of $\text{Cr}_{0.6}\text{Ru}_{0.4}\text{O}_2$ (450) in air with a heating rate of $5^\circ\text{C}/\text{min}$.

Nevertheless, as suggested, we added some carbonaceous additive (commercial acetylene black that has high conductivity) to mixed CrO_2 - RuO_2 , denoted as mixed CrO_2 - RuO_2/C . For the mixed CrO_2 - RuO_2/C ink preparation, 4 mg of carbon black additive was added into the mixture (the mass of CrO_2 and RuO_2 was 4 mg). As shown in Figure S28, the OER activity of mixed CrO_2 - RuO_2/C was enhanced after the addition, but still lower than that of pure RuO_2 .

Figure S28. LSV curves of pure RuO_2 , mixed CrO_2 - RuO_2 and CrO_2 - RuO_2/C .

The above Figures (Figure S27 and 28) were also included and highlighted in supporting information. The discussion added in main text is as follows:

Page 8, the last 9 lines

Note that the conductivity plays an important role in the OER process, and it might not be a good comparison to the chromium-ruthenium oxides if some residual carbon species inherent from MOF precursor exist in our samples. Therefore, we further preformed Raman and thermogravimetric (TG) measurement of the samples to detect the residual carbon. As shown in Figures S26 and S27, no signal of the residual carbon can be observed. Nevertheless, we added carbonaceous additive (commercial acetylene black that has high conductivity) to the mixed $\text{CrO}_2\text{-RuO}_2$, denoted as mixed $\text{CrO}_2\text{-RuO}_2/\text{C}$. As shown in Figure S28, the OER activity of mixed $\text{CrO}_2\text{-RuO}_2/\text{C}$ was enhanced after the addition of carbon black, but still lower than that of pure RuO_2 , indicating the important synergistic effect of Cr^{4+} role as a participating lattice ion.

Q5: It is not clear how did the authors generate the data reported in Table S9. Did they consider the neutral metal atoms (assuming from the number of valence electrons reported in the table)? Have they actually compared the electronegativity values of the metal ions in consideration with that of Ru to validate their claims? Even within the rutile structures, distortions and other variations in their crystal structure will lead to various effects in the bond-valence sum calculations, and the data reported in Table S9 is highly dubious.

Our response: We have added the explanation of Table S9, we also added a column of electronegativity in Table S9. The revision is as follows:

Page 15, the last paragraph

Moreover, we have also attempted to screen a series of potential solid solutions for further prediction, which are composed of RuO_2 and other rutile-like oxides, including TiO_2 , VO_2 , CrO_2 , MnO_2 , GeO_2 , NbO_2 , MoO_2 , RhO_2 , SnO_2 , WO_2 and PbO_2 . The cell sizes of the bulk models were allowed to relax in the calculations at the aforementioned level. The calculated theoretical lattice parameters are listed in Table S9. Ideally, the closer cell parameters for the two MO_2 crystals, the higher possibility the solid solution can be formed. Besides, the atomic charges on Ru atoms in these solid solution systems were calculated based on the Bader charge analysis. Here the higher

positive partial charge compared with Ru in bulk RuO₂ indicates that the Ru ion in solid solution would donate electrons to other metals, and accordingly its oxidizing ability is strengthened to promote OER performance. For comparison, the number of valence electron and electronegativity of various metals are labeled in Table S9. However, it seems that they have trivial influence on the electronic distribution on Ru.

Reviewers' Comments:

Reviewer #1:

Remarks to the Author:

The authors have adequately addressed my concerns. I believe their Vegard's law analysis of the pXRD data sufficiently supports their claim that the material is a substitutional solid solution. In addition, I believe they addressed my concerns about Cr₂O₃ contamination adequately and added appropriate language regarding the conductivity issues in the ball-mill experiments. I support this manuscripts publication in Nature Communications.

Reviewer #1

The authors have adequately addressed my concerns. I believe their Vegard's law analysis of the pXRD data sufficiently supports their claim that the material is a substitutional solid solution. In addition, I believe they addressed my concerns about Cr₂O₃ contamination adequately and added appropriate language regarding the conductivity issues in the ball-mill experiments. I support this manuscripts publication in Nature Communications.

Our response: We thank the reviewer for the careful reading and positive comments.